# Two-way coupled meteorology and air quality models in Asia: a systematic review and meta-analysis of impacts of aerosol feedbacks on meteorology and air quality

Chao Gao[1], Aijun Xiu[1, *], Xuelei Zhang[1, *], Qingqing Tong[1], Hongmei Zhao[1], Shichun Zhang[1], Guangyi Yang[1, 2], and Mengduo Zhang[1, 2]

[1]Key Laboratory of Wetland Ecology and Environment, Northeast Institute of Geography and Agroecology, Chinese Academy of Sciences, Changchun, 130102, China
[2]University of Chinese Academy of Sciences, Beijing, 100049, China
Correspondence to: A.J. Xiu (xiuaijun@iga.ac.cn) & X.L. Zhang (zhangxuelei@iga.ac.cn)

## Abstract

Atmospheric aerosols can exert influence on meteorology and air quality through aerosol-radiation interactions (ARI) and aerosol-cloud interactions (ACI) and this two-way feedback has been studied by applying two-way coupled meteorology and air quality models. As one of regions with high aerosol loading in the world, Asia has attracted many researchers to investigate the aerosol effects with several two-way coupled models (WRF-Chem, WRF-CMAQ, GRAPES-CUACE, WRF-NAQPMS and GATOR-GCMOM) over the last decade. This paper attempts to offer bibliographic analysis regarding the current status of applications of two-way coupled models in Asia, related research focuses, model performances and the effects of ARI or/and ACI on meteorology and air quality. There are total 160 peer-reviewed articles published between 2010 and 2019 in Asia meeting the inclusion criteria, with more than 79 % of papers involving the WRF-Chem model. The number of relevant publications has an upward trend annually and East Asia, India, China, as well as the North China Plain are the most studied areas. The effects of ARI and both ARI and ACI induced by natural aerosols (particularly mineral dust) and anthropogenic aerosols (bulk aerosols, different chemical compositions and aerosols from different sources) are widely investigated in Asia. Through the meta-analysis of surface meteorological and air quality variables simulated by two-way coupled models, the model performance affected by aerosol feedbacks depends on different variables, simulation time lengths, selection of two-way coupled models, and study areas. Future research perspectives with respect to the development, improvement, application, and evaluation of two-way coupled meteorology and air quality models are proposed.

## 1 Introduction

Atmospheric pollutants can affect local weather and global climate via many mechanisms as extensively summarized in the Intergovernmental Panel on Climate Change (IPCC) reports (IPCC, 2007, 2014, 2021), and also exhibit impacts on human health and ecosystems (Lelieveld et al., 2015; Wu and Zhang, 2018). Atmospheric pollutants can modify the radiation energy balance, thus influence meteorological conditions (Gray et al., 2010; Yiğit et al., 2016). Compared to other climate agents, the short-lived and localized aerosols could induce changes in meteorology and climate through aerosol-radiation interactions (ARI, Tremback et al., 1986; Satheesh and Moorthy, 2005) and aerosol-cloud interactions (ACI, Martin and Leight, 1949; Lohmann and Feichter, 2005) or both (Sud and Walker, 1990; Haywood and Boucher, 2000). ARI (previously known as direct effect and semi-direct effect) are based on scattering and absorbing solar radiation by aerosols as well as cloud dissipation by heating (McCormick and Ludwig, 1967; Ackerman et al., 2000; Koch and Del Genio, 2010; Wilcox, 2012), and ACI (known as indirect effect) are concerned with aerosols altering albedo and lifetime of clouds (Twomey, 1977; Albrecht, 1989; Lohmann and Feichter, 2005). As our knowledge base of aerosol-radiation-cloud interactions that involve extremely complex physical and chemical processes has been expanding, accurately assessing the effects of these interactions still remains a big challenge (Rosenfeld et al., 2008, 2019; Fan et al., 2016; Kuniyal and Guleria, 2019).

The interactions between air pollutants and meteorology can be investigated by observational analyses and/or air quality models. So far, many observational studies using measurement data from a variety of sources have been conducted to analyze these interactions (Wendisch et al., 2002; Bellouin et al., 2008; Groß et al., 2013; Rosenfeld et al., 2019). Yu et al (2006) reviewed research work that adopted satellite and ground-based measurements to estimate the ARI-induced changes of radiative forcing and the associated uncertainties in the analysis. Yoon et al. (2019) analyzed the effects of aerosols on the radiative forcing based on the Aerosol Robotic Network observations and

demonstrated that these effects depended on aerosol types. On the other hand, since the uncertainties in ARI estimations were associated with ACI (Kuniyal and Guleria, 2019), the simultaneous assessments of both ARI and ACI effects were needed and had gradually been conducted via satellite observations (Sekiguchi et al., 2003; Quaas et al., 2008; Illingworth et al., 2015; Kant et al., 2019). In the early stages, observational studies of ACI effects were based on several cloud parameters mainly derived from surface-based microwave radiometer (Kim et al., 2003; Liu et al., 2003) and cloud radar (Feingold et al., 2003; Penner et al., 2004). Later on, with the further development of satellite observation technology and enhanced spatial resolution of satellite measurement comparing against traditional ground observations, the satellite-retrieved cloud parameters (effective cloud droplet radius, liquid water path (LWP) and cloud cover) were utilized to identify the ACI effects studies on cloud scale. (Goren and Rosenfeld, 2014; Rosenfeld et al., 2014). Moreover, in order to clarify whether aerosols affect precipitation positively or negatively, the effects of ACI on cloud properties and precipitation were widely investigated but with various answers (Andreae and Rosenfeld, 2008; Rosenfeld et al., 2014; Casazza et al., 2018; Fan et al., 2018). Analyses of satellite and/or ground observations revealed that increased aerosols could suppress (enhance) precipitation in drier (wetter) environments (Rosenfeld, 2000; Rosenfeld et al., 2008; Li et al., 2011b; Donat et al., 2016). Most recently, Rosenfeld et al. (2019) further used satellite-derived cloud information (droplet concentration and updraft velocity at cloud base, LWP at cloud cores, cloud geometrical thickness and cloud fraction) to single out ACI under a certain meteorological condition, and found that the cloudiness change caused by aerosol in marine low-level clouds was much greater than previous analyses (Sato and Suzuki, 2019). Despite the fact that aforementioned studies had significantly improved our understanding of aerosol effects, many limitations still exist, such as low temporal resolution of satellite data, low spatial resolution of ground monitoring sites and lack of vertical distribution information of aerosol and cloud (Yu et al., 2006; Rosenfeld et al., 2014; Sato and Suzuki, 2019).

Numerical models can also be used to study the interactions between air pollutants and meteorology. Air quality models simulate physical and chemical processes in the atmosphere (ATM) and are classified as offline and online models (El-Harbawi, 2013). Offline models (also known as traditional air quality models) require outputs from meteorological models to subsequently drive chemical models (Seaman, 2000; Byun and Schere, 2006; ENVIRON, 2008). Comparing to online models, offline models usually are computationally efficient but incapable of capturing two-way feedbacks between chemistry and meteorology (North et al., 2014). Online models or coupled models are designed and developed to consider the two-way feedbacks and attempted to accurately simulate both meteorology and air quality (Grell et al., 2005; Wong et al., 2012; Briant et al., 2017). Two-way coupled models can be generally categorized as integrated and access models based on whether using a coupler to exchange variables between meteorological and chemical modules (Baklanov et al., 2014). As Zhang (2008) pointed out, Jacobson (1994, 1997a) and Jacobson et al. (1996a) pioneered the development of a fully-coupled model named Gas, Aerosol, Transport, Radiation, General Circulation, Mesoscale, and Ocean Model (GATOR-GCMOM) in order to investigate all the processes related to ARI and ACI. Currently, there are three representative two-way coupled meteorology and air quality models, namely the Weather Research and Forecasting-Chemistry (WRF-Chem) (Grell et al., 2005), WRF coupled with Community Multiscale Air Quality (CMAQ) (Wong et al., 2012) and WRF coupled with a multi-scale chemistry-transport model for atmospheric composition analysis and forecast (WRF-CHIMERE) (Briant et al., 2017). The WRF-Chem is an integrated model that includes various chemical modules in the meteorological model (i.e., WRF) without using a coupler. For the remaining two models, which belong to access model, the WRF-CMAQ uses a subroutine called *aqprep* (Wong et al., 2012) as its coupler while the WRF-CHEMERE a general coupling software named Ocean Atmosphere Sea Ice Soil-Model Coupling Toolkit (Craig et al. 2017). With more growing interest in coupled models and their developments, applications and evaluations, two review papers thoroughly summarized the related works published before 2008 (Zhang, 2008) and 2014 (Baklanov et al., 2014). Zhang (2008) overviewed the developments and applications of five coupled models in the United States (US) and the treatments of chemical and physical processes in these coupled models with emphasis on the ACI related processes. Another paper presented a systematic review on the similarities and differences of eighteen integrated or access models in Europe and discussed the descriptions of interactions between meteorological and chemical processes in these models as well as the model evaluation methodologies involved (Baklanov et al., 2014). Some of these coupled models can not only be used

to investigate the interactions between air quality and meteorology at regional scales but also at global and hemispheric scales (Jacobson, 2001; Grell et al., 2011; Xing et al., 2015b; Mailler et al., 2017), but large scale studies were not included in the two review papers by Zhang (2008) and Baklanov et al. (2014). These reviews only focused on application and evaluation of coupled models in US and Europe but there is still no systematic review targeting two-way coupled model applications in Asia.

Compared to US and Europe, Asia has been suffering more severe air pollution in the past three decades (Bollasina et al., 2011; Rohde and Muller, 2015; Gurjar et al., 2016) due to the rapid industrialization, urbanization and population growth together with unfavorable meteorological conditions (Jeong and Park, 2017; Li et al., 2017a; Lelieveld et al., 2018). Then, the interactions between atmospheric pollution and meteorology in Asia, which have received a lot of attention from scientific community, are investigated using extensive observations and a certain number of numerical simulations (Wang et al., 2010; Li et al., 2016; Nguyen et al., 2019a). Based on airborne, ground-based, and satellite-based observations, multiple important experiments have been carried out to analyze properties of radiation, cloud and aerosols in Asia, as briefly reviewed by Lin et al. (2014b). Recent observational studies confirmed that increasing aerosol loadings play important roles in radiation budget (Eck et al., 2018; Benas et al., 2020), cloud properties (Dahutia et al., 2019; Yang et al., 2019), precipitation intensity along with vertical distributions of precipitation types (Guo et al., 2014, 2018). According to previous observational studies in Southeast Asia (SEA), Tsay et al. (2013) and Lin et al. (2014b) comprehensively summarized the spatiotemporal characteristics of biomass burning (BB) aerosols and clouds as well as their interactions. Li et al. (2016) analyzed how ARI or ACI influenced climate/meteorology in Asia utilizing observations and climate models. With regard to the impacts of aerosols on cloud, precipitation and climate in East Asia (EA), a detailed review of observations and modeling simulations has also been presented by Li et al. (2019c). Since the 2000s, substantial progresses have been made in the climate-air pollution interactions in Asia based on regional climate models simulations, which have been summarized by Li et al. (2016). Moreover, starting from year of 2010, with the development and availability of two-way coupled meteorology and air quality models, more and more modeling studies have been conducted to explore the ARI or/and ACI effects in Asia (Wang et al., 2010, 2014a; Sekiguchi et al., 2018; Nguyen et al., 2019a). In recent studies, a series of WRF-Chem and WRF-CMAQ simulations were performed to assess the consequences of ARI on radiative forcing, planetary boundary layer height (PBLH), precipitation, and fine particulate matter ($PM_{2.5}$) and ozone concentrations (Wang et al., 2014a; Huang et al., 2016; Sekiguchi et al., 2018; Nguyen et al., 2019b). Different from current released version of WRF-CMAQ model (based on WRF version 4.3 and CMAQ version 5.3.3) that only includes ARI, WRF-Chem with ACI (starting from WRF-Chem version 3.0, Chapman et al., 2009) has been implemented for analyzing the complicated aerosol effects that lead to variations of cloud properties, precipitations and $PM_{2.5}$ concentrations (Zhao et al., 2017; Liu et al., 2018c; Park et al., 2018; Bai et al., 2020). To quantify the individual or joint effects of ARI or/and ACI on meteorological variables and pollutants concentrations, several modeling studies have been performed in Asia (Zhang et al., 2015a, 2018; Ma et al., 2016; Chen et al., 2019a). In addition, model comparisons (including offline and online models) targeting EA have been carried out recently under the Model Inter-Comparison Study for Asia (MICS-Asia) phase III (Gao et al., 2018b; Chen et al., 2019b; Li et al., 2019a). As mentioned above, even though there are already several reviews regarding the observational studies of ARI or/and ACI (Tsay et al., 2013; Lin et al., 2014b; Li et al., 2016, 2019c) it is necessary to conduct a systematic review in Asia focusing on applications of two-way coupled meteorology and air quality models as well as simulated variations of meteorology and air quality induced by aerosol effects.

This paper is constructed as follows: Section 2 describes the methodology for literature searching, paper inclusion, and analysis; Section 3 summarizes the basic information about publications as well as developments and applications of coupled models in Asia and Section 4 provides the recent overviews of their research points. Sections 5 to 6 present systematic review and meta-analysis of the effects of aerosol feedbacks on model performance, meteorology and air quality in Asia. The summary and perspective are provided in Section 7.

**2 Methodology**
**2.1 Criteria and synthesis**

Since 2010, in Asia, regional studies of aerosol effects on meteorology and air quality based

on coupled models have been increasing gradually, therefore in this study we performed a systematic search of literatures to identify relevant studies from January 1, 2010 to December 31, 2019. In order to find all the relevant papers in English, Chinese, Japanese and Korean, we deployed serval science-based search engines, including Google Scholar, the Web of Science, the China National Knowledge Infrastructure, the Japan Information Platform for S&T Innovation, the Korean Studies Information Service System. The different keywords and their combinations for paper searching are as follows: (1) model-related keywords including "coupled model", "two-way", "WRF", "NU-WRF", "WRF-Chem", "CMAQ", "WRF-CMAQ", "CAMx", "CHIMERE", "WRF-CHIMERE" and "GATOR-GCMOM"; (2) effect-related keywords including "aerosol radiation interaction", "ARI", "aerosol cloud interaction", "ACI", "aerosol effect" and "aerosol feedback"; (3) air pollution-related keywords including "air quality", "aerosol", "PM2.5", "O3", "CO", "SO2", "NO2", "dust", "BC", "black carbon", "blown carbon", "carbonaceous", "primary pollutants"; (4) meteorology-related keywords including "meteorology", "radiation", "wind", "temperature", "specific humidity", "relative humidity", "planetary boundary layer", "cloud" and "precipitation"; (5) region-related keywords including "Asia", "East Asia", "Northeast Asia", "South Asia", "Southeast Asia", "Far East", "China", "India", "Japan", "Korea", "Singapore", "Thailand", "Malaysia", "Nepal", "North China Plain", "Yangtze River Delta", "Pearl River Delta", "middle reaches of the Yangtze River", "Sichuan Basin", "Guanzhong Plain", "Northeast China", "Northwest China" "East China", "Tibet Plateau", "Taiwan", "northern Indian", "southern Indian", "Gangetic Basin", "Kathmandu Valley".

After applying the search engines and the keywords combinations mentioned above, we found 946 relevant papers. In order to identify which paper should be included or excluded in this paper, following criteria were applied: (1) duplicate literatures were deleted; (2) studies of using coupled models in Asia with aerosol feedbacks turned on were included, and observational studies of aerosol effects were excluded; (3) publications involving coupled climate model were excluded. According to these criteria, not only regional studies, but also studies using the coupled models at global or hemispheric scales involving Asia or its subregions were included. Then, we carefully examined all the included papers and further checked the listed reference in each paper to make sure that no related paper was neglected. A flowchart that illustrated the detailed procedures applied for article identification is presented in Appendix Figure A1 (Note: Although the deadline for literature searching is 2019, any literature published in 2020 is also included.). There was a total of 160 publications included in our study.

## 2.2 Analysis method

To summarize the current status of coupled models applied in Asia and quantitatively analyze the effects of aerosol feedbacks on model performance as well as meteorology and air quality, we carried out a series of analyses based on data extracted from the selected papers. We firstly compiled the publication information of the included papers as well as the information regarding model name, simulated time period, study region, simulation design, and aerosol effects. Secondly, we summarized the important findings of two-way coupled model applications in Asia according to different aerosol sources and components to clearly acquire what are the major research focuses in past studies. Finally, we gathered all the simulated results of meteorological and air quality variables with/out aerosol effects and their statistical indices (SI). For questionable results, the quality assurance was conducted after personal communications with original authors to decide whether they were deleted and/or corrected. All the extracted publication and statistical information were exported into an Excel file, which was provided in Table S1. Moreover, we performed quantitative analyses of the effects of aerosol feedbacks through following steps. (1) We discussed whether meteorological and air quality variables were overestimated or underestimated based on their SI. Then, variations of the SI of these variables were further analyzed in detail with/out turning on ARI or/and ACI in two-way coupled models. (2) We investigated the SI of simulation results at different simulation time lengths and spatial resolutions in coupled models. (3) More detailed inter-model comparisons of model performance based on the compiled SI among different coupled models are conducted. (4) Differences in simulation results with/out aerosol feedbacks were grouped by study regions and time scales (yearly, seasonal, monthly, daily and hourly). Toward a better understanding of the complicated interactions between air quality and meteorology in Asia, the results sections in this paper are organized following above analysis methods (1) - (3) and represented in Section 5, and the results following method (4) were represented in Section 6. In addition, Excel and Python

were used to conduct data processing and plotting in this study.
**3    Basic overview**
**3.1   Summary of applications of coupled models in Asia**
A total of 160 articles were selected according to the inclusion criteria, and their basic
information was compiled in Table 1. In Asia, five two-way coupled models are applied to study the
ARI and ACI effects. These include GATOR-GCMOM, two commonly used models, i.e., WRF-
Chem and WRF-CMAQ, and two locally developed models, i.e., the global-regional assimilation
and prediction system coupled with the Chinese Unified Atmospheric Chemistry Environment
forecasting system (GRAPES-CUACE) and WRF coupled with nested air-quality prediction
modeling system (WRF-NAQPMS). 127 out of total 160 papers involved the applications of WRF-
Chem in Asia since its two-way coupled version was publicly available in 2006 (Fast et al., 2006).
WRF-CMAQ was applied in only 16 studies due to its later initial release in 2012 (Wong et al.,
2012). GRAPES-CUACE was developed by the China Meteorological Administration and
introduced in details in Zhou et al. (2008, 2012, 2016), then firstly utilized in Wang et al. (2010) to
estimate impacts of aerosol feedbacks on meteorology and dust cycle in EA. The coupled version
of WRF-NAQPMS was developed by the Institute of Atmospheric Physics, Chinese Academy of
Sciences and could improve the prediction accuracy of haze pollution in the North China Plain (NCP)
(Wang et al., 2014c). Note that GRAPES-CUACE and WRF-NAQPMS were only applied in China.
There were only three published papers about the applications of GATOR-GCMOM in Northeast
Asia (NEA), NCP and India. In the included papers, 93, 33, 31 studies targeted various areas in
China, EA and India, respectively. There were 79 papers regarding effects of ARI (7 health), 63 both
ARI and ACI (1 health) and 18 ACI. ACI studies were much less than ARI related ones, which
indicated that ACI related studies need to be paid with more attention in the future. Considering that
the choices of cloud microphysics and radiation schemes can affect coupled models' results (Baró
et al., 2015; Jimenez et al., 2016), these schemes used in the selected studies were also summarized
in Table 1. This table presents a concise overview of coupled models' applications in Asia with the
purpose of providing basic information regarding models, study periods and areas, aerosol effects,
scheme selections, and reference. More complete information is summarized Table S1 including
model version, horizontal resolution, vertical layer, aerosol and gas phase chemical mechanisms,
photolysis rate, PBL, land surface, surface layer, cumulus, urban canopy schemes, meteorological
initial and boundary conditions (ICs and BCs), chemical ICs and BCs, spin-up time, and
anthropogenic natural emissions.
It should be noted that in Table 1 there were four model inter-comparison studies that aimed at
evaluating model performance, identifying error sources and uncertainties, and providing optimal
model setups. By comparing simulations from two coupled models (WRF-Chem and Spectral
Radiation-Transport Model for Aerosol Species) (Takemura et al., 2003) in India (Govardhan et al.,
2016), it was found that the spatial distributions of various aerosol species (black carbon (BC),
mineral dust and sea salt) were similar with the two models. Based on the intercomparisons of WRF-
Chem simulations in different areas, Yang et al. (2017) revealed that aerosol feedbacks could
enhance $PM_{2.5}$ concentrations in the Indo-Gangetic Plain but suppress the concentrations in the
Tibetan Plateau (TP). Targeting China and India, Gao et al. (2018c) also applied the WRF-Chem
model to quantify the contributions of different emission sectors to aerosol radiative forcings,
suggesting that reducing the uncertainties in emission inventories were critical, especially for India.
Moreover, for the NCP region, Gao et al. (2018b) presented a comparison study with multiple online
models under the MICS-Asia Phase III and pointed out noticeable discrepancies in the simulated
secondary inorganic aerosols under heavy haze conditions and the importance of accurate wind
speed at 10 meters above surface (WS10) predictions by these models. Comprehensive comparative
studies for Asia have been emerging lately but are still limited, comparing to those for North
America and Europe, such as the Air Quality Model Evaluation International Initiative Phase II
(Brunner et al., 2015; Campbell et al., 2015; Im et al., 2015a, b; Kong et al., 2015; Makar et al.,
2015a, b; Wang et al., 2015b; Forkel et al., 2016).
Table 1. Basic information of coupled model applications in Asia during 2010-2019.

| No. | Model | Study period | Region | Aerosol effect | Short/long-wave radiation scheme | Microphysics scheme | Reference |
|-----|-------|--------------|--------|----------------|----------------------------------|---------------------|-----------|
| 1 | WRF-Chem | 2013 | India | ARI | Dudhia/RRTM | Thompson | Singh et al. (2020)* |

| | | | | | | | |
|---|---|---|---|---|---|---|---|
| 2 | WRF-Chem | 12/2015 | India | ARI | Goddard/RRTM | Lin | Bharali et al. (2019) |
| 3 | WRF-Chem | 10/13/2016 to 11/20/2016 | India | ARI | RRTMG | † | Shahid et al. (2019) |
| 4 | WRF-Chem | 12/27/2017 to 12/30/2017 | NCP | ARI | RRTMG | Lin | Wang et al. (2019a) |
| 5 | WRF-Chem | 12/05/2015 to 01/04/2016 | NCP | ARI | Goddard | WSM 6-class graupel | Wu et al. (2019a) |
| 6 | WRF-Chem | 12/05/2015 to 01/04/2016 | NCP | ARI | Goddard | WSM 6-class graupel | Wu et al. (2019b) |
| 7 | WRF-Chem | 06/01/2006 to 12/31/2011 | NWC | ARI | RRTMG | Morrison | Yuan et al. (2019) |
| 8 | WRF-Chem | 07/2016, 10/2016, 01/2017, 04/2017 | NCP | ARI | Goddard/RRTM | Lin | Zhang et al. (2019) |
| 9 | WRF-Chem | 02/17/2014 to 02/26/2014, 10/21/2014 to 10/25/2014, 11/05/2014 to 11/11/2014, 12/18/2015 to 12/24/2015 | NCP | ARI | RRTMG | Morrison | Zhou et al. (2019) |
| 10 | WRF-Chem | 03/15/2012 to 03/25/2012 | WA | ARI | RRTMG | Morrison | Bran et al. (2018) |
| 11 | WRF-Chem | 2013 | China & India | ARI | RRTMG | Lin | Gao et al. (2018bc) |
| 12 | WRF-Chem | 05/01/2007 to 05/07/2007 | CA | ARI | RRTM | Lin | Li and Sokolik (2018) |
| 13 | WRF-Chem | 06/02/2012 to 06/15/2012 | YRD | ARI | RRTMG | Lin | Li et al. (2018b) |
| 14 | WRF-Chem | 12/15/2016 to 12/21/2016 | NCP | ARI | RRTMG | Morrison | Liu et al. (2018b) |
| 15 | WRF-Chem | 11/30/2016 to 12/04/2016 | NCP | ARI | RRTMG | Lin | Miao et al. (2018) |
| 16 | WRF-Chem | 2010 | India | ARI | RRTMG | Morrison | Soni et al. (2018) |
| 17 | WRF-Chem | 01/01/2013 to 01/31/2013 | NCP | ARI | Goddard/RRTM | Lin | Wang et al. (2018c) |
| 18 | WRF-Chem | 12/2013 | EC | ARI | RRTMG | Lin | Wang et al. (2018d) |
| 19 | WRF-Chem | 2013 | TP | ARI | RRTMG | Morrison | Yang et al. (2018) |
| 20 | WRF-Chem | 03/11/2015 to 03/26/2015 | EA | ARI | RRTMG | Lin | Zhou et al. (2018) |
| 21 | WRF-Chem | 01/2013 | EC | ARI | RRTMG | Lin | Gao et al. (2017b) |
| 22 | WRF-Chem | 10/15/2015 to 10/17/2015 | YRD | ARI | Goddard/RRTM | Lin | Li et al. (2017b) |
| 23 | WRF-Chem | 03/16/2014 to 03/18/2014 | YRD | ARI | RRTMG | Lin | Li et al. (2017c) |
| 24 | WRF-Chem | 02/21/2014 to 02/27/2014 | NCP | ARI | RRTMG | Lin | Qiu et al. (2017) |
| 25 | WRF-Chem | 07/21/2012 | NCP | ARI | RRTMG | Lin | Yang and Liu (2017a) |
| 26 | WRF-Chem | 07/21/2012 | NCP | ARI | RRTMG | Lin | Yang and Liu (2017b) |
| 27 | WRF-Chem | 05/30/2013 to 06/27/2013 | EC | ARI | RRTMG | Lin | Yao et al. (2017) |
| 28 | WRF-Chem | 11/15/2013 to 12/30/2013 | SEC | ARI | RRTMG | Lin | Zhan et al. (2017) |
| 29 | WRF-Chem | 03/2012 | India | ARI | RRTMG | Thompson | Feng et al. (2016) |
| 30 | WRF-Chem | 1960-2010 | NCP | ARI | Goddard/RRTM | Lin | Gao et al. (2016b) |
| 31 | WRF-Chem | 04/2011 | NCP | ARI | RRTMG | Single-Moment 5-class | Liu et al. (2016a) |
| 32 | WRF-Chem | 01/2008, 04/2008, 07/2008, 10/2008 | EA | ARI | Goddard/RRTM | Lin | Liu et al. (2016b) |
| 33 | WRF-Chem | 09/21/2011 to 09/23/2011 | NCP | ARI | RRTMG | Lin | Miao et al. (2016) |
| 34 | WRF-Chem | 03/2005 | EA | ARI | Goddard/RRTM | Morrison | Wang et al. (2016) |
| 35 | WRF-Chem | 06/23/2008 to 07/20/2008 | NWC | ARI | RRTMG | Morrison | Yang et al. (2016) |
| 36 | WRF-Chem | 01/2007, 04/2007, 07/2007, 10/2007 | EA | ARI | RRTM | Lin | Zhong et al. (2016) |
| 37 | WRF-Chem | 05/2011, 10/2011 | India | ARI | RRTMG | Thompson | Govardhan et al. (2015) |
| 38 | WRF-Chem | 2006 | China | ARI | RRTMG | Lin | Huang et al. (2015) |
| 39 | WRF-Chem | 2007 to 2011 | EA | ARI | Goddard/RRTM | Lin | Chen et al. (2014) |
| 40 | WRF-Chem | 11/2007 to 12/2008 | EA | ARI | RRTMG | Lin | Gao et al. (2014) |
| 41 | WRF-Chem | 10/2006 | SEA | ARI | RRTM | Lin | Ge et al. (2014) |
| 42 | WRF-Chem | 04/17/2010 to 04/22/2010 | India | ARI | RRTM | Thompson | Kumar et al. (2014) |
| 43 | WRF-Chem | 01/11/2013 to 01/14/2013 | NCP | ARI | Goddard/RRTM | Lin | Li and Liao (2014) |
| 44 | WRF-Chem | 03/15/2008 to 03/18/2008 | EA | ARI | RRTMG | Morrison | Lin et al. (2014a) |
| 45 | WRF-Chem | 07/21/2006 to 07/30/2006 | NWC | ARI | RRTMG | Morrison | Chen et al. (2013) |

| | | | | | | | |
|---|---|---|---|---|---|---|---|
| 46 | WRF-Chem | 05/12/2009 to 05/22/2009 | India | ARI | Goddard/RRTM | Milbrandt-Yau | Dipu et al. (2013) |
| 47 | WRF-Chem | 2008 | India | ARI | Goddard/RRTM | Thompson | Kumar et al. (2012a) |
| 48 | WRF-Chem | 2008 | India | ARI | Goddard/RRTM | Thompson | Kumar et al. (2012b) |
| 49 | WRF-Chem | 1999 | India | ARI | Goddard/* | Lin | Seethala et al. (2011) |
| 50 | WRF-Chem | 2006 | China | ARI | † | † | Zhuang et al. (2011) |
| 51 | WRF-Chem | 12/14/2013 to 12/16/2013 | PRD | ARI & ACI | RRTMG | Morrison | Liu et al. (2020)* |
| 52 | WRF-Chem | 11/30/2009 to 12/01//2009 | NCP | ARI & ACI | Goddard/RRTM | Morrison | Jia et al. (2019) |
| 53 | WRF-Chem | 11/25/2013 to 12/26/2013 | EC | ARI & ACI | RRTMG | Lin | Wang et al. (2019b) |
| 54 | WRF-Chem | 01/2014 | China | ARI & ACI | RRTMG | Morrison | Archer-Nicholls et al. (2019) |
| 55 | WRF-Chem | 12/01/2016 to 12/09/2016, 12/19/2016 to 12/24/2016 | YRD | ARI & ACI | RRTMG | Lin | Li et al. (2019b) |
| 56 | WRF-Chem | 05/06/2013 to 20/06/2013 & 24/08/2014 to 08/09/2014 | India | ARI & ACI | RRTM | Lin | Kedia et al. (2019a) |
| 57 | WRF-Chem | 06/2010 to 09/2010 | India | ARI & ACI | RRTM | Lin, Morrison, Thompson | Kedia et al. (2019b) |
| 58 | WRF-Chem | 04/2013 | PRD | ARI & ACI | RRTMG | Lin | Huang et al. (2019) |
| 59 | WRF-Chem | 11/30/2013 to 12/10/2013 | EC | ARI & ACI | RRTMG | Morrison | Ding et al. (2019) |
| 60 | WRF-Chem | 12/01/2015 | NCP | ARI & ACI | RRTMG | Lin | Chen et al. (2019a) |
| 61 | WRF-Chem | 04/12/2015 to 27/12/2015 | EA | ARI & ACI | Goddard | WSM 6-class graupel | An et al. (2019) |
| 62 | WRF-Chem | 06/2015 to 02/2016 | MRYR | ARI & ACI | Goddard/RRTM | WSM 6-class graupel | Liu et al. (2018a) |
| 63 | WRF-Chem | 06/2008, 06/2009, 06/2010, 06/2011, 06,2012 | PRD | ARI & ACI | RRTMG | Morrison | Liu et al. (2018c) |
| 64 | WRF-Chem | 01/2014, 04/2014, 07/2014, 10/2014 | China | ARI & ACI | RRTMG | Lin | Zhang et al. (2018) |
| 65 | WRF-Chem | 10/01/2015 to 10/26/2015 | YRD | ARI & ACI | RRTMG | Lin | Gao et al. (2018a) |
| 66 | WRF-Chem | 2001, 2006, 2011 | EA | ARI & ACI | RRTMG | Morrison | Zhang et al. (2017) |
| 67 | WRF-Chem | 06/01/2011 to 06/06/2011 | EC | ARI & ACI | Goddard/RRTM | Lin | Wu et al. (2017) |
| 68 | WRF-Chem | 11/27/2013 to 12/12/2013 | YRD | ARI & ACI | Goddard/RRTM | Single-Moment 5-class | Sun et al. (2017) |
| 69 | WRF-Chem | 2005 & 2009 | YRD | ARI & ACI | RRTMG | Morrison | Zhong et al. (2017) |
| 70 | WRF-Chem | 01/2013 | NCP | ARI & ACI | Goddard/RRTM | Lin | Gao et al. (2017a) |
| 71 | WRF-Chem | 11/05/2014 to 11/11/2014 | NCP | ARI & ACI | Goddard/RRTM | Lin | Gao et al. (2017c) |
| 72 | WRF-Chem | 01/2010, 07/2010 | China | ARI & ACI | † | † | Ma and Wen (2017) |
| 73 | WRF-Chem | 06/01/2008 to 07/05/2008 | India | ARI & ACI | † | † | Lau et al. (2017) |
| 74 | WRF-Chem | 01/2013 | NCP | ARI & ACI | Goddard/RRTM | Morrison | Kajino et al. (2017) |
| 75 | WRF-Chem | 03/01/2009 to 03/31/2009 | TP & India | ARI & ACI | RRTMG | Morrison | Yang et al. (2017) |
| 76 | WRF-Chem | 2001, 2006, 2011 | EA | ARI & ACI | RRTMG | Morrison | He et al. (2017) |
| 77 | WRF-Chem | 05/2008 to 08/2008 | YRD | ARI & ACI | † | † | Campbell et al. (2017) |
| 78 | WRF-Chem | 01/2006, 04/2006, 07/2006, 10/2006 | China | ARI & ACI | Goddard/RRTM | Lin | Ma et al. (2016) |
| 79 | WRF-Chem | 01/2005, 04/2005, 07/2005, 10/2005 | EC | ARI & ACI | Goddard/RRTM | Lin | Zhang et al. (2016a) |
| 80 | WRF-Chem | 01/2005, 04/2005, 07/2005, 10/2005 | EC | ARI & ACI | Goddard/RRTM | Lin | Zhang et al. (2016b) |
| 81 | WRF-Chem | 12/07/2013 to 12/09/2013 | EC | ARI & ACI | Goddard/RRTM | Morrison | Zhang et al. (2016c) |
| 82 | WRF-Chem | 06/2012 | EC | ARI & ACI | RRTMG | Lin | Huang et al. (2016) |
| 83 | WRF-Chem | 01/2010, 07/2010 | YRD | ARI & ACI | Goddard/RRTM | Lin | Xie et al. (2016) |
| 84 | WRF-Chem | 11/12/2012 to 11/16/2012, 11/02/2013 to 11/06/2013 | India | ARI & ACI | Goddard/RRTM | Lin | Srinivas et al. (2016) |
| 85 | WRF-Chem | 07/2010 | India | ARI & ACI | RRTMG | Lin | Kedia et al. (2016) |
| 86 | WRF-Chem | 05/20/2008 to 08/31/2015 | India | ARI & ACI | Goddard/RRTM | Lin | Jin et al. (2016a) |
| 87 | WRF-Chem | 05/20/2008 to 08/31/2015 | India | ARI & ACI | Goddard/RRTM | Lin | Jin et al. (2016b) |
| 88 | WRF-Chem | 01/2010 | NCP | ARI & ACI | Goddard/RRTM | Lin | Gao et al. (2016a) |
| 89 | WRF-Chem | 01/05/2008 to 01/09/2008 | NCP | ARI & ACI | RRTMG | Lin | Gao et al. (2016c) |

| | | | | | | |
|---|---|---|---|---|---|---|
| 90 | WRF-Chem | 12/2013 | EC | ARI & ACI | RRTMG | Lin | Ding et al. (2016) |
| 91 | WRF-Chem | 02/15/2013 to 02/17/2013 | NCP | ARI & ACI | Goddard/RRTM | † | Yang et al. (2015) |
| 92 | WRF-Chem | 01/2010, 04/2010, 07/2010, 10/2010 | NCP | ARI & ACI | Goddard/RRTM | Lin | Shen et al. (2015) |
| 93 | WRF-Chem | 2006 & 2011 | EA | ARI & ACI | RRTMG | Morrison | Zhang et al. (2015d) |
| 94 | WRF-Chem | 2006 & 2011 | EA | ARI & ACI | RRTMG | Morrison | Chen et al. (2015b) |
| 95 | WRF-Chem | 06/27/2008 to 06/28/2008 | NCP | ARI & ACI | RRTM | Lin | Zhong et al. (2015) |
| 96 | WRF-Chem | 05/20/2008 to 08/31/2015 | India | ARI & ACI | Goddard/RRTM | Lin | Jin et al. (2015) |
| 97 | WRF-Chem | 03/2005, 04/2005, 05/2005 | India | ARI & ACI | Goddard/RRTM | Thompson | Jena et al. (2015) |
| 98 | WRF-Chem | 01/02/2013 to 01/26/2013 | NCP | ARI & ACI | RRTMG | Morrison | Gao et al. (2015b) |
| 99 | WRF-Chem | 07/08/2013 to 07/09/2013 | SWC | ARI & ACI | RRTMG | † | Fan et al. (2015) |
| 100 | WRF-Chem | 01/2010, 04/2010, 07/2010, 10/2010 | NCP | ARI & ACI | Goddard/RRTM | Lin | Chen et al. (2015a) |
| 101 | WRF-Chem | 01/2013 | EC | ARI & ACI | Goddard/RRTM | Lin | Zhang et al. (2015a) |
| 102 | WRF-Chem | 2006 & 2007 | EA | ARI & ACI | Goddard/† | Lin | Wu et al. (2013) |
| 103 | WRF-Chem | 09/27/2010 to 10/22/2010 | India | ARI & ACI | Goddard/RRTM | Lin | Beig et al. (2013) |
| 104 | WRF-Chem | 12/1/2009 | NCP | ARI & ACI | Goddard/RRTM | Lin | Jia and Guo, (2012) |
| 105 | WRF-Chem | 01/2001, 07/2001 | EA | ARI & ACI | Goddard/RRTM | Lin | Zhang et al. (2012) |
| 106 | WRF-Chem | 11/10/2007 to 01/01/2008 | China | ARI & ACI | RRTMG | Lin | Gao et al. (2012) |
| 107 | WRF-Chem | 06/18/2018 to 06/19/2018 | MRYR | ACI | Goddard/RRTM | † | Bai et al. (2020)* |
| 108 | WRF-Chem | 06/07/2017 to 06/12/2017 | YRD | ACI | RRTMG | Morrison | Liu et al. (2019) |
| 109 | WRF-Chem | 03/2010 to 05/2010 | EA | ACI | RRTMG | Morrison | Wang et al. (2018b) |
| 110 | WRF-Chem | 03/09/2012 to 04/30/2012 | EA | ACI | RRTMG | Thompson | Su and Fung (2018a) |
| 111 | WRF-Chem | 03/09/2012 to 04/30/2012 | EA | ACI | RRTMG | Thompson | Su and Fung (2018b) |
| 112 | WRF-Chem | 05/18/2015 to 06/13/2015 | NEA | ACI | RRTMG | Morrison | Park et al. (2018) |
| 113 | WRF-Chem | 08/2008 | EC | ACI | RRTMG | Lin | Gao and Zhang (2018) |
| 114 | WRF-Chem | 10/03/2013 to 10/07/2013 | SEC | ACI | RRTMG | Morrison | Shen et al. (2017) |
| 115 | WRF-Chem | 01/2013, 07/2013 | China | ACI | Fu-Liou-Gu | Morrison | Zhao et al. (2017) |
| 116 | WRF-Chem | 06/04/2004 to 07/10/2004 | India | ACI | Goddard | Lin | Bhattacharya et al. (2017) |
| 117 | WRF-Chem | 09/20/2013 to 09/23/2013 | PRD | ACI | RRTMG | Lin | Jiang et al. (2016) |
| 118 | WRF-Chem | 2005 & 2010 | EA | ACI | RRTMG | Morrison | Zhang et al. (2015c) |
| 119 | WRF-Chem | 08/20/2009 to 08/29/2008 | India | ACI | Goddard/RRTM | Morrison | Sarangi et al. (2015) |
| 120 | WRF-Chem | 01/2001, 04/2001, 07/2001, 10/2001, 01/2005, 04/2005, 07/2005, 10/2005, 01/2008, 04/2008, 07/2008, 10/2008 | EA | ACI | † | † | Zhang et al. (2014b) |
| 121 | WRF-Chem | 07/2008 | EC | ACI | RRTMG | Morrison | Lin et al. (2014a) |
| 122 | WRF-Chem | 1980 to 2010 | SEC | ACI | † | † | Bennartz et al. (2011) |
| 123 | WRF-Chem | 2008 & 2050 | China | ARI (Health) | † | † | Zhong et al. (2019) |
| 124 | WRF-Chem | 2014 | India | ARI (Health) | RRTM | Thompson | Conibear et al. (2018a) |
| 125 | WRF-Chem | 2015 & 2050 | India | ARI (Health) | RRTM | Thompson | Conibear et al. (2018b) |
| 126 | WRF-Chem | 2011 | India | ARI (Health) | Goddard/RRTM | Thompson | Ghude et al. (2016) |
| 127 | WRF-Chem | 2013 | NCP | ARI (Health) | RRTMG | † | Gao et al. (2015a) |
| 128 | WRF-CMAQ | 03/2006 & 04/2006 to 03/2010 & 04/2010 | EA | ARI | † | † | Dong et al. (2019) |
| 129 | WRF-CMAQ | 04/10/2016 to 06/19/2016 | NEA | ARI | RRTMG | Single-Moment 3-class | Jung et al. (2019) |
| 130 | WRF-CMAQ | 2014 | EA | ARI | RRTMG | Morrison | Nguyen et al. (2019a) |
| 131 | WRF-CMAQ | 2014 | SEA | ARI | RRTMG | Morrison | Nguyen et al. (2019b) |
| 132 | WRF-CMAQ | 02/2015 | NEA | ARI | RRTMG | Single-Moment 5-class | Yoo et al. (2019) |
| 133 | WRF-CMAQ | 01/2014, 02/2014, 03/2014 | EA | ARI | RRTMG | Morrison | Sekiguchi et al. (2018) |

| | | | | | | | |
|---|---|---|---|---|---|---|---|
| 134 | WRF-CMAQ | 2006 to 2010, 2013 | EA | ARI | RRTMG | Morrison | Hong et al. (2017) |
| 135 | WRF-CMAQ | 01/2013, 07/2013 | China | ARI | RRTMG | Morrison | Xing et al. (2017) |
| 136 | WRF-CMAQ | 1990 to 2010 | EA | ARI | RRTMG | Morrison | Xing et al. (2016) |
| 137 | WRF-CMAQ | 1990 to 2010 | EC | ARI | RRTMG | Morrison | Xing et al. (2015a) |
| 138 | WRF-CMAQ | 1990 to 2010 | EC | ARI | RRTMG | Morrison | Xing et al. (2015b) |
| 139 | WRF-CMAQ | 1990 to 2010 | EC | ARI | RRTMG | Morrison | Xing et al. (2015c) |
| 140 | WRF-CMAQ | 01/2013 | China | ARI | RRTMG | Morrison | Wang et al. (2014a) |
| 141 | WRF-CMAQ | 01/2013, 04/2013, 07/2013, 10/2013 | China | ACI | RRTMG | Morrison | Chang (2018) |
| 142 | WRF-CMAQ | 2050 | China | ARI (Health) | RRTMG | Morrison | Hong et al. (2019) |
| 143 | WRF-CMAQ | 1990 to 2010 | EA & India | ARI (Health) | RRTMG | Morrison | Wang et al. (2017) |
| 144 | GRAPES-CUACE | 12/15/2016 to 12/24/2016 | NCP | ARI | Goddard | † | Wang et al. (2018a) |
| 145 | GRAPES-CUACE | 07/07/2008 to 07/11//2008 | EC | ARI | CLIRAD | † | Wang et al. (2015a) |
| 146 | GRAPES-CUACE | 04/26/2006 | EA | ARI | Goddard/† | † | Wang and Niu .(2013) |
| 147 | GRAPES-CUACE | 04/26/2006 | EA | ARI | Goddard/† | † | Wang et al. (2013) |
| 148 | GRAPES-CUACE | 07/13/2008 to 07/31/2008 | NCP | ARI | † | † | Zhou et al. (2012) |
| 149 | GRAPES-CUACE | 04/26/2006 | EA | ARI | Goddard/† | † | Wang et al. (2010) |
| 150 | GRAPES-CUACE | 01/2013 | EC | ACI | † | Single-Moment 6-class | Zhou et al. (2016) |
| 151 | WRF-NAQPMS | 2013 | EA | ARI | † | † | Li et al. (2018a) |
| 152 | WRF-NAQPMS | 09/27/2013 to 10/01/2013 | NCP | ARI | Goddard/RRTM | Lin | Wang et al. (2014b) |
| 153 | WRF-NAQPMS | 01/01/2013 | EC | ARI | Goddard/RRTM | Lin | Wang et al. (2014c) |
| 154 | GATOR-GCMOM | 2000 & 2009 | NEA | ARI & ACI | † | † | Ten Hoeve and Jacobson, 2012 |
| 155 | GATOR-GCMOM | 2002 & 2009 | India | ARI & ACI | † | † | Jacobson et al. (2019) |
| 156 | GATOR-GCMOM | 2000 & 2009 | NCP | ARI & ACI | † | † | Jacobson et al. (2015) |
| 157 | Multi-model comparison | † | EA | ARI & ACI | † | † | Chen et al. (2019b) |
| 158 | Multi-model comparison | 2010 | EA | ARI & ACI | † | † | Li et al., (2019a) |
| 159 | Multi-model comparison | 01/2010 | NCP | ARI & ACI | † | † | Gao et al. (2018b) |
| 160 | Multi-model comparison | 05/2011 | India | ARI & ACI | † | † | Govardhan et al. (2016) |

†: Unclear; *: A preprint version of this study was available online on October 31, 2019, and was formally published on January 1, 2020. (EA: East Asia, NEA: Northeast Asia, SEA: Southeast Asia, EC: East China, NCP: North China Plain, YRD: Yangtze River Delta, SEC: Southeast China, NWC: Northwest China, TP: Tibetan Plateau, MRYR: middle reaches of the Yangtze River, SWC: Southwest China; PRD: Pearl River Delta).

**3.2 Spatiotemporal distribution of publications**

To gain an overall understanding of applications of coupled models in Asia, the spatial distributions of study areas of the selected literatures and the temporal variations of the annual publication numbers were extracted from Table 1 and summarized. Figure 1 illustrates the spatial distributions of study regions as well as the number of papers involving coupled models in Asia (Fig. 1a) and China (Fig. 1b). In this figure, the color and number in the pie charts represent individual (WRF-Chem, WRF-CMAQ, GRAPES-CUACE, WRF-NAQPMS, and GATOR-GCMOM) or multiple coupled models and the quantity of corresponding articles, respectively. At subregional scales, most studies targeted EA where high anthropogenic aerosol loading occurred in recent decades, mainly using WRF-Chem and WRF-CMAQ (Fig. 1a). For other subregions, such as NEA, SEA, Central Asia (CA), and West Asia (WA), there were rather limited research activities taking into account aerosol feedbacks with two-way coupled models. National scale applications of two-way coupled models targeted mostly modeling domains covering India and China but much less work were carried out in other countries, such as Japan and Korea, where air pollution levels are much lower. With respect to various areas in China (Fig. 1b), the research activities concentrated mostly in NCP and secondly in the East China (EC), then in the Yangtze River Delta (YRD) and Pearl River Delta (PRD) areas. WRF-Chem was the most popular model applied in all areas, but

there were a few applications of GPRAPES-CUACE and WRF-NAQPMS in EC and NCP.

Figure 2 depicts the temporal variations of research activities with two-way coupled models in Asia over the period of 2010 to 2019. The total number of papers related to two-way coupled models had an obvious upward trend in the past decade. Prior to 2014, applications of two-way coupled models in Asia were scarce, with about 1 to 6 publications per year. A noticeable increase of research activities emerged starting from 2014 and the growth was rapid from 2014 to 2016, at a rate of 7-9 more papers per year, especially in China. It could be related to the Action Plan on Prevention and Control of Atmospheric Pollution (2013-2017) implemented by the Chinese government. The growth was rather flat during 2016-2018 before reaching a peak of 31 articles in 2019. In addition, the pie charts in Fig. 2 indicates that modeling activities had been picking up with a diversified pattern in study domain from 2010 to 2019. The modeling domains extended from EA to China and India and then several subregions in Asia and various areas in China. For EA and India, investigations of aerosol feedbacks based on two-way coupled models rose from 1-2 papers per year during 2010-2013 to 4-8 during 2014-2019. Since 2014, most model simulations were carried out towards areas with severe air pollution in China, especially the NCP area where attracted 5-7 publications per year.

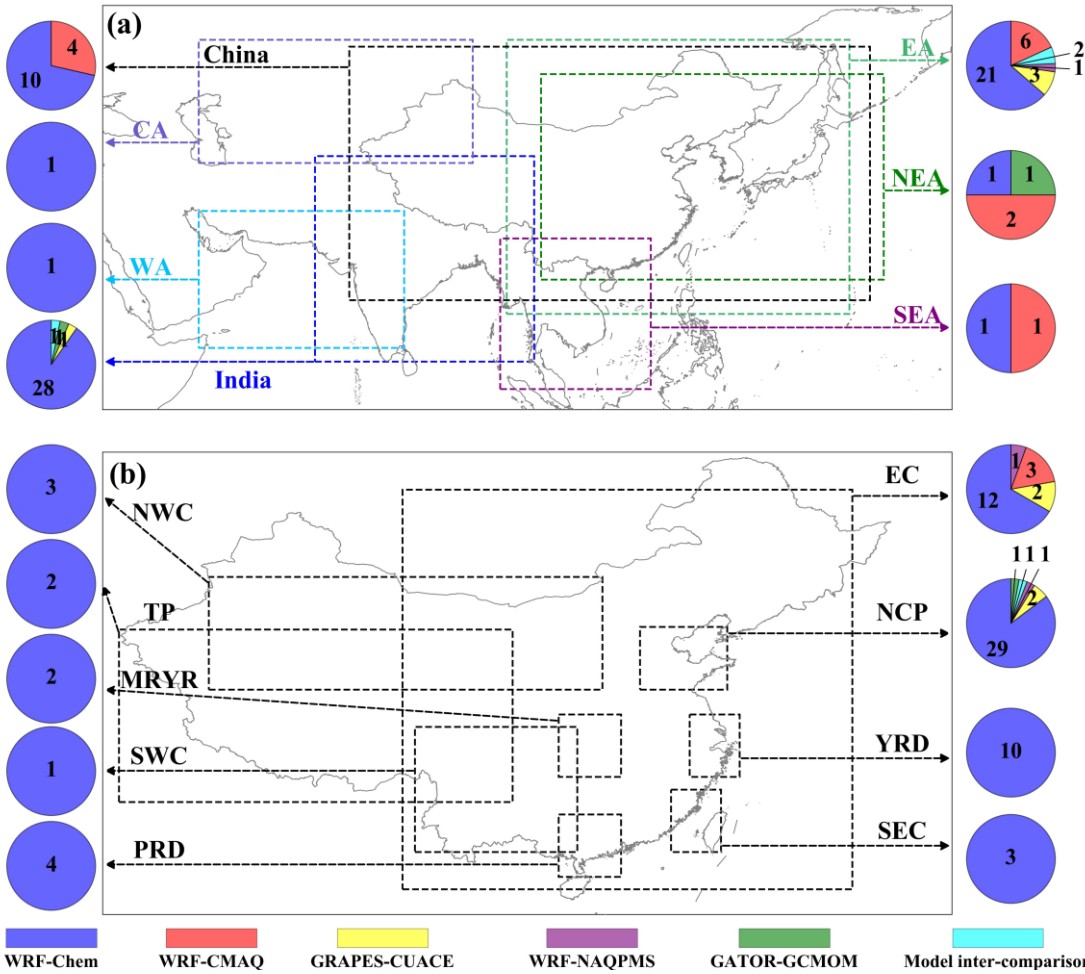

Figure 1. The spatial distributions of study domains as well as the two-way coupled modeling publication numbers in different subregions or countries of Asia (a) and areas of China (b). (EA: East Asia, NEA: Northeast Asia, SEA: Southeast Asia, EC: East China, NCP: North China Plain, YRD: Yangtze River Delta, SEC: Southeast China, NWC: Northwest China, TP: Tibetan Plateau, MRYR: middle reaches of the Yangtze River, SWC: Southwest China; PRD: Pearl River Delta).

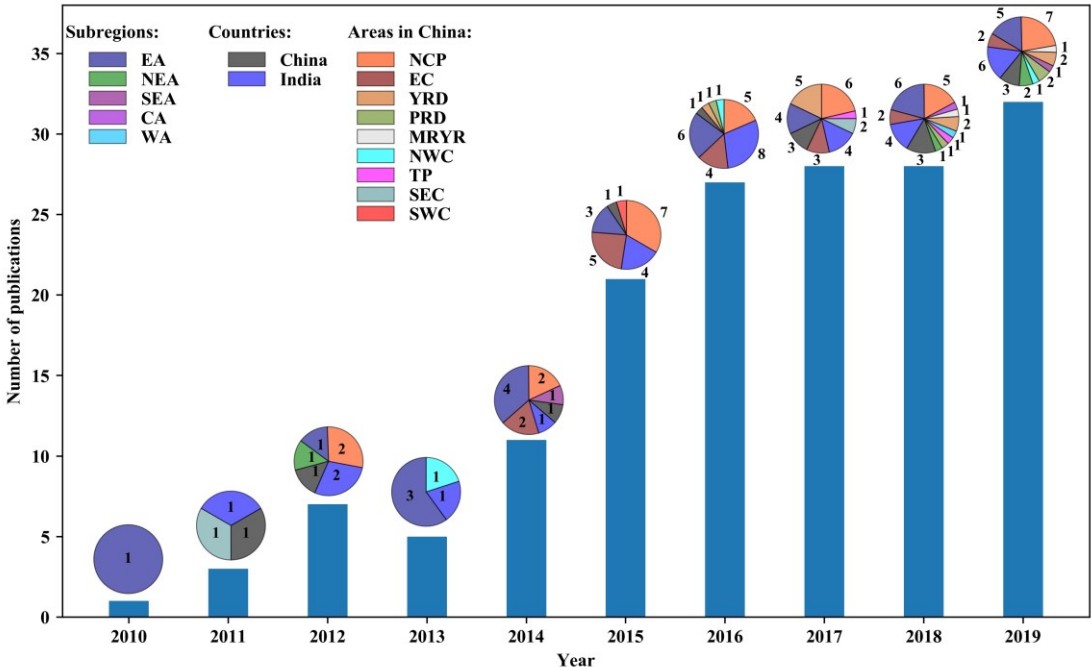

Figure 2. The temporal variations of study activities adopting two-way coupled models in Asia during 2010-2019. (EA: East Asia, NEA: Northeast Asia, SEA: Southeast Asia, EC: East China, NCP: North China Plain, YRD: Yangtze River Delta, SEC: Southeast China, NWC: Northwest China, TP: Tibetan Plateau, MRYR: middle reaches of the Yangtze River, SWC: Southwest China; PRD: Pearl River Delta).

## 3.3 Summary of modeling methodologies

The physiochemical processes involved with ARI and ACI are sophisticated in actual conditions of atmospheric environment but their representations in two-way coupled models can be rather different. Also, simulation results depend on how these models are configured and set up. Therefore, the treatments of aerosol and cloud microphysics, and aerosol-radiation-cloud interactions in WRF-Chem, WRF-CMAQ, GRAPES-CUACE, WRF-NAQPMS and GATOR-GCMOM applied in Asia, as well as the various aspects of how the modeling studies being set up in the selected papers are summarized in Tables 2-5, respectively, and outlined in this section.

Aerosol microphysics processes consist of particle nucleation, coagulation, condensation/evaporation, gas/particle mass transfer, inorganic aerosol thermodynamic equilibrium, aqueous chemistry and formation of secondary organic aerosol (SOA). Their representations in a variety of aerosol mechanisms offered in the five two-way coupled models applied in Asia and relevant references are compiled in Table 2. Note that the GOCART scheme in WRF-Chem is based on a bulk aerosol mechanism that is not able to consider the details of these microphysics processes. The binary homogeneous nucleation schemes with/out hydration developed by different authors are applied in the five coupled models for simulating the new particle formation and GATOR-GCMOM also adopts the ternary nucleation parameterization scheme for $H_2SO_4$, $NH_3$ and $H_2O$ vapors. All the five coupled models calculate the aerosol-aerosol coagulation rate coefficients based the Brownian coagulation theory, with certain enhancements in GATOR-GCMOM as stated in details by Jacobson (1999). The dynamic condensation/evaporation approaches of inorganic gases (e.g., $H_2SO_4$, $NH_3$, $HNO_3$, and HCl) and organic gases (VOCs) based on the Fuchs-Sutugin expression are implemented in various aerosol mechanisms offered by WRF-Chem, WRF-CMAQ, GRAPES-CUACE, and WRF-NAQPMS, while GATOR-GCMOM deploys the condensation/evaporation approach in which several terms of processes are factored in the 3-D equations of discrete size-resolved aerosol growth (Jacobson, 2012a). The mass transfer between gaseous and aerosol particles are treated via two typical methods (i.e., bulk equilibrium and kinetic) in most coupled models, and the hybrid and Henry's law equilibrium methods are also applied in the MADRID (WRF-Chem) and the 6th/7th generation CMAQ aerosol modules (AERO6/AERO7) (WRF-CMAQ), respectively. Different versions of the ISORROPIA module, the Model for an Aerosol Reacting System-version A (MARS-A), the Multicomponent Equilibrium Solver for Aerosols with the Multicomponent Taylor Expansion Method (MESA-MTEM), and the EQUIlibrium SOLVer version 2 (EQUISOLV

II) modules are implemented for computing the inorganic aerosol thermodynamic equilibrium in
these two-way coupled models. For aqueous chemistry, the bulk aqueous chemistry scheme and
variations of the CMAQ's standard aqueous chemistry module (AQCHEM) are the most applied,
and the CBM-IV aqueous chemistry scheme, the Regional Acid Deposition Model (RADM)
aqueous chemistry module, and the size-resolved aqueous chemistry module are utilized as well.
Multiple approaches have been incorporated into the five coupled models for calculating the SOA
formation and include the volatility basis set (VBS) approach, approaches considering reversible
absorption or combined absorption and dissolution, fixed or bulk two-product yield approaches, and
the approach of time-dependent organics condensation/evaporation with considering vapor pressure.
Table 2. Treatments of aerosol microphysics processes in two-way coupled models (WRF-Chem, WRF-CMAQ,
GRAPES-CUACE, WRF-NAQPMS and GATOR-GCMOM) applied in Asia.

| | WRF-Chem | | | | | | WRF-CMAQ | | | GRAPES-CUACE | WRF-NAQPMS | GATOR-GCMOM |
|---|---|---|---|---|---|---|---|---|---|---|---|---|
| | GOCART | MADE/SORGAM | AERO5 | MAM3/MAM7 | MOSAIC | MADRID | AERO5 | AERO6 | AERO7 | CUACE[※] | AERO5 | GATOR2012* |
| New particle formation/if with hydration | None | $H_2SO_4$-$H_2O$ binary homogeneous nucleation (Kulmala et al., 1998)/Yes | $H_2SO_4$-$H_2O$ binary homogeneous nucleation (Kulmala et al., 1998)/Yes | $H_2SO_4$-$H_2O$ binary homogeneous nucleation (Vehkamäki et al., 2002)/Yes | $H_2SO_4$-$H_2O$ binary homogeneous nucleation (Wexler, et al., 1994)/Yes | $H_2SO_4$-$H_2O$ binary homogeneous nucleation (McMurry and Friedlander, 1979)/Unclear | $H_2SO_4$-$H_2O$ binary homogeneous nucleation (Kulmala et al., 1998)/Yes | $H_2SO_4$-$H_2O$ binary homogeneous nucleation (Vehkamäki et al., 2002)/Yes | $H_2SO_4$-$H_2O$ binary homogeneous nucleation (Vehkamäki et al., 2002)/Yes | $H_2SO_4$-$H_2O$ binary homogeneous nucleation (Kulmala et al., 1998 )/Yes | $H_2SO_4$-$H_2O$ binary homogeneous nucleation (Yu, 2006)/Yes | $H_2SO_4$-$H_2O$ binary homogeneous nucleation (Vehkamäki et al., 2002)/Yes; $H_2SO_4$-$NH_3$-$H_2O$ ternary homogeneous nucleation (Napari et al., 2002)/Yes |
| Coagulation | None | Brownian motion (Binkowski and Shankar, 1995) | Brownian motion (Binkowski and Roselle, 2003) | Brownian motion (Whitby, 1978) | Brownian motion (Jacobson et al., 1994) | Brownian motion (Jacobson et al., 1994) | Brownian motion (Binkowski and Roselle, 2003) | Brownian motion (Binkowski and Roselle, 2003) | Brownian motion (Binkowski and Roselle, 2003) | Brownian motion (Jacobson et al., 1994) | Brownian motion (Jacobson et al., 1994; Chen et al., 2017d) | Brownian motion, Brownian diffusion enhancement, turbulent shear, turbulent inertial motion, gravitational setting, Van der Waals forces, viscous forces, fractal geometry (Jacobson, 2003) |
| Condensation/ Evaporation | None | Dynamical condensation/ evaporation of $H_2SO_4$ vapor and VOCs based on Fuchs-Sutugin expression (Binkowski and Shankar, 1995) | Dynamical condensation/ evaporation of $H_2SO_4$ vapor and VOCs based on Fuchs-Sutugin expression (Binkowski and Shankar, 1995); Condensation/ evaporation of volatile inorganic gases to/from the gas-phase concentrations of coarse particle surfaces using ISORROPIA in reverse mode (CMAQ User's Guide) | Dynamical condensation of $H_2SO_4$ vapor, $NH_3$ (7 modes) and semi-volatile organics; Condensation/ evaporation of SOA gas (Liu et al., 2012) | Dynamical condensation/ evaporation of $H_2SO_4$ vapor, methanesulfonic acid, $HNO_3$, HCl and $NH_3$ with adaptive step time-split Euler approach (Zaveri et al., 2008) | Dynamical condensation/ evaporation of semi-volatile species for analytical predictor of condensation with moving-center approach (Zhang et al., 2010) | Dynamical condensation/ evaporation of $H_2SO_4$ vapor and VOCs based on Fuchs-Sutugin expression (Binkowski and Shankar, 1995); Condensation/ evaporation of volatile inorganic gases to/from the gas-phase concentrations of coarse particle surfaces using ISORROPIA in reverse mode (CMAQ User's Guide) | Same as in AERO5 | Same as in AERO5 | Dynamical condensation/ evaporation of $H_2SO_4$ vapor and gaseous precursors based on modified Fuchs-Sutugin expression (Jacobson, et al., 1994; Gong et al., 2003a) | Condensation/ evaporation of $H_2SO_4$ with advanced particle microphysics approach (Li et al., 2018a; Yu and Luo, 2009; Chen et al., 2019c; Yu, 2006) | Dynamical condensation of $H_2O$ and involatile species with Analytical Predictor of Nucleation, Condensation, and Dissolution scheme (Jacobson, 2002); Evaporation of a volatile component over a single particle (Jacobson and Turco, 1995) |
| Gas/particle mass transfer | None | 1. Bulk equilibrium approach for $HNO_3$ and $NH_3$ (Zhang et al., 2005) 2. Kinetic approach for $H_2SO_4$ (Zhang et al., 2016d) | Kinetic approach for all species (Foley et al., 2010) | Bulk equilibrium approach for $(NH_4)_2SO_4$ (He and Zhang, 2014) | Kinetic approach for all species (Zaveri et al., 2008) | 1. Bulk equilibrium approach for $HNO_3$ and $NH_3$ (Zhang et al., 2010) 2. Kinetic approach for all species (Zhang et al., 2010) 3. Hybrid approach (Zhang et al., 2010) | Kinetic approach for all species (Foley et al., 2010) | 1. Henry's law equilibrium (Foley et al., 2017) 2. Kinetic approach for all species (Foley et al., 2017) | Same as in AERO6 | Kinetic approach for all species (Zhou et al., 2021) | Kinetic for all species (Chen et al., 2021) | Kinetic approach for all species (Jacobson, 1999) |
| Inorganic aerosol thermodynamic equilibrium | None | MARS-A (Binkowski and Shankar, 1995) | ISORROPIA (Byun and Kenneth, 2006) | ISORROPIA II (He and Zhang, 2014) | MESA-MTEM (Zaveri et al., 2008) | ISORROPIA (Zhang et al., 2010) | ISORROPIA (Byun and Kenneth, 2006) | ISORROPIA II (Appel et al., 2013) | ISORROPIA II (Appel et al., 2013) | ISSOROPIA (Zhou et al., 2012) | ISSOROPIA (Li et al., 2011) | EQUISOLV II (Jacobson, 1999) |
| Aqueous chemistry | None | Bulk cloud-chemistry scheme (Fahey and Pandis, 2001; Zhang et al., 2015b) | AQCHEM (Fahey et al., 2017) | Based on algorithm developed by Barth et al. (2000) (He and Zhang, 2014) | Same as in MADE/ SORGAM (Fahey and Pandis, 2001; Chapman et al., 2009) | Same as in MADE/ SORGAM (Fahey and Pandis, 2001; Zhang et al., 2004) | 1. AQCHEM 2. AQCHEM-KMT (Fahey et al., 2017) | 1. AQCHEM-KMT 2. AQCHEM-KMTI (Fahey et al., 2017) | 1. AQCHEM-KMT 2. AQCHEM-KMTI (Fahey et al., 2017) | Based on aqueous chemistry in CBM-IV mechanism by Gery et al. (1989) | Based on the RADM mechanism used in CMAQ v4.6 (AERO5) (Li et al., 2011a) | Bulk or size-resolved cloud-chemistry module (GATOR2012) |
| SOA formation | None | 1. Reversible absorption of 8 classes volatile organic compounds (VOCs) based on Caltech smog-chamber data (Odum et | Combined absorption and dissolution approaches for 9 parent VOCs and 32 SOA species (Carlton, et al., 2010) | Treatment of SOA from fixed mass yields for anthropogenic and biogenic precursor VOCs (Liu et al., 2012) | 1. Based on ambient ageing measurement of organic aerosols by Hodzic and Jimenez (2011) 2. Based on volatility basis set approach | 1. Absorptive approach for 14 parent VOCs and 38 SOA species 2. Combined absorption and | Combined absorption and dissolution approaches for 9 parent VOCs and 32 SOA species (Carlton, et al., 2010) | On the basis of SOA scheme in AERO5, adding parameterization of in-cloud SOA formation from biogenic VOCs (Foley et al., 2017) | On the basis of SOA scheme in AERO5/6, updated parametrization of monoterpene SOA yielded from photooxidation | Reversible absorption of 8 classes VOCs based on Caltech smog-chamber data (Zhou et al., 2012) | Bulk two-product yield parametrization (Fu et al., 2016; Odum et al., 1997) | Using Henry's Law to determine vapor pressure of organics and perform either time-dependent condensation or evaporation calculations. |

| | | | | |
|---|---|---|---|---|
| al., 1997; Griffin et al., 1999) 2. Based on volatility basis set approach (Ahmadov et al., 2012) | (Knote et al., 2014) | dissolution approaches for 42 hydrophilic and hydrophobic VOCs (Zhang et al., 2004) | (Foley et al., 2021) | (Jacobson, 2002) |

※CUACE is the aerosol mechanism implemented in the GRAPES-CUACE model (Zhou et al., 2012).
*GATOR2012 is the aerosol mechanism implemented in the GATOR-GCMOM model (Jacobson et al., 2012b).

In addition to aerosol microphysics processes, the cloud properties included in cloud
microphysics schemes and the treatment of aerosol-cloud processes in the five two-way coupled
models are different in terms of hydrometeor classes, cloud droplet size distribution, aerosol water
uptake, in-/below-cloud scavenging, hydrometeor-aerosol coagulations, and sedimentation of
aerosols and cloud droplets (Table 3). Among the microphysics schemes implemented in the five
coupled models, mass concentrations of different hydrometeors (including cloud water, rain, ice,
snow or graupel) are included but their number concentrations are only considered if the cloud
microphysics schemes are two-moment or three-moment. The single modal approach with either
lognormal or gamma distribution and the sectional approach with discrete size distributions for
cloud droplets are applied in different microphysics schemes. Based on the Mie theory, WRF-Chem,
WRF-CMAQ, GRAPES-CUACE, WRF-NAQPMS and GATOR-GCMOM calculate cloud
radiative properties (including extinction/scattering/absorption coefficient, single scattering albedo
and asymmetry factor of liquid and ice clouds) in their radiation schemes (e.g., RRTMG,
GODDARD, GATOR2012). In atmosphere, the hygroscopic growth of aerosols due to water uptake
is parameterized based on the Köhler or Zdanovskii-Stokes-Robinson theory and the hysteresis
effects depending on the deliquescence and crystallization RH are taken into account in the five
coupled models. The removal processes of aerosol particles include wet removal and sedimentation.
Aerosol particles in accumulation and coarse modes can act as CCN or IN via activations in cloud,
which can further develop to different types of hydrometeors (cloud water, rain, ice, snow and
graupel), and then gradually form precipitations. These processes are named as in-cloud scavenging
or rainout. The aerosol particles below cloud base also can be coagulated with the falling
hydrometeors, which are known as below-cloud scavenging or wash out. Both representations of
in- and below-cloud scavenging processes are based on scavenging rate approach in aerosol
mechanisms of WRF-Chem, WRF-CMAQ, GRAPES-CUACE and WRF-NAQPMS except
GATOR-GCMOM. Size-resolved sedimentation of aerosols are computed from one model layer to
layers below down to the surface layer using setting velocity in most coupled models and the
MOSAIC aerosol mechanism in WRF-Chem only considers the sedimentation in the lowest model
level (Marelle et al., 2017).
Table 3. Compilation of cloud properties and aerosol-cloud processes in two-way coupled models (WRF-Chem,
WRF-CMAQ, GRAPES-CUACE, WRF-NAQPMS and GATOR-GCMOM) applied in Asia.

| | WRF-Chem | WRF-CMAQ | GRAPES-CUACE | WRF-NAQPMS | GATOR-GCMOM |
|---|---|---|---|---|---|
| Hydrometeor (Cloud microphysics scheme) | Mass concentrations: Cloud water, rain, ice, snow and graupel (Morrison, Lin, Thompson, WSM 6 class and Milbrandt-Yau) Cloud water, rain, ice and snow (WSM 5 class) Number concentrations: Rain, ice, snow and graupel (Morrison and Milbrandt-Yau) Rain and ice (Thompson) None (Lin, WSM 5 class and WSM 6 class) | Mass concentrations: Cloud water, rain, ice, snow and graupel (Morrison) Cloud water, rain, ice and snow (WSM 5 class) Cloud water and rain (WSM 3 class) Number concentrations: Rain, ice, snow and graupel (Morrison) None (WSM 3 class and WSM 5 class) | Mass concentrations: Cloud water, rain, ice, snow and graupel (WSM 6 class) Number concentrations: None (WSM 6 class) | Mass concentrations Cloud water, rain, ice, snow and graupel (Lin) Number concentrations: None (Lin) | Mass concentrations: Cloud water, ice and graupel (GATOR2012) Number concentrations: Cloud water, ice and graupel (GATOR2012) |
| Cloud droplet size distribution (Cloud microphysics scheme) | 1. Single, modal approach with lognormal distribution (Morrison and Lin) 2. Gamma distribution (Thompson, WSM 5 class and WSM 6 class) | 1. Single, modal approach with lognormal distribution (Morrison) 2. Gamma distribution (WSM 3 class and WSM 5 class) | Gamma distribution (WSM 6 class) | Single, modal approach with lognormal distribution (Lin) | Sectional approach with multiple size distributions (GATOR2012*) (Jacobson, et al., 2007) |
| Cloud radiative properties (Radiation scheme) | Extinction coefficient, single scattering albedo and asymmetry factor of liquid and ice clouds based on Mie scattering theory (RRTMG SW) Absorption coefficient of liquid and ice clouds using constant values (RRTMG LW) Extinction coefficient, single scattering albedo and asymmetry factor of liquid and ice clouds from lookup tables (Goddard SW and LW) | Extinction coefficient, single scattering albedo and asymmetry factor of liquid and ice clouds based on Mie scattering theory (RRTMG SW) Absorption coefficient of liquid and ice clouds using constant values (RRTMG LW) | Extinction coefficient, single scattering albedo and asymmetry factor of liquid and ice clouds using lookup tables (Goddard SW) Extinction coefficient, single scattering albedo and asymmetry factor of liquid and ice clouds from lookup tables (Goddard LW) | Extinction coefficient, single scattering albedo and asymmetry factor of liquid and ice clouds using lookup tables (Goddard SW) Clear sky optical depth from lookup table (RRTM LW) | Integrating spectral optical properties over each size bin of each hydrometeor particle size distribution (Toon SW and LW) (Jacobson and Jadhav, 2018) |
| Aerosol water uptake | Equilibrium with RH based on Köhler theory, and hysteresis is treated (Ghan and Zaveri, 2007) | The empirical equations of deliquescence and crystallization RH developed by Martin et al (2003), and hysteresis is treated (CMAQ source code) | Equilibrium with the mutual deliquescence and crystallization RH using the Zdanovskii-Stokes-Robinson equation, and hysteresis is treated (Personal communication) | Equilibrium with the mutual deliquescence and crystallization RH using the Zdanovskii-Stokes-Robinson equation, and hysteresis is treated (Nenes et al., 1998; Li et al., 2011) | Size-resolved equilibrium with the mutual deliquescence and crystallization RH using the Zdanovskii-Stokes-Robinson equation, and hysteresis is treated (Jacobson et al., 1996b) |
| In-cloud scavenging (Aerosol mechanism) | Scavenging via nucleation, Brownian diffusion, collection and autoconversion in both grid-scale and sub-grid clouds with a first-order removal rate (MADE/SORGAM, MOSAIC, MAM3 and MAM7) (Easter et al., 2004) | Scavenging of interstitial aerosol in the Aitken mode and nucleation scavenging of aerosol in the accumulation and coarse modes by the cloud droplets in both grid-scale and sub-grid clouds (AERO5, AERO6 and AERO7) (Binkowski and Roselle, 2004; Fahey et al., 2017) | Algorithm of rainout removal tendency by Giorgi and Chameides (1986) | Employing a scavenging coefficient approach based on relationships described by Seinfeld and Pandis (1998), only hydrophilic particles can be scavenged (Chen et al., 2017d) | Size-resolved aerosol activation; nucleation scavenging and autoconversion for size-resolved cloud droplets (GATOR2012) (Jacobson, 2003) |

| | | | | | |
|---|---|---|---|---|---|
| Below-cloud scavenging (Aerosol mechanism) | Scavenged aerosols are instantly removed by interception and impaction but not resuspended by evaporating rain (MADE/SORGAM, MOSAIC, MAM3 and MAM7) (Slinn, 1984; Easter et al., 2004) | All aqueous species are scavenged from the cloud top to the ground in both grid-scale and sub-grid clouds (AERO5, AERO6 and AERO7) (CMAQ User's Guide; Fahey et al., 2017) | Aerosol particles between sizes ranging from 0.5 to 1 μm radius are instantly removed with considering cloud fraction, and scavenged rate depends on aerosol and hydrometeor sizes (Slinn, 1984; Gong et al., 2003a) | Employing a scavenging coefficient approach based on relationships described by Seinfeld and Pandis (1998), considering accretion of in-cloud droplets particles into precipitation and impaction of ambient particles into precipitation | Discrete size-resolved coagulation between hydrometeors and aerosol particles (aerosol-liquid, aerosol-ice and aerosol-graupel) (GATOR2012) (Jacobson, 2003) |
| Sedimentation of aerosols (Aerosol mechanism) | Sedimentation with considering mass and number concentrations of aerosols at surface (MOSAIC) (Marelle et al., 2017) | Only considering gravitational sedimentation for aerosols (AERO5, AERO6 and AERO7) | Size-resolved sedimentation of aerosol particles above surface layer is computed with the setting velocity (CUACE) (Gong et al., 2003) | Using size-resolved sedimentation velocity to simulate sedimentation of aerosols (AERO5) | Sedimentation of size-resolved aerosols is computed from one model layer to layers below down to the surface, and the sedimentation velocities are calculated by two-step iterative method (GATOR2012) (Beard, 1976; Jacobson, 1997b, 2003) |

* GATOR2012 refers to either the aerosol or cloud microphysics scheme used in Jacobson (2012b).

Table 4 further lists various aspects with regards to how ARI and ACI being calculated in the
five two-way coupled models (WRF-Chem, WRF-CMAQ, GRAPES-CUACE, WRF-NAQPMS,
and GATOR-GCMOM) applied in Asia. Note that the information in this table was extracted from
the latest released version of WRF-Chem (version 4.3.3) and WRF-CMAQ (based on WRF v4.3
and CMAQ v5.3.3) as well as relevant references for GRAPES-CUACE (Wang et al., 2015), WRF-
NAQPMS (Wang et al., 2014) and GATOR-GCMOM (Jacobson et al., 2012). These models all use
the Mie theory to compute ARI effects but differ in representations of aerosol optical properties and
radiation schemes. To simplify the calculation, aerosol species simulated by the chemistry
module/model are put into different groups (Table 4) and the refractive indices of these groups are
directly from the Optical Properties of Aerosols and Clouds (OPAC) database (Hess et al., 1998) in
WRF-Chem and WRF-CMAQ (Table B6 in Appendix B). In WRF-Chem, the aerosol optical
properties (AOD, extinction/scattering/absorption coefficient, single scattering albedo and
asymmetry factor) are calculated in terms of four spectral intervals (listed in Table B6 in Appendix
B) and then inter/extrapolated to 11 (14) SW intervals defined in the GODDARD (RRTMG) scheme.
For SW and LW radiation in both WRF-CMAQ and WRF-Chem, these optical parameters are
computed at each of corresponding spectral intervals in the RRTMG scheme. The aerosol optical
property for LW radiation is considered only at 5 thermal windows (listed in Table B6) in WRF-
CMAQ. No detailed information regarding how aerosol optical property and relevant parameters
being calculated in GRAPES-CUACE and WRF-NAQPMS can be found from the relevant
references.
With respect to ACI effects, the simulated aerosol characteristics (such as mass, size
distribution and species) are utilized for the calculation of cloud droplet activation and aerosol
resuspension based on the Köhler theory (Abdul-Razzak and Ghan, 2002) in several (one)
microphysics schemes (scheme) in WRF-Chem (GRAPES-CUACE). GATOR-GCMOM is the first
two-way coupled model adding IN activation processes including heterogeneous and homogeneous
freezing (Jacobson et al., 2003). None of the other four two-way coupled models considers the IN
formation processes (including immersion freezing, deposition freezing, contact freezing, and
condensation freezing) but they have been included in some specific versions of WRF-Chem (Keita
et al., 2020; Lee et al., 2020), which are not yet in the latest release version 4.3.3 of WRF-Chem.

Table 4. Summary of relevant information regarding calculations of aerosol-radiation interactions (ARI) and aerosol-
cloud interactions (ACI) in two-way coupled models (WRF-Chem, WRF-CMAQ, GRAPES-CUACE, WRF-
NAQPMS and GATOR-GCMOM) applied in Asia.

| Model | ARI | | | | | ACI | |
|---|---|---|---|---|---|---|---|
| | Aerosol species groups | Aerosol size distribution (Aerosol mechanism) | Mixing state‡ | SW scheme (# of spectral intervals) | LW scheme (# of spectral intervals) | CCN (Microphysics scheme) | IN (Microphysics scheme) |
| WRF-Chem | 1. Water<br>2. Dust<br>3. BC<br>4. OC<br>5. Sea-salt<br>6. Sulfate | 1. Bulk (GOCART)<br>2. Modal (MADE/SORGAM, AERO5, MAM3 and MAM7)<br>3. Sectional (MOSAIC (4bins and 8 bins) and MADRID (8bins)) | Internal mixing (Volume averaging, Core-shell, and Maxwell-Garnett) | 1. Goddard (11)<br>2. RRTMG (14) | RRTMG (16) | Activation under a certain supersaturation in an air parcel based on Köhler theory (Morrison, Lin, Thompson, WSM 6/5/3 class and Milbrandt-Yau) | Ice heterogeneous nucleation of mineral dust aerosols in based on classical nucleation theory (Milbrandt-Yau and Morrison)† |
| WRF-CMAQ | 1. Water<br>2. Water-soluble<br>3. BC<br>4. Insoluble<br>5. Sea-salt | Modal (AERO5, AERO6 and AERO7) | Internal mixing (Core-shell) | RRTMG (14) | RRTMG (16) | None | None |
| GRAPES-CUACE | 1. Nitrate<br>2. Dust<br>3. BC<br>4. OC<br>5. Sea-salt<br>6. Sulfate<br>7. Ammonium | Sectional (CUACE (12 bins)) | External mixing | Goddard (11) | Goddard (10) | Activation under a certain supersaturation in an air parcel based on Köhler theory (WSM 6-class) | None |
| WRF-NAQPMS | 1. Nitrate<br>2. Dust<br>3. BC<br>4. OC<br>5. Sea-salt<br>6. Sulfate | Modal (AERO5) | External mixing | Goddard (11) | RRTM (16) | Activation under a certain supersaturation in an air parcel based on Köhler theory (Lin) | None |

| GATOR-GCMOM | 1. Water<br>2. Dust<br>3. BC<br>4. HCO$_3$<br>5. SOA<br>6. Sulfate<br>...<br>42. MgCO$_3$(s) | Sectional (GATOR2012*<br>(17-30 bins)) | Internal mixing<br>(Core-shell‡) | Toon* (318) | Toon* (376) | Activation under a certain supersaturation in an air parcel based on Köhler theory (GATOR2012*) | Ice heterogeneous and homogeneous nucleation (GATOR2012*) |
| 7. Ammonium<br>8. Other primary particles | | | | | | | |

‡ Specific version of WRF-Chem, WRF-NAQPMS and GOTAR-GCMOM have the ability of simulating aerosol aging (Zhang et al., 2014a; Chen et al., 2017d; Li et al., 2018a; Jacobson, 2012b).

† Some specific versions of WRF-Chem consider IN (Keita et al., 2020; Lee et al., 2020).

※ The short- and long-wave radiation calculations in GATOR-GCMOM are based on the algorithm of Toon et al. (1989).

* GATOR2012 refers to either the aerosol or cloud microphysics scheme used in Jacobson (2012b).

How accurately ARI and ACI are simulated also rely on the representation of aerosol composition and size distribution in two-way coupled models. Table 5 presents the treatments of aerosol compositions and size distributions in the five two-way coupled models applied in Asia. As shown in Tables 4 and 5, GATOR-GCMOM considered more detailed aerosol species groups as high as 42 kinds, and others coupled models different numbers of species groups (such as 6, 5, 7, 8 aerosol species groups in WRF-Chem, CMAQ, NAQPMS and CUACE, respectively). Three typical representation approaches of size distribution (bulk, modal and sectional methods) are adopted by the five two-way coupled models and WRF-Chem offers all the three approaches, but other models only support one specific option. The Global Ozone Chemistry Aerosol Radiation and Transport (GOCART) model (Ginoux et al., 2001) in WRF-Chem is the only one that is based on a combination of bulk (for water, BC, OC, and sulfate aerosols) and sectional (for dust and sea salt aerosols) approaches. The widely used modal and sectional approaches in five coupled models and their detailed numerical settings of aerosol size distribution (namely, geometric diameter and standard deviation for modal approach or bin ranges for sectional method) are listed in Table 5. Regarding the modal method, same parameter values for Aitken and accumulation modes and geometric diameters for coarse mode in the latest version of WRF-Chem (v4.3.3) and older version of WRF-CMAQ (before v5.2) are set as default, except the standard deviations for coarse mode are slightly different. In the official version of WRF-CMAQ released after v5.2, there are some modifications to the default setting of geometric diameters in Aitken, accumulation and coarse modes, from 0.01 to 0.015 μm, 0.07 to 0.08 μm and 1.0 to 0.6 μm, respectively. For the GRAPES-CUACE model, the parameters of size distribution for certain aerosol species in the accumulation mode were updated from its older version (Zhou et al., 2012) to newer one (Zhang et al., 2021). With respect to the sectional approach, 4 or 8 (from 0.039 to 10 μm), 12 (from 0.005 to 20.48 μm) and 14 (from 0.002 to 50 μm) particle size bins are defined in WRF-Chem, CUACE and GATOR-GCMOM, respectively.

Table 5. Summary of numerical representations of aerosol size distribution and composition in two-way coupled models (WRF-Chem, WRF-CMAQ, GRAPES-CUACE, WRF-NAQPMS and GATOR-GCMOM) applied in Asia.

| Model | Aerosol mechanism | Modal approach | | | | | | Compositions | Reference |
| | | Aitken | | Accumulation | | Coarse | | | |
| | | Geometric diameters (μm) | Standard deviations (μm) | Geometric diameters (μm) | Standard deviations (μm) | Geometric diameters (μm) | Standard deviations (μm) | | |
| WRF-Chem v4.3.3 | MADE/ SORGAM | 0.010 | 1.7 | 0.07 | 2.0 | 1.0 | 2.5 | Water, BC, OC, and sulfate, dust and sea salt | WRF-Chem codes※ |
| WRF-Chem△ | MAM3 | 0.013 (Sulfate and secondary OM) | 1.6 (Sulfate and secondary OM) | 0.068 (Sulfate, secondary OM, primary OM, BC, dust and sea salt) | 1.8 (Sulfate, secondary OM, primary OM, BC, dust and sea salt) | 2.0 (Sea salt), 1.0 (Dust) | 1.8 (Sea salt and dust) | Sulfate, methane sulfonic acid (MSA), OM, BC, sea salt and dust | Easter et al. (2004) Liu et al. (2012) |
| WRF-Chem△ | MAM7 | 0.013 (Sulfate and secondary OM and BC) | 1.6 (Sulfate, OM and BC) | 0.068 (Sulfate and BC)<br>0.068 (Primary OM)<br>0.2 (Sea salt)<br>0.11 (Dust) | 1.8 (Sulfate and BC)<br>1.6 (Primary OM)<br>1.8 (Sea salt)<br>1.8 (Dust) | 2.0 (Sea salt)<br>1.0 (Dust) | 2.0 (Sea salt)<br>1.8 (Dust) | Sulfate, methane sulfonic acid (MSA), OM, BC, sea salt and dust | Easter et al. (2004) Liu et al. (2012) |
| WRF-CMAQ | AERO5 | 0.010 | 1.7 | 0.07 | 2.0 | 1.0 | 2.2 | Water, water-soluble BC, | CMAQ codes* |

| | | | | | | | | | |
|---|---|---|---|---|---|---|---|---|---|
| (before CMAQ v5.2) | | | | | | | | insoluble, sea salt | |
| WRF-CMAQ (after CMAQ v5.2) | AERO6 and AERO7 | 0.015 | 1.7 | 0.08 | 2.0 | 0.60 | 2.2 | Water, water-soluble BC, insoluble, sea salt | CMAQ codes[†] |
| WRF-NAQPMS | AERO5 | 0.052 | 1.9 | 0.146 | 1.8 | 0.80 | 1.9 | Nitrate, dust, BC, OC, sea-salt, sulfate, ammonium, other primary particles | Wang et al. (2014) |
| GRAPES-CUACE | CUACE | 0.10 (BC and OC) | 1.7 (BC and OC) | 0.25 (Sulfate and nitrate) | 1.7 (Sulfate and nitrate) | 3.0 (Dust) | 1.7 (Dust) | Nitrate, dust, BC, OC, sea-salt, sulfate, ammonium[*] | Zhou et al. (2012) |
| GRAPES-CUACE | CUACE | Unclear | Unclear | 0.37 (BC and OC) | 0.42 (BC and OC) | Unclear | Unclear | Nitrate, dust, BC, OC, sea-salt, sulfate, ammonium[†] | Zhang et al. (2021) |
| WRF-Chem v4.3.3 | MOSAIC | Sectional approach<br>0.039-0.156, 0.156-0.625, 0.625-2.5, 2.5-10.0 μm (4 bins)<br>0.039-0.078, 0.078-0.156, 0.156-0.312, 0.312-0.625, 0.625-1.25, 1.25-2.5, 2.5-5.0, 5.0-10.0 μm (8 bins) | | | | | | Water, BC, OC, sulfate, dust and sea salt | WRF-Chem codes[§] |
| WRF-Chem[△] | MADRID | 0.0216-10 μm (8 bins) | | | | | | Water, BC, OC, and sulfate, dust and sea salt | Zhang et al. (2016d) |
| WRF-Chem v4.3.3 | GOCART | 0.1-1.0, 1.0-1.8, 1.8-3.0, 3.0-6.0, 6.0-10.0 (5 bins for dust)<br>0.1-0.5, 0.5-1.5, 1.5-5.0, 5.0-10.0 (4 bins for sea salt) | | | | | | Dust and sea salt | WRF-Chem codes[§] |
| GRAPES-CUACE | CUACE | 0.005-0.01, 0.01-0.02, 0.02-0.04, 0.04-0.08, 0.08-0.16, 0.16-0.32, 0.32-0.64, 0.64-1.28, 1.28-2.56, 2.56-5.12, 5.12-10.24, 10.24-20.48 μm (12 bins) | | | | | | Nitrate, dust, BC, OC, sea-salt, sulfate, ammonium | Zhou et al. (2012) |
| GATOR-GCMOM | GATOR2012 | 0.002-50 μm (14 bins) | | | | | | 42 species[‡] | Jacobson (2002, 2012b) |

[§] Official released version of WRF-Chem.
[△] Specific version of WRF-Chem.
[*] https://github.com/USEPA/CMAQ/blob/5.1/models/CCTM/aero/aero6/AERO_DATA.F.
[†] https://github.com/USEPA/CMAQ/blob/5.2/CCTM/src/aero/aero6/AERO_DATA.F.
[‡] More detailed components were presented in the first column of Table 2.
[*] Initial size distribution is tri-modal log-normal distribution.

Not only the choice of methodologies for ARI and ACI calculations can impact simulation
results, but also the various aspects regarding the setup of modeling studies by applying two-way
coupled models. The extra/auxiliary information about model configuration, including horizontal
and vertical resolutions, aerosol and gas phase chemical mechanisms, PBL schemes, meteorological
and chemical ICs and BCs, anthropogenic and natural emissions, were extracted from the 160 papers
and presented in Table S4 of Supplement, which is organized in the same order as Table 1.
For two-way coupled model applications in Asia, horizontal resolutions were set from a few to
several hundred kilometers, sometimes with nests, and vertical resolutions were from 15 to about
50-70 levels, with only one study performed at 100 levels for studying a fog case (Wang et al.,
2019b). Wang et al. (2018b) evaluated the impacts of horizontal resolutions on simulation results
and found out surface meteorological variables were better modeled at finer resolution but no
significant improvements of ACI related meteorological variables and certain chemical species
between different grid resolutions. Through applying a single column model and then WRF-Chem
with ARI, Wang et al. (2019b) unraveled that better representation of PBL structure and relevant
variables with finer vertical resolution from the surface to PBL top could reduce model biases
noticeably, but balancing between vertical resolution and computational resource was important as
well. Among the 160 applications of two-way coupled models in Asia, the frequently used aerosol
module and gas-phase chemistry mechanism in WRF-CMAQ (WRF-Chem) were AERO6
(MOSAIC and MADE/SOGARM) and CB05 (CBMZ and RADM2), respectively. For PBL
schemes, most studies selected YSU in WRF-Chem and ACM2 in WRF-CMAQ. Regarding to
meteorological ICs and BCs, the FNL data were the first choice, and outputs from the Model for
Ozone and Related Chemical Tracer (MOZART) were used to generate chemical ICs and BCs by
most researchers. Georgiou et al. (2018) also unraveled that boundary conditions of dust and $O_3$
played an important role in WRF-Chem simulations. The modeling applications in Asia utilized
global (EDGAR), regional (e.g., MIX, INTEX-B, and REAS), and national (e.g., MEIC and JEI-
DB) anthropogenic emission inventories. Natural emission sources, such as mineral dust (Shao,
2004), biomass burning (FINN (Wiedinmyer et al., 2011) and GFED (Giglio et al., 2010)), biogenic
VOCs (MEGAN (Guenther et al., 2006)), and sea salt (Gong et al., 1997) were also considered. It
should be noted that only one paper by Gao et al. (2017c) reported that the WRF-Chem model with
the Gridpoint Statistical Interpolation (GSI) data assimilation could improve the simulation
accuracy during a wintertime pollution period.

**4 Overview of research focuses in Asia**
**4.1 Feedbacks of natural aerosols**
**4.1.1 Mineral dust aerosols**

Due to the fact that dust storm events frequently occurred over Asia during 2000-2010, the
research community has focused on dust transportation and associated climatic effects (Gong et al.,
2003b; Zhang et al., 2003a, b; Yasunari and Yamazaki, 2009; Lee et al., 2010; Choobari et al., 2014).
Also the detailed processes and physiochemical mechanisms of dust storms had been well
understood and reviewed in detail (Shao and Dong, 2006; Uno et al., 2006; Huang et al., 2014; Chen
et al., 2017b). To probe into the radiative feedbacks of dust aerosols in Asia, Wang et al. (2010, 2013)
initiated modeling studies by a two-way coupled model, i.e., the GRAPES-CUAUE model, to
simulate direct radiative forcing (DRF) of dust, and revealed that the feedback effects of dust
aerosols could lead to decreasing of surface wind speeds and then suppress dust emissions. Further
modeling simulations by the same model (Wang and Niu, 2013) indicated that considering dust
radiative effects did not substantially improve the model performance of the air temperature at 2
meters above the surface (T2), even with assimilating data from in-situ and satellite observations
into the model. Subsequently, several similar studies based on another two-way coupled model
(WRF-Chem with GOCART scheme) were conducted to investigate dust radiative forcing
(including shortwave radiative forcing (SWRF) and longwave radiative forcing (LWRF)) and ARI
effects of dust on meteorological variables (PBLH, T2 and WS10) in different regions of Asia
(Kumar et al., 2014; Chen et al., 2014; Jin et al., 2015, 2016b; Liu et al., 2016a; Bran et al., 2018;
Su and Fung, 2018a, b; Zhou et al., 2018). These studies demonstrated that dust aerosols could
induced negative radiative forcing (cooling effect) at top of atmosphere (TOA) as well as the surface
(including both Earth's and sea surfaces) and positive radiative forcing (warming effect) in the ATM
(Wang et al., 2013; Chen et al., 2014; Kumar et al., 2014; Li et al., 2017c; Bran et al., 2018; Li and
Sokolik, 2018; Su and Fung, 2018b). More thorough analyses of the radiative effects of dust in Asia
(Wang et al., 2013; Li and Sokolik, 2018) pointed out that dust aerosols played opposite roles in the
shortwave and longwave bands, so that the dust SWRF at TOA and the surface (cooling effects) as
well as in the ATM (warming effects) was offset partially by the dust LWRF (warming effects at
TOA and the surface but cooling effects in the ATM). It was noteworthy that adding more detailed
mineralogical composition into the dust emission for WRF-Chem could alter the dust SWRF at TOA
from cooling to warming and then lead to a positive net radiative forcing at TOA (Li and Sokolik,
2018). These different conclusions showed some degrees of uncertainties in the coupled model
simulations of dust aerosols' radiative forcing that need to be further investigated in the future.
Dust aerosols can act not only as water-insoluble cloud condensation nuclei (CCN) (Kumar et
al., 2009) but also as ice nuclei (IN) (Lohmann and Diehl, 2006) since they are referred to as ice
friendly (Thompson and Eidhammer, 2014). Therefore, activation and heterogeneous ice nucleation
parameterizations (INPs) with respect to dust aerosols were developed and incorporated into WRF-
Chem to explore ACI effects as well as both ARI and ACI effects of dust aerosols in Asia (Jin et al.,
2015, 2016b; Zhang et al., 2015c; Su and Fung, 2018a, b; Wang et al., 2018b). During dust storms,
including the adsorption activation of dust particles played vital roles in the simulations of ACI-
related cloud properties and a 45 % of increase of cloud droplet number concentration (CDNC),
comparing to a simpler aerosols activation scheme in WRF-Chem (Wang et al., 2018b). More
sophisticated INPs implemented in WRF-Chem that taking dust particles into account as IN resulted
in substantial modifications of cloud and ice properties as well as surface meteorological variables
and air pollutant concentrations in model simulations (Zhang et al., 2015c; Su and Fung, 2018b).
Zhang et al. (2015c) delineated that dust aerosols acting either as CCN or IN made model results
rather different regarding radiation, T2, precipitation, and number concentrations of cloud water and
ice. Su and Fu (2018b) described that the ACI effects of dust had less impacts on the radiative forcing
than its ARI effects and dust particles could promote (demote) ice (liquid) clouds in mid-upper (low-
mid) troposphere over EA. With turning on both ARI and ACI effects of dust, less low-level clouds
and more mid- and high-level clouds were detected that contributed to cooling at the Earth's surface
and in the lower atmosphere and warming in the mid-upper troposphere (Su and Fung, 2018b).
Mineral dust particles transported by the westerly and southwesterly winds from the Middle East
(ME) affected the radiative forcing at TOA and the Earth's surface and in the ATM by the dust-
induced ARI and ACI in the Arabian Sea and the India subcontinent, and subsequently changed the
circulation patterns, cloud properties, and characteristics related to the India summer monsoon (ISM;
Jin et al., 2015, 2016a). Moreover, the effects of dust on precipitation are not only complex but also
highly uncertain, evidencing from several modeling investigations targeting a variety of areas in
Asia (Jin et al., 2015, 2016a, b; Zhang et al., 2015c; Su and Fung, 2018b). Less precipitation from
model simulations including dust effects was found at EA and dust particles acting mainly as CCN
or IN influenced precipitation in a rather different way (Zhang et al., 2015c). A positive response of
ISM rainfall to dust particles from the ME was reported by Jin et al. (2015) and less affected by dust
storms from the local sources and NWC (Jin et al., 2016b). Jin et al. (2016a) further elucidated that
the impacts of ME dust on ISM rainfall were highly sensitive to the imaginary refractive index of
dust setting in the model, so that accurate simulations of the dust-rainfall interaction depended on
more precise representation of radiative absorptions of dust in two-way coupled models. About 20 %
of increase or decrease in rainfall due to the dust effects were detected in different areas over EA
from the WRF-Chem simulations (Su and Fung, 2018b). However, it should be mentioned that a
few studies that targeting DRF of dust in Asia based on WRF-Chem simulations but without
enabling aerosol-radiation feedbacks (Ashrafi et al., 2017; Chen et al., 2017c; Tang et al., 2018)
were not included in this paper.
Along with the modeling research on the effects of dust aerosols on meteorology, their impacts
on air quality in Asia were explored using two-way coupled models (Wang et al., 2013; Chen et al.,
2014; Kumar et al., 2014; Li et al., 2017c; Li and Sokolik, 2018). Many early modeling research
work involving two-way coupled models with dust only looked into the ARI or direct radiative
effects of dust particles, which are described as follows. Taking a spring-time dust storm from the
Thar Desert into consideration in WRF-Chem, the modeled aerosol optical depth (AOD) and
Angstrom exponent (as indicators of aerosol optical properties and unique proxies of the surface
particulate matter pollution) demonstrated that turning on the ARI effects of dust could reduce biases
in their simulations, but were underestimated in North India (Kumar et al., 2014). Wang et al. (2013)
pointed out that in EA, including the longwave radiative effects of dust in the GRAPES-
CUACE/dust model lowered relative errors of the modeled AOD by 15 %, as compared to
simulations that only considering shortwave effects of dust. Comparisons against both satellite and
in situ observations depicted that the WRF-Chem model was able to capture the general
spatiotemporal variations of the optical properties and size distribution of dust particles over the
main dust sources in EA, such as the Taklimakan Desert and Gobi Desert, but overestimated AOD
during summer and fall and also exhibited positive (negative) biases in the fine (coarse) mode of
dust particles (Chen et al., 2014). Besides the ARI effects of dust, the heterogeneous chemistry on
dust particles' surface added in WRF-Chem was accounted for 80 % of the net reductions of $O_3$,
$NO_2$, $NO_3$, $N_2O_5$, $HNO_3$, $\cdot OH$, $HO_2\cdot$ and $H_2O_2$ when a springtime dust storm striking the Nanjing
megacity of EC (Li et al., 2017c). In CA, AOD was overestimated by WRF-Chem model but its
simulation was improved when more detailed mineral components of dust particles were
incorporated in the model (Li and Sokolik, 2018). Later on, more investigations started to focus on
both ARI and ACI effects of dust aerosols. With consideration of ARI as well as both ARI and ACI
of dust particles from the ME, during the ISM period, the WRF-Chem model reproduced AOD's
spatial distributions but underpredicted (overpredicated) AOD over the Arabian Sea (the Arabian
Peninsula) comparing with satellite observations and AOD reanalysis data (Jin et al., 2015, 2016a,
b). In EA, Wang et al. (2018b) demonstrated that including both ARI and ACI effects of dust in
WRF-Chem caused lower $O_3$ concentrations and by incorporating INPs, the WRF-Chem model well
simulated the surface $PM_{10}$ concentrations (Su and Fung, 2018a) with reduced (elevated) surface
concentrations of OH, $O_3$, $SO_4^{2-}$, and $PM_{2.5}$ (CO, $NO_2$, and $SO_2$) (Zhang et al., 2015c). It is worth
noting that how to partition dust particles into fine mode and coarse mode or initialize their size
distribution in coupled models can affect simulations in many ways and requires more detailed
measurements at the source areas and further modeling studies.
**4.1.2 Wildfire, sea salt and volcanic ash**
In the Maritime SEA region, peat and forest fire triggered by El Niño induced drought
conditions released huge amount of smoke particles, which promoted dire air pollution problems in
the downstream areas, and their ARI effects simulated by WRF-Chem enhanced radiative forcing at
the TOA and the atmospheric stability (Ge et al., 2014). Ge et al. (2014) also pointed out the ARI
effects of these fires impaired (intensified) sea breeze at daytime (land breeze at nighttime) over this
region so that their impacts on cloud cover could be positive or negative in different areas and time
period (day or night). Sea salt and volcanic ash are also important natural aerosols for regions near
seashores and active volcanoes and surrounding areas but modeling studies of their ARI and ACI
effects are relatively scarce in Asia. Based on WRF-Chem simulations, Kedia et al. (2019b)
demonstrated that the feedbacks of sea salt aerosols impacted convective and nonconvective
precipitation rather variously in different areas of the India subcontinent. Jiang et al. (2019a, b) also
used WRF-Chem with/without sea-salt emissions to evaluate the effects of sea salt on rainfall in
Guangdong Province of China, but unfortunately, no feedbacks were considered in the simulations.
So far there is no investigation targeting aerosol effects of volcanic ash from volcano eruptions in
Asia using coupled models.
**4.2 Feedbacks of anthropogenic aerosols**
Atmospheric pollutants from anthropogenic sources are the leading causes of heavy pollution
events occurring in Asia due to the acceleration of urbanization, industrialization, and population
growth in recent decades, particularly in China and India, and their ARI or/and ACI effects on
meteorology and air quality had been quantitatively examined using two-way coupled models
(Kumar et al., 2012a, b; Li and Liao, 2014; Wang et al., 2014a; Zhang et al., 2015a; Gao et al., 2016a;
Yao et al., 2017; Wang et al., 2018d; Archer-Nicholls et al., 2019; Bharali et al., 2019). These
modeling research work had been primarily focused on the ARI or/and ACI effects of anthropogenic
aerosols, their specific chemical components (especially the light-absorbing aerosols, i.e., BC and
brown carbon (BrC)) and aerosols originated from different sources. The major findings are outlined
as follows, with respect to the effects of anthropogenic aerosol feedbacks on meteorology and air
quality.
Concerning the meteorological responses, most papers treated anthropogenic aerosols as a
whole to explore their effects on meteorological variables based on coupled model simulations with
enabling ARI or/and ACI in WRF-Chem, WRF-CMAQ, WRF-CMAQ, GRAPES-CUACE and
WRF-NAQPMS (Kumar et al., 2012a; Wang et al., 2014a, c, 2015a; Zhang et al., 2015a, 2018;
Zhao et al., 2017; Nguyen et al., 2019a, b; Bai et al., 2020). Generally, the main ARI effects of
anthropogenic aerosols resulted in decreases of SWRF, T2 and WS10, and PBLH, as well as
increases of surface relative humidity (RH2) and temperature in the ATM, which further suppressed
PBL development (Gao et al., 2015b; Xing et al., 2015a; Li et al., 2017b; Zhang et al., 2018; Nguyen
et al., 2019a, b). Wang et al. (2015a) utilized GRAPES-CUACE with ARI to study a summer haze
case in the NCP area and discovered that the ARI effects made the subtropical high less intense (-
14 hPa) to help pollutants in the area to dissipate. In Asia, ACI effects of anthropogenic aerosols on
cloud properties and precipitation are relatively complex. On the one hand, anthropogenic aerosols,
that being activated as CCN, enhanced CDNC and LWP and then slowed down the precipitation
onset, but their impacts on precipitation amounts varied in different seasons and areas in China
(Zhao et al., 2017). Targeting a summertime rainstorm in the middle reaches of the Yangtze River
(MRYR) in China, sensitivity studies using WRF-Chem unveiled that CDNC, cloud water contents,
and precipitation decreased (increased) with low (high) anthropogenic emission scenarios due to the
ACI effects and these variations tended to depend on atmospheric humidity (Bai et al., 2020). The
modeling investigations with WRF-Chem aiming at the ISM (Kedia et al., 2019b) and a disastrous
flood event in Southwest China (SWC) (Fan et al., 2015) pointed out that the simulated convective
process was suppressed and convective (nonconvective) precipitation was inhibited (enhanced) by
the ARI and ACI effects of accumulated anthropogenic aerosols, but these effects could invigorate
convection and rainfall in the downwind mountainous area at nighttime (Fan et al., 2015). On the
other hand, how anthropogenic aerosols act in the ice nucleation processes is still open to question
(Zhao et al., 2019) and these processes need to be represented accurately in two-way coupled models,
however until now no study had been performed to simulate the ACI effects of anthropogenic
aerosol serving as IN in Asia using two-way coupled models. Therefore, in Asia, further
investigations are needed that targeting cloud or/and ice processes involving anthropogenic aerosols
(including their size, composition, and mixing state) in two-way coupled models. Meanwhile,
several studies not only discussed aerosol feedbacks but also focused on the additional effects of
topography or urban heat island on meteorology (Zhong et al., 2015, 2017; Wang et al., 2019a).
Utilizing the GATOR-GCMOM model at global and local scales, Jacobson et al. (2015, 2019)
explored the impacts of landuse changes due to the unprecedented expansions of megacities, such
as Beijing and New Delhi in Asia, from 2000 to 2009 on meteorology and air quality.
Hitherto there were several attempts to ascertain the effects of different chemical components
of anthropogenic aerosols on meteorology in Asia (Huang et al., 2015; Ding et al., 2016, 2019; Gao
et al., 2018a; Wang et al., 2018d; Archer-Nicholls et al., 2019). First of all, Asia is the region in the
world with the highest BC emissions due to burning of large amount of fossil fuels and biomass and
this has increasingly attracted many researchers to probe into the ARI or/and ACI effects of BC
(IPCC, 2014). As the most important absorbing aerosol, BC induced the largest positive, positive
and negative mean DRF at the TOA, in the ATM, and at the surface, respectively, over China during
2006 (Huang et al., 2015). Ding et al. (2016) and Wang et al. (2018d) further applied WRF-Chem
with feedbacks to investigate how aerosol-PBL interactions involving BC suppressed the PBL
development, which deteriorated air quality in Chinese cities and was described as "dome effect"
(namely BC warms the atmosphere and cools the surface, suppresses the PBL development and
eventually results in more accumulation of pollutants). This "dome effect" of BC promoted the
advection-radiation fog and fog-haze formation in the YRD area through altering the land-sea
circulation pattern and increasing the moisture level (Ding et al., 2019). Gao et al. (2018a) also
pointed out BC in the ATM modified the vertical profiles of heating rate and equivalent potential
temperature in Nanjing, China. In India, the ARI effects of BC enhanced convective activities,
meridional flows, and rainfall in North-East India during the pre-monsoon season but could either
enhance or suppress precipitation during the monsoon season in different parts of the India
subcontinent (Soni et al., 2018). Moreover, the ARI effects of BC on surface meteorological
variables were larger than its ACI effects in EC (Archer-Nicholls et al., 2019; Ding et al., 2019).
Besides BC, the BrC portion of organic aerosols (OA) emitted from agriculture residue burning
(ARB) were included in WRF-Chem with the parameterization scheme suggested by Saleh et al.
(2014) and the model simulations in EC revealed that at the TOA, the net DRF of OA was -0.22
W·m$^{-2}$ (absorption and scattering DRF were +0.21 W·m$^{-2}$ and -0.43 W·m$^{-2}$ respectively), but the
BC's DRF was still the highest (+0.79 W·m$^{-2}$) (Yao et al., 2017). As mentioned above, it is obvious
that ARI and ACI effects of different aerosol components are substantially distinctive, and many
other aerosol compositions (e.g., sulfate, nitrate and ammonium) besides BC and BrC should be
taken into considerations in future modeling studies in Asia.
ARB is a common practice in many Asian countries after harvesting and before planting and
can deteriorate air quality quickly as one of the most important sources of anthropogenic aerosols,
so that it has been attracting much attention among the public and scientists worldwide (Reid et al.,
2005; Koch and Del Genio, 2010; Chen et al., 2017a; Yan et al., 2018; Hodshire et al., 2019).
Recently, the effects of ARB aerosols on meteorology had widely been explored using the two-way
coupled model (WRF-Chem) in many Asian countries and regions, such as EC (Huang et al., 2016;
Wu et al., 2017; Yao et al., 2017; Li et al., 2018b), South China (SC) (Huang et al., 2019), and South
Asia (SA) (Singh et al., 2020). In general, when ARB occurred, the WRF-Chem simulations from
all the studies showed that the changes in radiative forcing induced by ARB aerosols were greater
than by those from other anthropogenic sources, especially in the ATM. Also all the modeling studies
indicated that ARB aerosols reduced (increased) radiative forcing at the surface (in the ATM), cooled
(warmed) the surface (the atmosphere), and increased (decreased) atmospheric stability (PBLH).
Furthermore, the WRF-Chem simulations with ARI demonstrated that light-absorbing carbonaceous
aerosols (CAs) from ARB caused daytime (nighttime) precipitation decreased (increased) over
Nanjing in EC during a post-harvest ARB event (Huang et al., 2016). Yao et al. (2017) pointed out
their WRF-Chem simulations in EC exhibited larger DRE induced by BC from ARB at the TOA
than previous studies. Lately, several modeling studies using WRF-Chem had targeted the effects of
ARI and both ARI and ACI due to ARB aerosols from countries in the Indochina, SEA, and SA
regions during the planting and harvesting time (Zhou et al., 2018; Dong et al., 2019; Huang et al.,

2019; Singh et al., 2020). Zhou et al. (2018) investigated how ARB aerosols from SEA mixed with mineral dust and other anthropogenic aerosols while being lifted to the mid-low troposphere over the source region and transported to the YRD area and then affected meteorology and air quality there. The influences of ARI and ACI caused by ARB aerosols from Indochina were contrary, namely, the ARI (ACI) effects made the atmosphere over SC warmer (cooler) and drier (wetter), and the ARI effects hindered cloud formation and suppressed precipitation there (Huang et al., 2019). Dong et al. (2019) found the warming ARI effects of ARB aerosols were smaller over the source region (i.e., SEA) than the downwind region (i.e., SC) with cloudier conditions. Annual simulations regarding the ARI effects of ARB aerosols from SA (especially Myanmar and Punjab) indicated that CAs released by ARB reduced the radiative forcing at the TOA but did not change the precipitation processes much when only the ARI effects were considered in WRF-Chem (Singh et al., 2020).

Besides ARB, to our best knowledge, there were only a few research work quantitatively assessing the effects of anthropogenic aerosols from different emission sources on meteorology using WRF-Chem. Gao et al. (2018c) evaluated the responses of radiative forcing in China and India to aerosols from five emission sectors (power, industry, residential, BB, and transportation), and found that the power (residential) sector was the dominate contributor to the negative (positive) DRF at the TOA over both countries due to high emissions of sulfate and nitrate precursors (BC) and the total sectoral contributions were in the order of power > residential > industry > BB > transportation (power > residential > transportation > industry > BB) for China (India) during 2013. To pinpoint the ARI and ACI effects, Archer-Nicholls et al. (2019) reported that during January 2014, the aerosols from the residential emission sector induced larger SWRF (+1.04 $W \cdot m^{-2}$) than LWRF (+0.18 $W \cdot m^{-2}$) at the TOA and their DRF (+0.79 $W \cdot m^{-2}$) was the largest, followed by their semidirect effects (+0.54 $W \cdot m^{-2}$) and indirect effects (-0.29 $W \cdot m^{-2}$) over EC. This study further emphasized a realistic ratio of BC to total carbon from the residential emission was critical for accurate simulations of the ARI and ACI effects with two-way coupled models.

In terms of anthropogenic aerosol effects on air quality, the responses of $PM_{2.5}$ had been widely investigated (Wang et al., 2014a, c, 2015a; Gao et al., 2015b, 2016a, 2018a; Zhang et al., 2015a, 2018; Zhao et al., 2017; Chen et al., 2019a; Nguyen et al., 2019a, b; Wu et al., 2019a) but less studies explored the responses of $O_3$ and other species (Kumar et al., 2012a; Zhang et al., 2015a; Xing et al., 2017; Li et al., 2018a; Nguyen et al., 2019a, b). As summarized by Wu et al. (2019a) in their Table 1, observations and model simulations with WRF-Chem, WRF-CMAQ, WRF-CMAQ, GRAPES-CUACE, and WRF-NAQPMS all pointed out that the ARI effects promoted higher $PM_{2.5}$ concentrations in China (Wang et al., 2014a, c, 2015a; Gao et al., 2015b, 2016a; Zhang et al., 2015a, 2018; Chen et al., 2019a) and this was also true in other areas of Asia (e.g., India, EA, Continental SEA) (Gao et al., 2018c; Nguyen et al., 2019a, b) during different seasons. At the same time, all the modeling investigations revealed that the positive aerosol-meteorology feedbacks could further exacerbate pollution problems during heavy haze episodes. Based on WRF-Chem simulations, the ACI effects on $PM_{2.5}$ was negligible comparing to the ARI effects over EC (Zhang et al., 2015a) but was subject to a certain degree of uncertainty if no consideration of the ACI effects induced by cumulus clouds in the model (Gao et al., 2015b). Annual WRF-Chem simulations for 2014 by Zhang et al. (2018) indicated that even though the ARI effects had bigger impacts on $PM_{2.5}$ during wintertime than the ACI effects, the ARI and ACI impacts on $PM_{2.5}$ were similar during other seasons and the increase of $PM_{2.5}$ due to the ACI effects was more noticeable in wet season than dry season. Using the process analysis method to distinguish the contributions of different physical and chemical processes to $PM_{2.5}$ over the NCP area, Chen et al. (2019a) applied WRF-Chem with ARI and ACI and found that besides local emissions and regional transport processes, vertical mixing contributed the most to the accumulation and dispersion of $PM_{2.5}$, comparing to chemistry and advection, and the ARI effects changed the vertical mixing contribution to daily $PM_{2.5}$ variation from negative to positive. Regarding surface $O_3$ concentrations, all the two-way coupled models with ARI, ACI, and both ARI and ACI predicted reduced photolysis rate and $O_3$ concentrations under heavy pollution conditions, through the radiation attenuation induced by aerosols and clouds. Further analyses indicated that the ARI effects impacted $O_3$ positively through reducing vertical dispersions (WRF-CMAQ, Xing et al., 2017), reduced $O_3$ more during wintertime than summertime in EC (WRF-NAQPMS, Li et al., 2018a), and suppressed (enhanced) $O_3$ in dry (wet) season in continental SEA (WRF-CMAQ, Nguyen et al., 2019b). Xing et al. (2017) applied the process analysis method in WRF-CMAQ with ARI and revealed that the impacts of ARI on the contributions of atmospheric dynamics and photochemistry processes to $O_3$ over China varied in winter and

summer months and ARI induced largest changes in photochemistry (dry deposition) of surface $O_3$
at noon time in January (July). The process analysis in WRF-Chem with ARI and ACI identified
that the vertical mixing process played the most important role among the other physical and
chemical processes (advection and photochemistry) in surface $O_3$ growth during 10-14 local time in
Nanjing, China (Gao et al., 2018a). ARI and ACI not only affected $PM_{2.5}$ and $O_3$, but also other
chemical species. For instance, CO and $SO_2$ increased due to ARI and ACI over EC (Zhang et al.,
2015a), ARI caused midday (daily average) OH increased (decreased) in July (January) over China
(Xing et al., 2017), $SO_2$, $NO_2$, BC, $SO_4^{2-}$, $NO_3^-$ were enhanced but OH was reduced over China by
ACI (Zhao et al., 2017), and ARI impacted $SO_2$ and $NO_2$ positively over EA (Nguyen et al., 2019a).
Wu et al. (2019b) further analyzed how the aerosol liquid water involved in ARI and chemical
processes (i.e., photochemistry and heterogeneous reactions) and influenced radiation and $PM_{2.5}$
(esp. secondary aerosols) over NCP during an intense haze event. Moreover, evaluations and
sensitivity studies indicated that turning on aerosol feedbacks could improve the model performance
for surface $PM_{2.5}$, particularly during severe haze episodes (Zhang et al., 2015a, 2018; Li et al.,
2018a; Wang et al., 2018a).
With reference to the feedback effects of anthropogenic aerosol compositions on air quality,
most modeling research work with WRF-Chem had focused on the ARI and ACI effects of BC and
BrC, especially the "dome effect" that prompted the accumulation of pollutants (aerosols and $O_3$)
near surface and in PBL (Li and Liao, 2014; Ding et al., 2016, 2019; Gao et al., 2018a; Wang et al.,
2018d). At the same time, the ARI effects of BC undermined the low-level wind convergence and
then led to decrease of aerosols (sulfate and nitrate) and $O_3$ (Li and Liao, 2014). With the process
analysis methodology in WRF-Chem, Gao et al. (2018a) indicated that comparing to simulations
without BC, the BC and PBL interaction slowed the $O_3$ growth from late morning to early afternoon
somewhat before $O_3$ reaching its maximum value at noon due to less vertical mixing in PBL.
Studies on the feedback effects of aerosols from different emission sectors on air quality were
relatively limited and mainly involved with ARB emissions and assessments of emission controls
during certain major air pollution events. Jena et al. (2015) applied WRF-Chem with aerosol
feedbacks and investigated $O_3$ and its precursors in SA due to regional ARB. Based on WRF-Chem
simulations with enabling ARI and ACI, Wu et al. (2017) denoted that aerosols emitted from ARB
could be mixed or/and coated with urban aerosols while being transported to cities and contributed
to heavy air pollution events there, such as in Nanjing, China. The ARI effects induced by ARB
aerosols on $O_3$ and $NO_2$ concentrations (-1 % and 2 %, respectively) were small compared to the
contribution of precursors emitted from ARB to $O_3$ chemistry (40 %) in the ARB zone (Li et al.,
2018b). Pollutants emitted from natural and anthropogenic BB over Indochina affected pollution
levels over SC and their ACI effects removed aerosols more efficiently than the ARI effects that
could make BB aerosols last longer in the ATM (Huang et al., 2019). Gao et al. (2017b) and Zhou
et al. (2019) both utilized WRF-Chem to evaluate what role the ARI effects played when dramatic
emission reductions implemented during the week of Asia Pacific Economic Cooperation Summit
and concluded that the ARI reduction induced by decreased emission led to 6.7-10.9 % decline in
$PM_{2.5}$ concentrations in Beijing.

**4.3 Human health effects**

Poor air quality posts risks to human health (Brunekreef and Holgate, 2002; Manisalidis et al.,
2020), therefore, in the past several decades, air quality models had been used in epidemiology
related research to establish quantitative relationships between concentrations of various pollutants
and burden of disease (including mortality or/and morbidity) as well as associated economic loss
(Conti et al., 2017). In Asia, there were several studies that applied coupled air quality models with
feedbacks to assess human health effects of air pollutants under historical and future scenarios (Gao
et al., 2015a, 2017c; Ghude et al., 2016; Xing et al., 2016; Wang et al., 2017; Conibear et al., 2018a,
b; Hong et al., 2019; Zhong et al., 2019). By applying WRF-Chem with ARI and ACI, Gao et al.
(2015a) estimated the health and financial impacts induced by an intense air pollution event
happened in the NCP area during January, 2013 and concluded that the mortality, morbidity, and
financial loss over Beijing area were 690, 69070, and 253.8 million US$, respectively. Targeting
the same case, Gao et al. (2017c) pointed out that turning on the data assimilation of surface $PM_{2.5}$
observations in WRF-Chem not only improved model simulations but also made the premature
death numbers increased by 2 % in the NCP area, comparing to simulations without the $PM_{2.5}$ data
assimilation. In India, WRF-Chem simulations with aerosol feedbacks and updated population data
revealed that the premature (COPD related) deaths caused by $PM_{2.5}$ ($O_3$) were 570,000 (12,000),
resulting in shortened life expectancy (3.4±1.1 years) and financial expenses (640 million US$)
during 2011 (Ghude et al., 2016). Based on WRF-CMAQ simulations with ARI for 21 years (1990-
2010), Xing et al. (2016) pointed out that in EA the population-weighted $PM_{2.5}$ induced mortality
had an upward trend from 1990 (+3187) to 2010 (+3548) and the mean mortality caused by ARI-
enhanced $PM_{2.5}$ was 3.68 times more than that decreased by ARI-reduced temperature. The same
21 year simulations also showed that from 1990 to 2010, the $PM_{2.5}$ related mortalities in EA and
SA rose by 21 % and 85 %, respectively, while they declined in Europe and high-income North
America by 67 % and 58 %, respectively (Wang et al., 2017). Conibear et al. (2018a) applied WRF-
Chem with ARI to study how different emission sectors affected human health in India and
demonstrated that the residential energy use sector played the most critical role among other sectors
and could cause 511,000 premature deaths in 2014. Furthermore, Conibear et al. (2018b)
investigated future $PM_{2.5}$ pollution levels as well as health impacts in India under different emission
scenarios (business as usual and two emission control pathways) and deduced that the burden of
disease driven by $PM_{2.5}$ and population factors (growth and aging) in 2050 increased by 75 % under
the business as usual scenario but decreased by 9 % and 91 % under the International Energy
Agencies New Policy Scenario and Clean Air Scenario, respectively, comparing with that in 2015.
The sensitivity study using WRF-Chem with ARI under a variety of emission scenarios, population
projections, and concentration-response functions (CRFs) for the years of 2008 and 2050
demonstrated that CRFs (future emission projections) were the main sources of uncertainty in the
total mortality estimations related to $PM_{2.5}$ ($O_3$) in China (Zhong et al., 2019). Applying a suite of
models, including WRF-CMAQ with ARI, climate and epidemiology, Hong et al. (2019) inferred
that under Representative Concentration Pathway 4.5, the future mortalities could be 12100 and
8900 per year in China led by $PM_{2.5}$ and $O_3$, respectively, and the climate-driven weather extremes
could add 39 % and 6 % to future mortalities due to stable atmosphere and heat waves, respectively.
Ten Hoeve and Jacobson (2012) applied GATOR-GCMOM and a human exposure model to
estimate the local and worldwide health effects induced by the 2011 Fukushima nuclear accident
and a hypothetical one in California of US.
## 5 Effects of aerosol feedbacks on model performance

Even though there are a certain number of research papers using two-way coupled models to
quantify the effects of aerosol feedbacks on regional meteorology and air quality in Asia, model
performances impacted by considering aerosol effects varied to some extent. This section provides
a summary of model performance by presenting the SI of meteorology and air quality variables as
shown in Table S2. These SI were collected from the selected papers that supplying these indices
and being defined as papers with SI (PSI) (listed in Tables B2-B3 of Appendix B). As
aforementioned in Section 3, investigations of ACI effects were very limited and no former studies
simultaneously exploring aerosol feedbacks with and without both ARI and ACI turned on. Here,
we only compared the SI for simulations with and without ARI in the same study, as summarized in
Appendix Tables B4-B5. It should be pointed out that all the reported evaluation results either from
individual model or inter-model comparison studies were extracted and put into the Table S2.
### 5.1 Model performance for meteorology variables

With certain emissions, accurate simulations of meteorological elements are critical to air
quality modeling and prediction (Seaman, 2000; Bauer et al., 2015; Appel et al., 2017; Saylor et al.,
2019). Targeting meteorological variables, we summarized their SI and further analyzed the
variations of SI on different simulated time scales and among multiple models.
### 5.1.1 Overall performance

Figure 3 shows the compiled statistical indicators (correlation coefficient (R) is in black, and
mean bias (MB) and root mean square error (RMSE) are in blue) of T2 (°C), RH2(%) and specific
humidity (SH2, $g \cdot kg^{-1}$) at 2 meters, and WS10 ($m \cdot s^{-1}$) from PSI (a-d), and simulations with and
without ARI (marked as ARI and NO-ARI in e-h). In this figure and following figures, NP and NS
are number of publications and samples with SI, respectively and summed up in Appendix Table
B2. In these two tables, we also listed the NS of positive (red upward arrow) and negative (blue
downward arrow) biases for the meteorological and air quality variables in parentheses in the MB

column. Note that NS in Fig. 3e-h and Appendix Table B4 counted the samples of SI provided by the simulations simultaneously with and without ARI. Also, the 5th, 25th, 75th and 95th percentiles of SI are illustrated in box-and-whisker plots, and the dashed line in the box is the mean value (not median) and the circles are outliers.

The evaluations for T2 (Fig. 3a) from PSI revealed that in Asia coupled models performed rather well for temperature (mean R = 0.90) with RMSE ranging from 0.64 to 5.90 °C, but 60 % of samples showed the tendency towards temperature underestimations (mean value of MB = -0.20 °C) with the largest average MB (-0.31 °C) occurring during winter months (70 samples). The underestimations of temperature had been reported not only from modeling studies by using WRF or coupled models, but also in Asia, Europe and North America (García-Díez et al., 2013; Brunner et al., 2015; Makar et al., 2015a; Yahya et al., 2015; Gao et al., 2019). The WRF simulations in China (Gao et al., 2019) and US (EPA, 2018) also showed wintertime cold biases of T2 but in Europe warm biases were reported (García-Díez et al., 2013). This temperature bias was probably related to the impacts of model resolutions (Kuik et al., 2016), urban canopies (Liao et al., 2014) and PBL schemes (Hu et al., 2013). With the ARI turned on in the coupled models, modeled temperatures (limited papers with 12 samples) were improved somewhat and the mean correlation coefficient increased from 0.93 to 0.95 and RMSE decreased slightly (Fig. 3e), but average MB of temperature was decreased from -0.98 to -1.24 °C. In short, temperatures from PSI or simulations with/without ARI turned on agreed well with observations but were mostly underestimated, and the negative bias of T2 simulated by models with ARI turned on got worse and reasons behind it will be explained in Section 6.

Figures 3b-c illustrate that RH2 was simulated reasonably well (mean R = 0.73) and the modeled SH2 was also well correlated with observations (R varied between 0.85 and 1.00). RH2 and SH2 from more than half of samples had slightly positive and negative mean biases with average MB values of 0.4 % and -0.01 g·kg$^{-1}$, respectively. The overestimations of RH2 could be caused by the negative bias of T2 (Cuchiara et al., 2014). Compared with results without ARI effects, statistics of RH2 and SH2 from simulations with ARI showed better R and RMSE. However, the increased positive mean biases (average MBs of RH2 and SH2 were from 6.4 % to 7.6 % and from 0.07 g·kg$^{-1}$ to 0.11 g·kg$^{-1}$, respectively) indicated that turning on ARI could cause further overprediction of humidity variables. Overall, the modeled RH2 and SH2 were in good agreement with observations with slight over- and under-estimations, respectively, and the limited studies showed that RH2 and SH2 simulated by models with ARI turned on had marginally larger positive biases relative to the results without ARI.

Compared with the correlation coefficients of T2, RH2 and SH2, mean R (0.59) of WS10 was smallest with a large fluctuation ranging from 0.14 to 0.98 (Fig. 3d). The meta-analysis also indicated that most modeled WS10 tended to be overestimated (81 % of the samples) with the average MB value of 0.79 m·s$^{-1}$, and the mean RMSE value was 2.76 m·s$^{-1}$. The general overpredictions of WS10 by WRF (Mass and Ovens, 2011) and coupled models (Gao et al., 2015b, 2015bs) had been explained with possible reasons such as out-of-date geographical data, coarse model resolutions and lacking of better representations of urban canopy physics. The PSI with ARI effects suggested that the correlation of wind speed was slightly improved (mean R from 0.56 to 0.57) and the average RMSE and positive MB decreased by 0.003 m·s$^{-1}$ and 0.051 m·s$^{-1}$, respectively (Fig. 3h). The collected SI indicated relatively poor performance of modeled WS10 (most wind speeds were overestimated) compared to T2 and humidity, but turning on ARI in coupled models could improve WS10 simulations somewhat.

Besides the SI discussed above, very limited papers reported the normalized mean error (NME) (%) of surface meteorological variables (T2, SH2, RH2 and WS10) simulated by two-way coupled models (WRF-Chem and WRF-CMAQ) in Asia, which is summarized in Appendix Table B7. The evaluations with two-way coupled models in Asia showed that the overall mean percent errors of T2, SH2, RH2 and WS10 were 22.71%, 10.32%, 13.94%, and 51.28%, respectively. The ranges of NME (%) values were quite wide for T2 (from -0.48 to 270.20 %) and WS10 (from 0.33 to 112.28%) reported by the limited studies. Note that no NME of surface meteorological variables simulated by two-way coupled models simultaneously with and without enabling the ARI effects was mentioned in these studies.

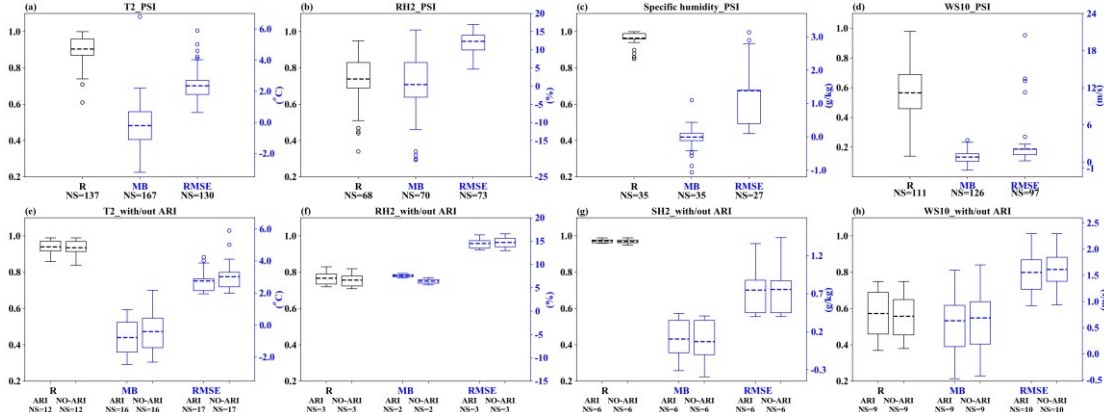

Figure 3. Quantile distributions of R, MB and RMSE for simulated surface meteorological variables by the five
coupled models (WRF-Chem, WRF-CMAQ, GRAPES-CUACE, WRF-NAQPMS and GATOR-GCMOM) (a-d) and
comparisons of statistical indices with/out ARI (e-h) in Asia.

### 5.1.2 Comparisons of SI for meteorology using different coupled models

Also, to examine how different coupled models (i.e., WRF-Chem, WRF-CMAQ, WRF-NAQPMS, GRAPES-CUACE and GATOR-GCMOM) performed in Asia with respect to meteorological variables, the SI were extracted from PSI in term of these five coupled models and displayed in Fig. 4. The SI for T2, RH2, SH2, and WS10 from WRF-NAQPMS, GRAPES-CUACE and GATOR-GCMOM simulations were missing or with rather limited samples so that the discussions here only focused on the WRF-Chem and WRF-CMAQ simulations. Moreover, the SI sample size from studies involving WRF-Chem was generally larger than that involving WRF-CMAQ, except for SH2.

As seen in Fig. 4a, the modeled T2 by both WRF-CMAQ and WRF-Chem was well correlated with observations but WRF-CMAQ (mean R = 0.95) outperformed WRF-Chem (mean R = 0.90) to some extent. On the other hand, WRF-CMAQ underestimated T2 (mean MB = -1.39 ℃) but WRF-Chem slightly overestimated it (mean MB = 0.09 ℃) (Fig. 4e). The RMSE of modeled T2 by both models was at the similar level with mean RMSE values of 2.51 ℃ and 2.31 ℃ by WRF-CMAQ and WRF-Chem simulations, respectively (Fig. 4i).

Both WRF-Chem and WRF-CMAQ performed better for SH2 (mean R = 0.96 and 0.97, respectively) than RH2 (mean R = 0.75 and 0.73, respectively) (Figures 4b and 4c), which might be due to the influence of temperature on RH2 (Bei et al., 2017). Also the modeled RH2 (SH2) by WRF-Chem correlated better (worsen) with observations than those by WRF-CMAQ. The mean RMSE of modeled RH2 (Fig. 4j) by WRF-Chem (11.1 %) was lower than that by WRF-CMAQ (14.3%) but the mean RMSE of modeled SH2 (Fig. 4k) by WRF-Chem (2.25 g·kg$^{-1}$) higher than that by WRF-CMAQ (0.71 g·kg$^{-1}$). It was seen in Figures 4f and 4d that WRF-CMAQ overestimated RH2 and SH2 (average MB were 5.30 % and 0.07 g·kg$^{-1}$, respectively), and WRF-Chem underpredicted RH2 (average MB = -0.32 %) and SH2 (average MB = -0.06 g·kg$^{-1}$). Generally, the modeled RH2 and SH2 were reproduced more reasonably by WRF-Chem than those by WRF-CMAQ.

The modeled WS10 by both WRF-Chem and WRF-CMAQ (Fig. 4d) correlated with observations on the same level with the mean R of 0.56. The mean RMSE of modeled WS10 by WRF-Chem and WRF-CMAQ were 1.54 m·s$^{-1}$ and 2.28 m·s$^{-1}$, respectively, as depicted in Fig. 4l. Both models overpredicted WS10 to some extend with average MBs of 0.55 m·s$^{-1}$ (WRF-CAMQ) and 0.84 m·s$^{-1}$ (WRF-Chem), respectively. These results demonstrated that overall WRF-CMAQ and WRF-Chem had similar model performance of WS10.

In general, WRF-CMAQ performed better than WRF-Chem for T2 but worse for humidity (RH2 and SH2), and both models' performance for WS10 was very similar. WRF-Chem overestimated T2, RH2 and WS10 and underestimated SH2 slightly, while WRF-CMAQ overpredicted humidity and WS10 but underpredicted T2. Compared to WRF-Chem and WRF-CMAQ, the very few SI samples indicated that for the meteorological variables excluding SH2, WRF-NAQPMS simulations matched with observations better than GRAPES-CUACE simulations but more applications and statistical analysis of these two models are needed to make this kind of comparison conclusive.

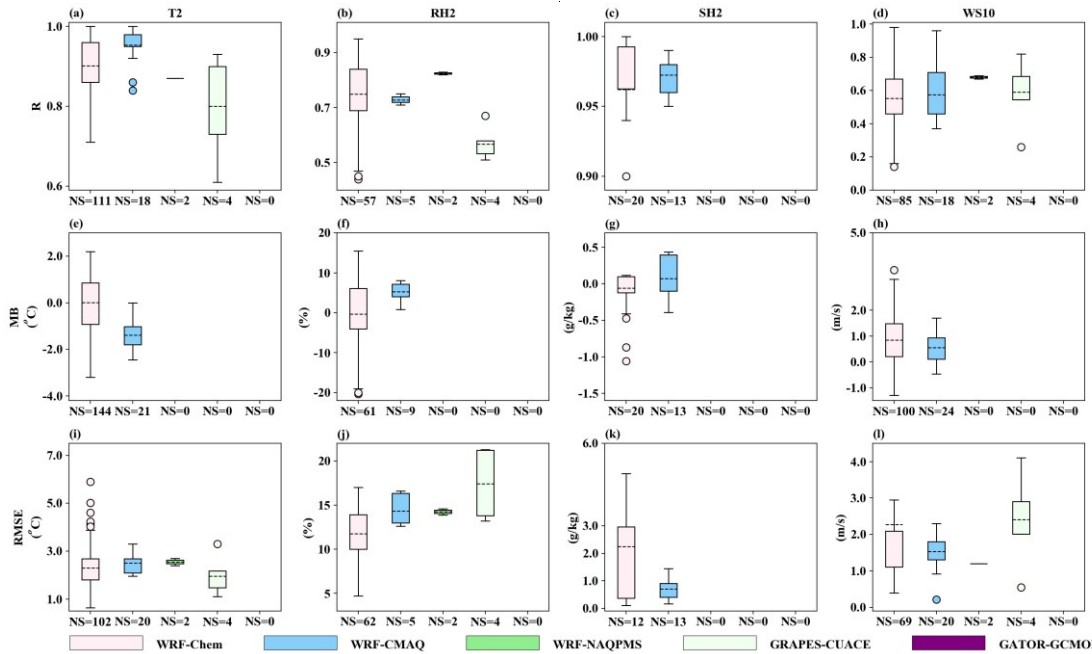

Figure 4. Quantile distributions of the statistical indices for simulated surface meteorological variables by WRF-
Chem, WRF-CMAQ, GRAPES-CUACE, WRF-NAQPMS and GATOR-GCMOM in Asia.

**5.2 Model performance for air quality variables**
**5.2.1 Overall performance**
The results of the overall statistical evaluation for the online air quality simulations are
presented in Figure 5, and all labels and colors indicating SI were the same as those for
meteorological variables. In this figure and following figures, NP and NS are number of publications
and samples with SI, respectively and summed up in Appendix Table B3. In Fig. 5a, the correlation
between the simulated and observed $PM_{2.5}$ concentrations from PSI showed that in Asia coupled
models performed relatively well for $PM_{2.5}$ (mean R = 0.63), but RMSE was between -87.60 and
80.90 and more than half of samples of simulated $PM_{2.5}$ were underestimated (mean MB = -2.08
$\mu g \cdot m^{-3}$). Note that NS in Fig. 5c-d and Appendix Table B5 counted the samples of SI provided by
the simulations simultaneously with and without ARI. With the ARI turned on in the coupled models,
modeled $PM_{2.5}$ concentrations (limited papers with 15 samples) were improved somewhat and the
mean R slightly increased from 0.71 to 0.72 and mean absolute MB decreased from 4.10 to 1.33
$\mu g \cdot m^{-3}$ (Fig. 5c), but RMSE of $PM_{2.5}$ concentrations slightly increased from 35.40 to 36.20 $\mu g \cdot m^{-3}$.
In short, $PM_{2.5}$ with/without ARI agreed well with observations but were mostly underestimated,
and $PM_{2.5}$ bias simulated by models became overpredicted.
Compared with $PM_{2.5}$, mean R (0.59) of $O_3$ was relatively smaller (Fig. 5b). The statistical
analysis also showed the most modeled $O_3$ concentrations tended to be overestimated (76 % of the
samples) with the average MB value of 8.05 $\mu g \cdot m^{-3}$, and the mean RMSE value was 32.65 $\mu g \cdot m^{-3}$.
The 14 PSI with ARI effects suggested that the correlation of $O_3$ was slightly improved (mean R
from 0.58 to 0.64) and the average RMSE and MB were decreased by 15.93 $\mu g \cdot m^{-3}$ and 1.55 $\mu g \cdot m^{-3}$
1030, respectively (Fig. 5d). The collected studies indicated relatively poor performance of modeled $O_3$
compared to $PM_{2.5}$, but turning on ARI in coupled models improved $O_3$ simulations somewhat.
In addition to the SI analyzed above and similar to the surface meteorological variables, the
NME (%) of $PM_{2.5}$ and $O_3$ is listed in Table B7. The limited studies with WRF-Chem and WRF-
CMAQ indicated that the overall mean percent errors of $PM_{2.5}$ and $O_3$ were 47.63% (from 29.55 to
104.70 %) and 43.03% (from 21.10 to 127.00 %), respectively. With the ARI effects enabled in
WRF-Chem in different seasons over the China domain, the NME (%) of $PM_{2.5}$ increased slightly
during most seasons, except during a spring month with little change (Zhang et al., 2018). Another
study by Nguyen et al. (2019b) revealed that the NME (%) of $PM_{2.5}$ and $O_3$ simulated by WRF-
CMAQ became a little worse in SEA comparing to the simulations without ARI.

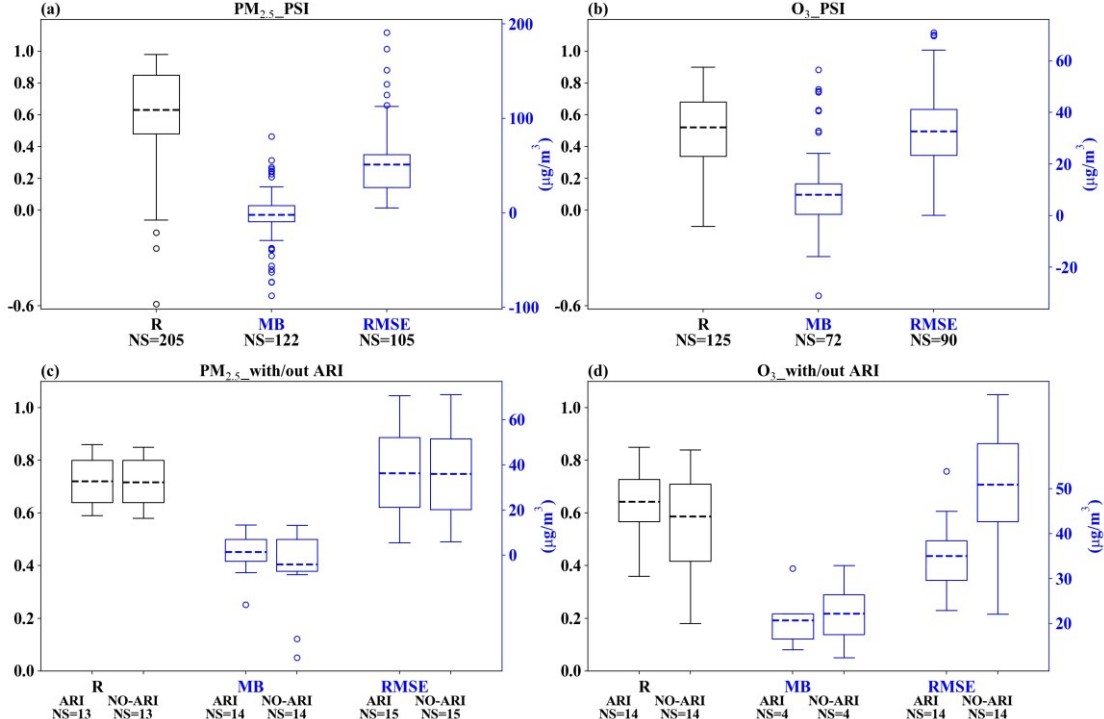


Figure 5. Quantile distributions of statistical indices for simulated PM$_{2.5}$ and O$_3$ (a-b) by the five two-way coupled
models (WRF-Chem, WRF-CMAQ, GRAPES-CUACE, WRF-NAQPMS and GATOR-GCMOM) and comparisons
of statistical indices with/out ARI (c-d) in Asia.

**5.2.2 Comparisons of SI for air quality using different coupled models**
Figure 6 showed the SI for PM$_{2.5}$ and O$_3$ from different coupled models, and only WRF-Chem
and WRF-CMAQ simulations were discussed for the same reason as in Section 5.1.2. The modeled
PM$_{2.5}$ by WRF-CMAQ (mean R = 0.69) outperformed WRF-Chem (mean R = 0.62) to some extent
(Fig. 6a) and the RMSE of modeled PM$_{2.5}$ by WRF-CMAQ (33.24 μg·m$^{-3}$) was smaller than that by
WRF-Chem (56.16 μg·m$^{-3}$). With respect to MB, WRF-CMAQ overestimated PM$_{2.5}$ (mean MB =
+1.60 μg·m$^{-3}$) but WRF-Chem slightly underestimated it (mean R = -3.12 μg·m$^{-3}$) (Fig. 6c). Figure
6b showed that the modeled O$_3$ by WRF-CMAQ (0.60) correlated better with observations than
those by WRF-Chem (0.47), but the mean RMSE of modeled O$_3$ (Fig. 6f) by WRF-Chem (27.13
μg·m$^{-3}$) was lower than that by WRF-CMAQ (35.19 μg·m$^{-3}$). It was seen in Figures 6d that both
WRF-CMAQ and WRF-Chem overestimated O$_3$, with mean MBs as 11.98 and 7.21 μg·m$^{-3}$,
respectively. Generally, the modeled PM$_{2.5}$ and O$_3$ were reproduced more reasonably by WRF-
CMAQ than by WRF-Chem, even though there were much more samples available from WRF-
Chem simulations than WRF-CMAQ simulations.

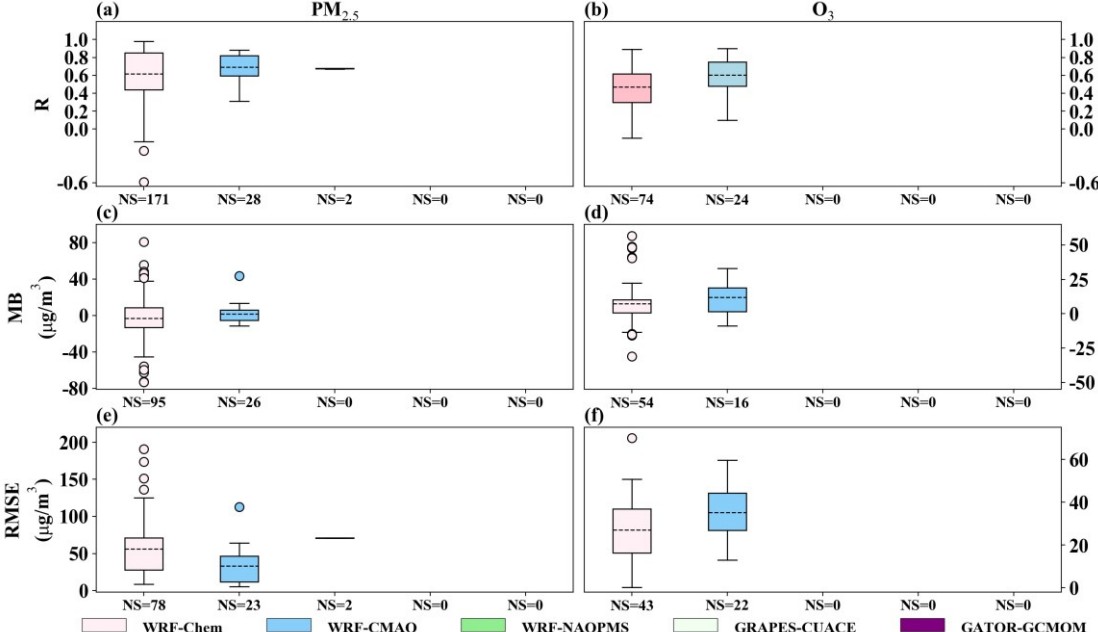

Figure 6. Quantile distributions of R, MB and RMSE of $PM_{2.5}$ and $O_3$ simulated by WRF-Chem, WRF-CMAQ, GRAPES-CUACE, WRF-NAQPMS and GATOR-GCMOM in Asia.

## 6 Impacts of aerosol feedbacks in Asia

Aerosol feedbacks not only impact the performances of two-way coupled models but also the simulated meteorological and air quality variables to a certain extent. In this section, we collected and quantified the variations (Table S3) of these variables induced by ARI or/and ACI from the modeling studies in Asia. Due to limited sample sizes in the collected papers, the target variables only include radiative forcing, surface meteorological parameters (T2, RH2, SH2 and WS10), PBLH, cloud, precipitation, and $PM_{2.5}$ and gaseous pollutants.

### 6.1 Impacts of aerosol feedbacks on meteorology
### 6.1.1 Radiative forcing

With regard to radiative forcing, most studies with two-way coupled models in Asia had focused on the effects of dust aerosols (Dust), BC emitted from ARB (ARB_BC) and anthropogenic sources (Anthro_BC), and total anthropogenic aerosols (Anthro). Figure 7 presents the variations of simulated SWRF and LWRF at the bottom (BOT) and TOA and in the ATM due to aerosol feedbacks, and detailed information of these variations are compiled in Table S5. In this figure, the color bars show the range of radiative forcing variations and the black tick marks inside the color bars represent these variations extracted from all the collected papers. It should be noted that in this figure all the radiative forcing variations were plotted regardless of temporal resolutions of data reporting and simulation durations. Apparently in Asia, most studies targeted the SWRF variations induced by anthropogenic aerosols at the BOT that exhibited the largest differences ranging from -140.00 to -0.45 W·m$^{-2}$, with the most variations (88 % of samples) concentrated in the range of -50.00 to -0.45 W·m$^{-2}$. The SWRF variations due to anthropogenic aerosols in the ATM and at the TOA were -2.00 to +120.00 W·m$^{-2}$ and -6.50 to 20.00 W·m$^{-2}$, respectively. There were much less studies reported LWRF variations caused by anthropogenic aerosols, which ranged from -10.00 to +5.78 W·m$^{-2}$, -1.91 to +3.94 W·m$^{-2}$, and -4.26 to +1.21 W·m$^{-2}$ at the BOT and TOA, and in the ATM, respectively.

Considering BC from anthropogenic sources and ARB, they both led to positive SWRF at the TOA (with mean values of 2.69 and 7.55 W·m$^{-2}$, respectively) and in the ATM (with mean values of 11.70 and 25.45 W·m$^{-2}$, respectively) but negative SWRF at the BOT (with mean values of -18.43 and -14.39 W·m$^{-2}$, respectively). The responses of LWRF to Anthro_BC and ARB_BC at the BOT (in the ATM) on average were 4.01 and 0.72 W·m$^{-2}$ (-1.89 and -3.24 W·m$^{-2}$), respectively, and weak at the TOA (+0.92 and -0.53 W·m$^{-2}$, respectively). The SWRF variations induced by dust were in the range of -233.00 to -1.94 W·m$^{-2}$ and -140.00 to +25.70 W·m$^{-2}$, and +1.44 to +164.80 W·m$^{-2}$ at the BOT and TOA, and in the ATM, respectively. The LWRF variations caused by dust were the

largest (with mean values of 22.83 W·m⁻² and +5.20 W·m⁻², and -22.12 W·m⁻² at the BOT and TOA,
and in the ATM, respectively), comparing to the ones caused by anthropogenic aerosols and BC
aerosols from anthropogenic sources and ARB.

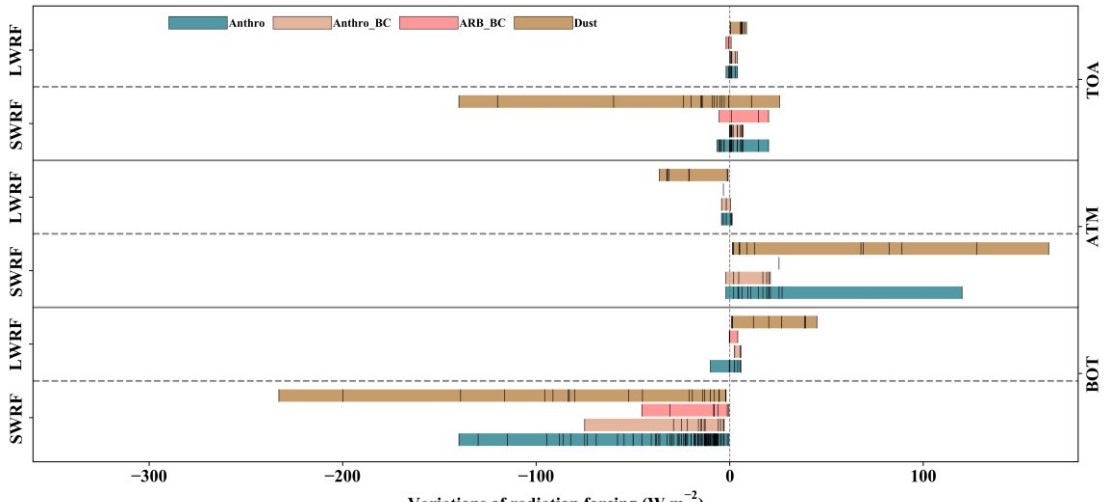

Figure 7. Variations of shortwave and longwave radiative forcing (SWRF and LWRF) simulated by two-way
coupled models (WRF-Chem, WRF-CMAQ, GRAPES-CUACE, WRF-NAQPMS and GATOR-GCMOM) with
aerosol feedbacks at the bottom and top of atmosphere (BOT and TOA), and in the atmosphere (ATM) in Asia.

As shown in Fig. 7, SWRF variations at the BOT caused by total aerosols (sum of Anthro,
Anthro_BC, ARB_BC and Dust) had been widely assessed in Asia. Therefore, we further analyzed
their spatiotemporal distributions and inter-regional differences, which are displayed in Fig. 8.
Figure 8a presents the SWRF variations over different areas of Asia (the acronyms used in Fig. 8
are listed in Appendix Table B1) at different time scales. In Asia, almost 41 % of the selected papers
investigated SWRF towards its monthly variations, 36 % towards its hourly and daily variations,
and 23 % towards its seasonal and yearly variations. Most studies reported aerosol-induced SWRF
variations were primarily conducted in NCP, EA, China, and India. At the hourly scale, the range of
SWRF decreases was from -350.00 to -5.90 W·m⁻² (mean value of -106.92 W·m⁻²) during typical
pollution episodes, and significant variations occurred in EA. The daily and monthly mean SWRF
reductions varied from -73.71 to -5.58 W·m⁻² and -82.20 to -0.45 W·m⁻², respectively, with relative
large perturbations in NCP. At the seasonal and yearly scales, the SWRF changes ranged from -
22.54 to -3.30 W·m⁻² and -30.00 to -2.90 W·m⁻² with mean value of -11.28 and -11.82 W·m⁻²,
respectively, with EA as the most researched area.
To identify the differences of aerosol-induced SWRF variations between high- (Asia) and low-
polluted regions (Europe and North America), their inter-regional comparisons are depicted in Fig.
8b. This figure does not include information about temporal resolutions of data reporting and
durations of model simulations with ARI or/and ACI, but intends to delineate the range of SWRF
changes due to aerosol feedbacks. The SWRF variations fluctuated from -233.00 to -0.45 W·m⁻², -
100.00 to -1.00 W·m⁻², and -600.00 to -1.00 W·m⁻² in Asia, Europe, and North America, respectively.
It should be pointed out that the two extreme values were caused by dust (-233.00 W·m⁻²) in Asia
and wildfire (-600.00 W·m⁻²) in North America. Overall, the median value of SWRF reductions due
to ARI or/and ACI in Asia (-15.92 W·m⁻²) was larger than those in North America (-10.50 W·m⁻²)
and Europe (-7.00 W·m⁻²).
.

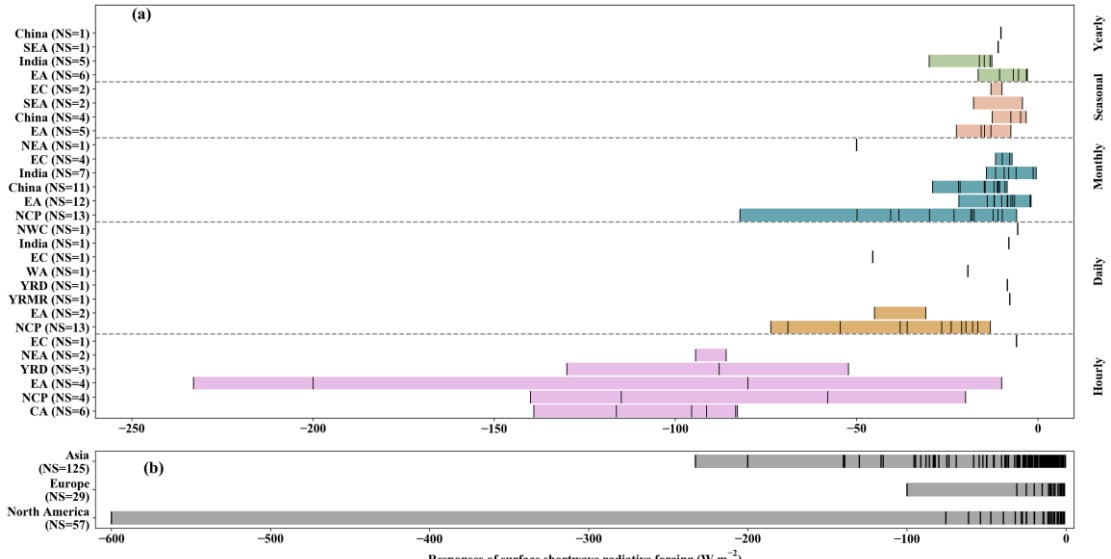

Figure 8. Responses of shortwave radiation forcing to aerosol feedbacks in different areas/periods in Asia (a) and
the inter-regional comparisons of its variations in Asia, Europe and North America (b).

### 6.1.2 Temperature, wind speed, humidity and PBLH

The impact of aerosols on radiation can influence energy balance, which eventually alter other
meteorological variables. The summary of aerosol-induced variations of T2, WS10, RH2, SH2 and
PBLH in different regions of Asia as well as at different temporal scales are provided in Table 6. In
this table, the minimum and maximum values were collected from the corresponding papers and the
mean values were calculated with adding all the variations from these papers and then divided by
the number of samples.
Overall, aerosol effects led to decreases of T2, WS10 and PBLH with average changes of -
0.65 °C, -0.13 m·s⁻¹ and -60.70 m, respectively, and increases of humidity (mean ΔRH2 = 2.56 %)
in most regions of Asia. On average, the hourly aerosol-induced changes of surface meteorological
variables (T2, WS10 and RH2) and PBLH were the largest among the different time scales. At the
hourly time scale, the mean variations of T2, WS10, RH2 and PBLH due to ARI or/and ACI were -
1.85 °C, -0.32 m·s⁻¹, 4.60 % and -165.84 m, respectively, and their absolute maximum values in EC,
YRD, NCP and NCP, respectively. Compared to variations at the hourly time scale, smaller daily
variations of T2, WS10, RH2 and PBLH were caused by aerosol effects, and their mean values were
-0.63 °C, -0.15 m·s⁻¹, +2.89 % and -34.61 m, respectively. The largest daily variations of T2, WS10,
RH2 and PBLH occurred in NCP, EC, EC and SEC, respectively. For other time scales (monthly,
seasonal and yearly), the respective mean variations of T2, RH2 and PBLH induced by aerosol
effects were comparable. However, the WS10 perturbations at the monthly time scale were about
two to three times higher than those at the seasonal and yearly time scales. High variations at the
monthly, seasonal and yearly time scales were reported in NCP (T2, RH2 and PBLH), EA (T2,
WS10 and PBLH) and PRD (T2 and PBLH), respectively. In addition, comparing to T2 and PBLH,
the aerosol-induced variations of WS10 and humidity were less revealed.
Table 6. Summary of variations of surface meteorological variables and planetary boundary layer height (PBLH)
caused by aerosol feedbacks simulated by two-way coupled models (WRF-Chem, WRF-CMAQ, GRAPES-CUACE,
WRF-NAQPMS and GATOR-GCMOM) in different regions of Asia and at different temporal scales.

| Region | Time scale | ΔT2 [mean] (°C) | ΔWS10 [mean] (m·s⁻¹) | ΔRH2/SH2 [mean] | ΔPBLH [mean] (m) |
|---|---|---|---|---|---|
| EC | hours | -8.00 to -0.20 [-2.68] | | | -300.00 to -50.00 [-175.00] |
| EA | hours | -3.00 to -2.00 [-2.50] | | | |
| YRD | hours | -1.40 to -1.00 [-1.15] | -0.80 to -0.10 [-0.41] | | -276.00 to -29.90 [-105.42] |
| NCP | hours | -2.80 to -0.20 [-1.05] | -0.30 to -0.10 [-0.23] | 1.00 % to 12.00 % [4.60 %] | -287.20 to -147.00 [-217.10] |
| Hourly mean | | -1.85 | -0.32 | 4.60% | -165.84 |
| NCP | days | -2.00 to -0.10 [-0.88] | -0.4 to -0.01 [-0.17] | 0.51 % to 4.10 % [2.52 %] | -111.40 to -10.00 [-49.07] |

| Region | Time | Column 1 | Column 2 | Column 3 | Column 4 |
|---|---|---|---|---|---|
| EC | days | -0.94 to -0.65 [-0.79] | -0.52 to -0.37 [-0.45] | 1.92 % to 9.75 % [5.84 %] | |
| India | days | -1.60 to 0.10 [-0.75] | | | |
| SEC | days | -1.38 to -0.18 [-0.70] | -0.07 to 0.05 [-0.023] | -0.37 % to 6.57 % [2.63 %] | -84.1 to -27.55 [-53.62] |
| NEA | days | -0.52 | -0.08 | | -46.39 |
| MRYR | days | -0.16 | -0.01 | 0.56 % | -16.46 |
| India | days | | | | -6.90 |
| **Daily mean** | | **-0.63** | **-0.15** | **2.89 %** | **-34.61** |
| India | months | -0.45 | | | |
| NCP | months | -1.30 to -0.06 [-0.43] | | 1.30 % to 4.70 % [2.53 %] | -109.00 to -5.48 [-36.01] |
| NEA | months | -0.30 | -0.10 | | -50.00 |
| PRD | months | -0.60 to 0.13 [-0.16] | | | |
| EA | months | -0.45 to -0.03 [-0.13] | | | -35.70 to -13.00 [-24.35] |
| China | months | -0.89 to 0.60 [-0.12] | | | -66.60 to -2.30 [-25.67] |
| EC | months | -0.30 to -0.05 [-0.11] | | | -13.10 to -6.20 [-9.65] |
| **Monthly mean** | | **-0.24** | **-0.10** | **2.53 %** | **-29.13** |
| EA | seasons | -0.58 to -0.30 [-0.40] | -0.05 to -0.02 [-0.035] | | -64.62 to -30.70 [-43.27] |
| SEA | seasons | -0.39 to -0.03 [-0.21] | -0.06 to -0.01 [-0.035] | | -48.33 to -6.71 [-27.52] |
| **Seasonal mean** | | **-0.31** | **-0.035** | | **-34.61** |
| PRD | years | -0.27 | | | -45.00 |
| TP | years | -0.24 | | | |
| SEA | years | -0.21 | -0.03 | | -27.25 |
| EA | years | | -0.03 | 0.13 g·kg$^{-1}$ | -46.47 to -45.00 [-45.74] |
| EC | years | | -0.014 | 0.21 % | |
| **Yearly mean** | | **-0.24** | **-0.025** | **0.21 %** | **-39.33** |

### 6.1.3 Cloud and precipitation

In the included publications, only a few papers focusing on the effects of aerosol feedbacks on cloud properties (cloud fraction, LWP, ice water path (IWP), CDNC and cloud effective radius) and precipitation characteristics (amount, spatial distribution, peak occurrence and onset time) using two-way coupled models in Asia, as shown in Table 7. In this table, the abbreviations representing aerosol emission sources (Dust, ARB_BC, Anthro_BC, and Anthro) and regions in Asia are defined in Appendix Table B1. The plus and minus signs indicate increase and decrease, respectively.

The variations of cloud properties and precipitation characteristics induced by ARI or/and ACI are rather complex and not uniform in different parts of Asia and time periods. BC from both ARB and anthropogenic sources reduced cloud fraction through ARI and both ARI and ACI in several areas in China. ARI or/and ACI induced by anthropogenic aerosols could increase or decrease cloud fraction and affect cloud fraction differently in various atmospheric layers and time periods. Considering EA and subareas in China, anthropogenic aerosols tended to increase LWP through ARI and ACI as well as ACI alone but decrease LWP in some areas of SC (ARI and ACI) at noon and in afternoon during summertime and NC (ACI) in winter. ARI and ACI induced by anthropogenic BC aerosols had negative effects on LWP except at daytime in CC. Dust aerosols increased both LWP and IWP through ACI in EA, which was reported only by one study. The increase (decrease) of CDNC caused by the ARI and ACI effects of anthropogenic (anthropogenic BC) aerosols in EC during summertime was reported. Through ACI, anthropogenic aerosols affected CDNC positively in EA and China. Compared to anthropogenic aerosols, dust aerosols could have much larger positive impacts on CDNC via ACI in springtime over EA. The ACI effects of anthropogenic aerosols reduced cloud effective radius over China (January) and EA (July).

Among all the variables describing cloud properties and precipitation characteristics, the variations of precipitation amount were studied the most using two-way coupled models in Asia. How turning on ARI or/and ACI in coupled models can change precipitation amount is not unidirectional and depends on many factors, including different aerosol sources, areas, emission levels, atmospheric humidity, precipitation types, seasons, and time of a day. Under the high

emission levels as well as at slightly different humidity levels of RH > 85 % with increasing emissions, the ACI effects of anthropogenic aerosols induced precipitation increase in the MRYR area of China. Over the same area, precipitation decreased due to the ACI effects of anthropogenic aerosols with the low emission levels and RH < 80 %. In PRD, wintertime precipitation was enhanced by the ACI effects of anthropogenic aerosols but inhibited by ARI. In SK, summertime precipitation was both enhanced and inhibited by the ACI and ARI effects of anthropogenic aerosols. In locations upwind (downwind) of Beijing, rainfall amount was raised (lowered) by the ARI effects of anthropogenic aerosols but lowered (raised) by ACI. Both ARI and ACI induced by anthropogenic aerosols had positive impacts on total, convective, and stratiform rain in India during the summer season and the increase of convective rain was larger than those of stratiform. Summertime precipitation amounts could be enhanced or inhibited at various subareas inside simulation domains over India, China, and Korea and during day- or night-time due to ARI and ACI of anthropogenic aerosols. Over China, dust-induced ACI decreased (increased) springtime precipitation in CC (western part of NC), and over India, dust aerosols from local sources and ME had positive impacts on total, convective, and stratiform rain through ARI and ACI. Simulations in India also revealed that precipitation could be increased in some subareas but decreased in another and absorptive (non-absorptive) dust enhanced (inhibited) summertime precipitation via ARI and ACI. The ARI (ACI) effects of BC from ARB caused precipitation reduction (increase) in SEC but CAs emitted from ARB (ARB_CAs) caused rainfall enhancement in Myanmmar. During pre-monsoon (monsoon) season, ARI induced by anthropogenic BC could lead to +42 % (-5 to -8 %) variations of precipitation in NEI (SI). Considering both ARI and ACI effects, BC from ARB and sea salt aerosols enhanced or inhibited precipitation in different parts of India and BC from anthropogenic sources enhanced (inhibited) nighttime (daytime) rainfall in CC (NC and SC) at the rate of +1 to +4 mm·day$^{-1}$ (-2 to -6 mm·day$^{-1}$) during summer season. With respect to spatial variations, 6.5 % larger rainfall area in PRD was caused by ARI and ACI effects under 50 % reduced anthropogenic emissions. ACI induced by anthropogenic aerosols tended to delay the peak occurrence time and onset time of precipitation by one to nine hours in China and South Korea.

Table 7. Summary of changes of cloud properties and precipitation characteristics due to aerosol feedbacks simulated by two-way coupled models (WRF-Chem, WRF-CMAQ, GRAPES-CUACE, WRF-NAQPMS and GATOR-GCMOM) in Asia.

| Variables | | Variations (aerosol effects) | Simulation time period | Regions | References |
|---|---|---|---|---|---|
| Cloud properties | Cloud fraction | -7 % low-level cloud (ARB_BC ARI) | Apr., 2013 | SEC | Huang et al., 2019 |
| | | +0.03 to +0.08 below 850 hPa and at 750 hPa (Anthro ARI & ACI), esp. at early morning and nighttime | Aug., 2008 | EC | Gao and Zhang, 2018 |
| | | Max -0.06 between 750 hPa and 850 hPa (Anthro ARI & ACI), esp. in afternoon and evening | Aug., 2008 | CC | Gao and Zhang, 2018 |
| | | -0.02 to -0.06 below 750 hPa (Anthro_BC ARI & ACI), esp. in afternoon | Aug., 2008 | SC & NC | Gao and Zhang, 2018 |
| | | -0.04 to -0.06 between 750 hPa and 850 hPa (Anthro_BC ARI & ACI), esp. in afternoon | Aug., 2008 | CC | Gao and Zhang, 2018 |
| | | -6.7 % to +3.8 % (Anthro ARI) | Jun. 6-9 & Jun. 11-14, 2015 | SK | Park et al., 2018 |
| | | +22.7 % (Anthro ACI) | Jun. 6-9 & Jun. 11-14, 2015 | SK | Park et al., 2018 |
| | | -0.03 % low-, -0.54 % middle- and -0.58 % high-level cloud (Anthro ACI) | 2008 to 2012 | PRD | Liu et al., 2018c |
| | LWP | +5 to +50 g·m$^{-2}$ (Anthro ARI & ACI) | Aug., 2008 | EC | Gao and Zhang, 2018 |
| | | +10 to +20 g·m$^{-2}$ (Anthro_BC ARI & ACI) at daytime | Aug., 2008 | CC | Gao and Zhang, 2018 |
| | | -5 to -40 g·m$^{-2}$ (Anthro ARI & ACI) at noon and in afternoon | Aug., 2008 | Part of SC | Gao and Zhang, 2018 |
| | | -2 to -20 g·m$^{-2}$ (Anthro_BC ARI & ACI) | Aug., 2008 | SC | Gao and Zhang, 2018 |
| | | -2 to -30 g·m$^{-2}$ (Anthro_BC ARI & ACI) | Aug., 2008 | NC | Gao and Zhang, 2018 |
| | | Max+18 g·m$^{-2}$ (Dust ACI) | Mar.-May., 2010 | EA | Wang et al., 2018b |
| | | +40 to +60 g·m$^{-2}$ (Anthro ACI) | Jan., 2008 | SC | Gao et al., 2012 |
| | | +40 g·m$^{-2}$ (Anthro ACI) | Jan., 2008 | CC | Gao et al., 2012 |
| | | Less than +5 g·m-2 or -5 g·m$^{-2}$ (Anthro ACI) | Jan., 2008 | NC | Gao et al., 2012 |
| | | +30 to +50 g·m$^{-2}$ (Anthro ACI) | Jul., 2008 | EA | Gao et al., 2012 |
| | IWP | +5 to +10 g·m$^{-2}$ (Dust ACI) | Mar. 17-Apr. 30, 2012 | EA | Su and Fung, 2018b |
| | CDNC | +20 to +160 cm$^{-3}$ (Anthro ARI & ACI) | Aug., 2008 | EC | Gao and Zhang, 2018 |

| | | | | | |
|---|---|---|---|---|---|
| | | -5 to -60 cm⁻³ (Anthro_BC ARI & ACI) | Aug., 2008 | EC | Gao and Zhang, 2018 |
| | | Max +10500 cm-3 (Dust ACI) | Mar.-May., 2010 | EA | Wang et al., 2018b |
| | | +650 cm-3 (Anthro ACI) | Jan., 2008 | EC | Gao et al., 2012 |
| | | +400 cm-3 (Anthro ACI) | Jan., 2008 | CC & SWC | Gao et al., 2012 |
| | | Less than +200 cm-3 (Anthro ACI) | Jan., 2008 | NC | Gao et al., 2012 |
| | | +250 to +400 cm-3 (Anthro ACI) | Jul., 2008 | EA | Gao et al., 2012 |
| Cloud effective radius | | More than -4 µm (Anthro ACI) | Jan., 2008 | SWC, CC & SEC | Gao et al., 2012 |
| | | More than -2 µm (Anthro ACI) | Jan., 2008 | NC | Gao et al., 2012 |
| | | -3 µm (Anthro ACI) | Jul., 2008 | EA | Gao et al., 2012 |
| | | Enhancement/inhibition of precip. due to high/low Anthro emissions, ACI inhibited (enhanced) precip. at RH < 80 % (> 85 %) with increasing Anthro emissions | Jun. 18-19, 2018 | MRYR | Bai et al., 2020 |
| | | -4.72 mm (Anthro ARI) and +33.7 mm (Anthro ACI) | Dec. 14-16, 2013 | PRD | Liu Z. et al., 2020 |
| | | +2 to +5 % (ARB CAs ARI) | Mar.-Apr., | Myanmar | Singh et al., 2020 |
| | | -1.09 mm·day⁻¹ (ARB_BC ARI) | Apr., 2013 | SEC | Huang et al., 2019 |
| | | +0.49 mm·day⁻¹ (ARB_BC ACI) | Apr., 2013 | SEC | Huang et al., 2019 |
| | | -0 to -4 mm·day⁻¹ (Anthro ARI & ACI) | Jun.-Sep., 2010 | Indus basin & eastern IGP | Kedia et al., 2019b |
| | | +1 to +3 mm·day⁻¹ non-convective rain (Anthro ARI & ACI) | Jun.-Sep., 2010 | WG of India | Kedia et al., 2019b |
| | | +5 mm·day⁻¹ non-convective rain (Anthro ARI & ACI) | Jun.-Sep., 2010 | NEI | Kedia et al., 2019b |
| | | Increase of total rain (Dust ARI & ACI) | Jun.-Sep., 2010 | NI, CI, WG, NEI & central IGP | Kedia et al., 2019b |
| | | Decrease of total rain (Dust ARI & ACI) | Jun.-Sep., 2010 | NWI & SPI | Kedia et al., 2019b |
| | | Decrease of total rain (ARB_BC ARI & ACI) | Jun.-Sep., 2010 | WG, SPI, NWI, EI & NEI | Kedia et al., 2019b |
| | | Increase of total rain (ARB_BC ARI & ACI) | Jun.-Sep., 2010 | CI, Central IGP & EPI | Kedia et al., 2019b |
| | | Decrease of total rain (Sea salt ARI & ACI) | Jun.-Sep., 2010 | EPI, WPI, CPI & SPI | Kedia et al., 2019b |
| | | Increase of total rain (Sea salt ARI & ACI) | Jun.-Sep., 2010 | NCI & central IGP | Kedia et al., 2019b |
| | | -20 to -200mm (Anthro ARI & ACI) | Aug., 2008 | SC & NC | Gao and Zhang, 2018 |
| Precipitation (precip.) | Amount | +20 to +100 mm (Anthro_BC ARI & ACI) | Aug., 2008 | CC | Gao and Zhang, 2018 |
| | | +1 to +4 mm·day⁻¹ nighttime precip. (ARI & ACI of Anthro or Anthro_BC) | Aug., 2008 | CC | Gao and Zhang, 2018 |
| | | -2 to -6 mm·day⁻¹ daytime precip. (ARI & ACI of Anthro or Anthro_BC) | Aug., 2008 | NC | Gao and Zhang, 2018 |
| | | -2 to -4 mm·day⁻¹ daytime precip. (Anthro ARI & ACI) | Aug., 2008 | SC | Gao and Zhang, 2018 |
| | | -2 to -6 mm·day⁻¹ daytime precip. (Anthro_BC ARI & ACI) | Aug., 2008 | SC | Gao and Zhang, 2018 |
| | | -54.6 to +24.1 mm (Anthro ARI) | Jun. 6-9, 2015 | SK | Park et al., 2018 |
| | | -23.8 to +24.0 mm (Anthro ARI) | Jun. 6-9, 2015 | SK | Park et al., 2018 |
| | | -63.2 to +27.1 mm (Anthro ARI & ACI) | Jun. 6-9, 2015 | SK | Park et al., 2018 |
| | | Min -7.0 mm (Anthro ARI) | Jun. 11-14, 2015 | SK | Park et al., 2018 |
| | | Min -36.6 mm (Anthro ACI) | Jun. 11-14, 2015 | SK | Park et al., 2018 |
| | | +42 % (Anthro_BC ARI) during pre-monsoon season | Mar.-May., 2010 | NEI | Soni et al., 2018 |
| | | -5 to -8 % (Anthro_BC ARI) during monsoon season | Jun.-Sep., 2010 | SI | Soni et al., 2018 |
| | | +1 mm·day⁻¹ precip. (Dust ACI) | Mar. 17-Apr. 30, 2012 | Western part of NC | Su and Fung, 2018b |
| | | -1 mm·day⁻¹ precip. (Dust ACI) | Mar. 17-Apr. 30, 2012 | CC | Su and Fung, 2018b |
| | | +0.95 mm·day⁻¹ precip. (absorptive Dust ARI & ACI) | Jun.-Aug., 2008 | India | Jin et al., 2016a |
| | | -0.4 mm·day⁻¹ precip. (non-absorptive Dust ARI & ACI) | Jun.-Aug., 2008 | India | Jin et al., 2016a |
| | | +0.44 mm·day⁻¹ total precip. (Dust ARI & ACI over whole study domain) | Jun.-Aug., 2008 | India | Jin et al., 2016b |

| | | | | |
|---|---|---|---|---|
| | +0.34 mm·day$^{-1}$ total precip. (Dust ARI & ACI from ME) | Jun.-Aug., 2008 | India | Jin et al., 2016b |
| | +0.31 mm·day$^{-1}$ total precip. (Anthro ARI & ACI over whole study domain) | Jun.-Aug., 2008 | India | Jin et al., 2016b |
| | +0.32 mm·day$^{-1}$ convective precip. (Dust ARI & ACI over whole study domain) | Jun.-Aug., 2008 | India | Jin et al., 2016b |
| | +0.24 mm·day$^{-1}$ convective precip. (ARI & ACI of Dust from ME) | Jun.-Aug., 2008 | India | Jin et al., 2016b |
| | +0.20 mm·day$^{-1}$ convective precip. (Anthro ARI & ACI over whole study domain) | Jun.-Aug., 2008 | India | Jin et al., 2016b |
| | +0.12 mm·day$^{-1}$ stratiform precip. (Dust ARI & ACI over whole study domain) | Jun.-Aug., 2008 | India | Jin et al., 2016b |
| | +0.10 mm·day$^{-1}$ stratiform precip. (ARI & ACI of Dust from ME) | Jun.-Aug., 2008 | India | Jin et al., 2016b |
| | +0.11 mm·day$^{-1}$ stratiform precip. (Anthro ARI & ACI over whole study domain) | Jun.-Aug., 2008 | India | Jin et al., 2016b |
| | -48.29 %/+24.87 % precip. in downwind/upwind regions (Anthro ARI) | Jun. 27-28, 2008 | Beijing | Zhong et al. 2015 |
| | +33.26 % /-4.64 % precip. in downwind/upwind regions (Anthro ACI) | Jun. 27-28, 2008 | Beijing | Zhong et al. 2015 |
| | +0.44 mm·day$^{-1}$ precip. (Dust ARI & ACI) | Jun. 1-Aug. 31, 2008 | India | Jin et al., 2015 |
| Spatial variation | +6.5 % precip. area (ARI & ACI) with 50% Anthro emissions | Jun. 9-12, 2017 | YRD | Liu C. et al., 2019 |
| Peak occurrence time | 1 to 2h delay (Anthro ACI) | Jun. 18-19, 2018 | MRYR | Bai et al., 2020 |
| | 1h delay (ARI & ACI) with 50% Anthro emissions | Jun. 9-12, 2017 | YRD | Liu et al., 2019 |
| | 9h delay (Anthro ACI) | Jun. 7, 2015 | Gosan, SK | Park et al., 2018 |
| | 4h delay (Anthro ACI) | Jun. 7, 2015 | Jinju, SK | Park et al., 2018 |
| Onset time | 9h delay (Anthro ACI) | Jun. 7, 2015 | Gosan, SK | Park et al., 2018 |
| | 2h delay (Anthro ACI) | Jun. 7, 2015 | Jinju, SK | Park et al., 2018 |

## 6.2 Impacts of aerosol feedbacks on air quality

Aerosol effects not only gave rise to changes in meteorological variables but also air quality. Table 8 (the minimum, maximum and mean values were defined in the same way as in Table 6) summarizes the variations of atmospheric pollutant concentrations induced by aerosol effects in different regions of Asia and at different time scales. In Asia, most modeling studies with coupled models targeted the impacts of aerosol feedbacks on surface PM$_{2.5}$ and O$_3$ concentrations, with only few focusing on other gaseous pollutants.

Simulation results showed that turning on aerosol feedbacks in coupled models generally made PM$_{2.5}$ concentrations increased in different regions of Asia at various time scales, which stemmed from decrease of shortwave radiation, T2, WS10 and PBLH and increase of RH2. Some studies did show negative impacts of aerosol effects on hourly, daily, and seasonal PM$_{2.5}$ at some areas that could be attributed to ACI effects, changes in transport and dispersion patterns, reductions in humidity levels and secondary aerosol formations (Zhang et al., 2015a; Yang et al., 2017; Zhan et al., 2017; Wang et al., 2018b). Similar to the perturbations of surface meteorological variables due to aerosol effects, the hourly PM$_{2.5}$ variations and the range were the largest compared to those at other time scales. The largest PM$_{2.5}$ increases were reported in NCP, SEC, EA, SEA and PRD at the hourly, daily, monthly, seasonal and yearly time scales with average values of 23.48 µg·m$^{-3}$, 14.73 µg·m$^{-3}$, 16.50 µg·m$^{-3}$, 1.12 µg·m$^{-3}$ and 2.90 µg·m$^{-3}$, respectively.

In addition to PM$_{2.5}$, gaseous pollutants (O$_3$, NO$_2$, SO$_2$, CO and NH$_3$) are impacted by ARI or/and ACI effects as well. As shown in Table 8, general reductions of ozone concentrations were reported in Asia across all the modeling domains and time scales based on coupled models' simulations. However, the influences of aerosol feedbacks on atmospheric dynamics and stability, and photochemistry (photolysis rate and ozone formation regimes) could make ozone concentrations increase somewhat in summer months or during wet season (Xing et al., 2017; Jung et al., 2019; Nguyen et al., 2019b). The largest hourly, daily, monthly, seasonal, and annual variations of O$_3$ occurred in YRD (-32.80 µg·m$^{-3}$), EC (-5.97 µg·m$^{-3}$), China (-23.90 µg·m$^{-3}$), EA (-4.48 µg·m$^{-3}$) and EA (-2.76 µg·m$^{-3}$), respectively. Along with reduced O$_3$ due to ARI or/and ACI, NO$_2$ concentrations were enhanced with average changes of +12.30 µg·m$^{-3}$ (YRD) at the hourly scale and +0.66 µg·m$^{-3}$ (EA) at both the seasonal and yearly scales, which could be attributed to slower photochemical reactions, strengthened atmospheric stability and O$_3$ titration (Nguyen et al., 2019b). Regarding other gaseous pollutants, limited studies pointed out daily and annual SO$_2$ concentrations increased

in NEA and EA due to lower PBLH induced by the ARI effects of anthropogenic aerosols (Jung et
al.,2019; Nguyen et al., 2019b). The seasonal $SO_2$ reduction was rather large, which related to higher
PBLH induced by the ACI effects of dust aerosols in the NCP area of EA (Wang et al., 2018b). The
slight increase of seasonal $SO_2$ was reported in the whole domain of EA due to lower PBLH caused
by ARI effects of anthropogenic aerosols (Nguyen et al., 2019b). There was only one study depicted
increased CO ($NH_3$) concentration in EC (NEA) due to both the ARI and ACI (ARI) effects of
anthropogenic aerosols but these results may not be conclusive.
Table 8. Compilation of aerosol-induced variations of $PM_{2.5}$ and gaseous pollutants simulated by two-way
coupled models (WRF-Chem, WRF-CMAQ, GRAPES-CUACE, WRF-NAQPMS and GATOR-GCMOM) in
different regions of Asia and at different temporal scales.

| Region | Time scale | $\Delta PM_{2.5}$ [mean] ($\mu g \cdot m^{-3}$) | $\Delta O_3$ [mean] ($\mu g \cdot m^{-3}$) | $\Delta NO_2$ [mean] ($\mu g \cdot m^{-3}$) | $\Delta SO_2$ [mean] ($\mu g \cdot m^{-3}$) | $\Delta CO$ [mean] ($\mu g \cdot m^{-3}$) | $\Delta NH_3$ [mean] ($\mu g \cdot m^{-3}$) |
|---|---|---|---|---|---|---|---|
| NCP | hours | -3.50 to 90.00 [23.48] | | | | | |
| YRD | hours | 7.00 to 30.50 [15.17] | -32.80 to -0.20 [-11.25] | 12.30 | | | |
| Hourly mean | | 19.32 | -11.25 | 12.30 | | | |
| SEC | days | -1.91 to 32.49 [14.73] | | | | | |
| NCP | days | -5.00 to 56.00 [14.51] | | | | | |
| EC | days | 2.87 to 18.60 [10.74] | -5.97 to -1.45 [-3.71] | | | | |
| NEA | days | 1.75 | | | 0.97 | | 0.11 |
| Daily mean | | 10.43 | -3.71 | | 0.97 | | 0.11 |
| India | months | 3.00 to 30.00 [16.50] | | | | | |
| EC | months | 1.00 to 40.00 [16.33] | -2.40 to -1.00 [-1.70] | | | 4.00 to 6.00 [5.00] | |
| China | months | 1.60 to 33.20 [14.38] | -23.90 to 4.92 [-3.42] | | | | |
| EA | months | 3.60 to 10.20 [5.79] | | | | | |
| Monthly mean | | 13.25 | -2.56 | | | 5.00 | |
| SEA | seasons | 0.15 to 2.09 [1.12] | -1.92 to 0.26 [-0.83] | | | | |
| EA | seasons | -8.00 to 2.70 [-0.14] | -4.48 to -1.00 [-2.99] | 0.43 to 0.88 [0.66] | -4.29 to 0.72 [-0.42] | | |
| Seasonal mean | | 0.49 | -1.91 | 0.66 | -0.42 | | |
| PRD | years | 2.90 | | | | | |
| EA | years | 1.82 | -2.76 | 0.66 | 0.54 | | |
| NCP | years | 0.10 to 5.10 [1.70] | | | | | |
| SEA | years | 1.21 | -0.80 | | | | |
| Yearly mean | | 1.91 | -1.78 | 0.66 | 0.54 | | |


## 7 Conclusions

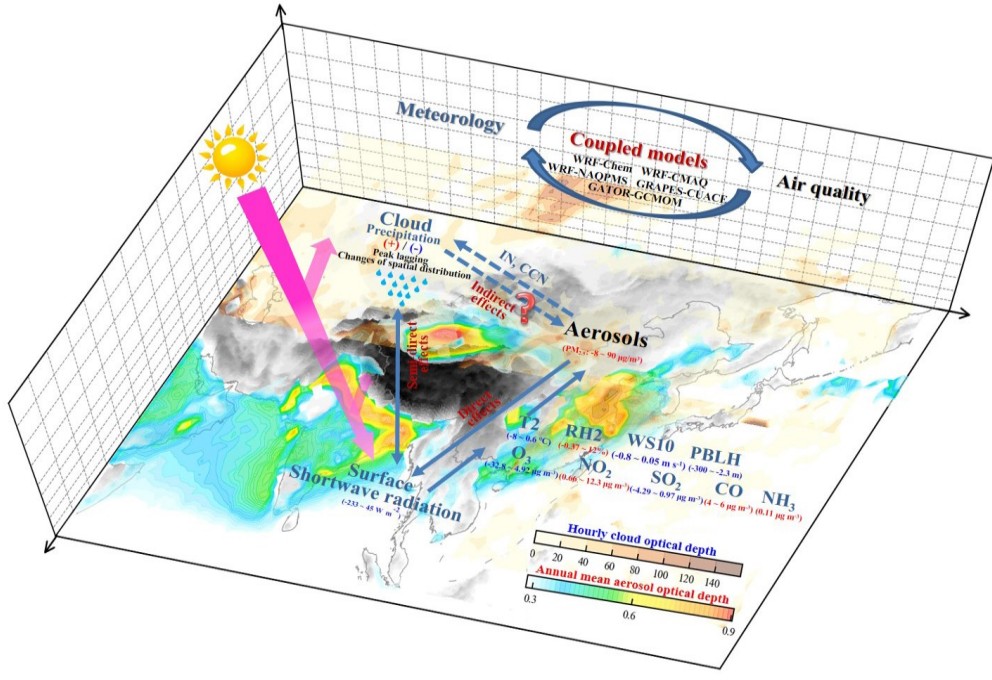

 Figure 9. A schematic diagram depicting aerosol-radiation-cloud interactions and quantitative effects of aerosol
 feedbacks on meteorological and air quality variables simulated by two-way coupled models in Asia.

Two-way coupled models have been applied in US and Europe extensively and then in Asia due to frequent occurrences of severe air pollution events accompanied with rapid economic growth in the region. Until now, no comprehensive study is conducted to elucidate the recent advances in two-way coupled models' applications in Asia. This paper provides a critical overview of current status and research focuses of related modeling studies using two-way coupled models in Asia between 2010 and 2019, and summarizes the effects of aerosol feedbacks on meteorological and air quality variables from these studies.

Through systematically searching peer-reviewed publications with several scientific-based search engines and a variety of key word combinations and applying certain selection criteria, 160 relevant papers were identified. Our bibliometric analysis results (as schematically illustrated in Fig. 9) showed that in Asia, the research activities with two-way coupled models had increased gradually in the past decade and the five two-way coupled models (WRF-Chem, WRF-CMAQ, WRF-NAQPMS, GRAPES-CUACE and GATOR-GCMOM) were extensively utilized to explore the ARI or/and ACI effects in Asia with focusing on several high aerosol loading areas (e.g., EA, India, China and NCP) during wintertime or/and severe pollution events, with less investigations looking into other areas and seasons with low pollution levels. Among the 160 papers, nearly 82 % of them focused on ARI (72 papers) and both ARI and ACI effects (60 papers), but papers that only considering ACI effects were relatively limited. The ARI or/and ACI effects of natural mineral dust, BC and BrC from anthropogenic sources and BC from ARB were mostly investigated, while a few studies quantitatively assessed the health impacts induced by aerosol effects.

Meta-analysis results revealed that enabling aerosol effects in two-way coupled models could improve their simulation/forecast capabilities of meteorology and air quality in Asia, but a wide range of differences occurred among the previous studies perhaps due to various model configurations (selections of model versions and parameterization schemes) and largest uncertainties related to ACI processes and their treatments in models. Compared to US and Europe, the aerosol-induced decrease of the shortwave radiative forcing was larger because of higher air pollution levels in Asia. The overall decrease (increase) of T2, WS10, PBLH and $O_3$ (RH2, $PM_{2.5}$ and other gaseous pollutant concentrations) caused by ARI or/and ACI effects were reported from the modeling studies using two-way coupled models in Asia. The ranges of aerosol-induced variations of T2, PBLH, $PM_{2.5}$ and $O_3$ concentrations were larger than other meteorological and air quality variables. For variables of CO, $SO_2$, $NO_2$, and $NH_3$, reliable estimates could not be obtained due to insufficient numbers of samples in past studies.

Even though noticeable progresses toward the application of two-way coupled meteorology
and air quality models have been made in Asia and the world during the last decade, several
limitations are still presented. Enabling aerosol feedbacks lead to higher computational cost
compared to offline models, but this shortcoming can be overcome with the new developments of
cluster computing technology (i.e., Graphics Processing Unit (GPU)-accelerated computing and
cloud computing). The latest advances in the measurements and research of cloud properties,
precipitation characteristics, and physiochemical characteristics of aerosols that play pivotal roles
in CCN or IN activation mechanisms can guide the improvements and enhancements in two-way
coupled models, especially to abate the uncertainties in simulating ACI effects. Special attention
needs to be paid to assess the accuracies of different methodologies in terms of ARI and ACI
calculations in two-way coupled models in Asia and other regions. Besides the five two-way coupled
models mentioned in this paper, more models capable of simulating aerosol feedbacks (such as
WRF-CHIMERE and WRF-GEOS-Chem) have become available and projects covering more
comprehensive intercomparisons of these coupled models should be conducted in Asia. Future
assessments of the ARI or/and ACI effects should pay extra attention to their impacts on dry and
wet depositions simulated by two-way coupled models. So far, the majority of two-way coupled
models' simulations and evaluations focuses on episodic air pollution events occurring in certain
areas, therefore their long-term applications and evaluations are necessary and their real-time
forecasting capabilities should be explored as well.
**Appendix A**

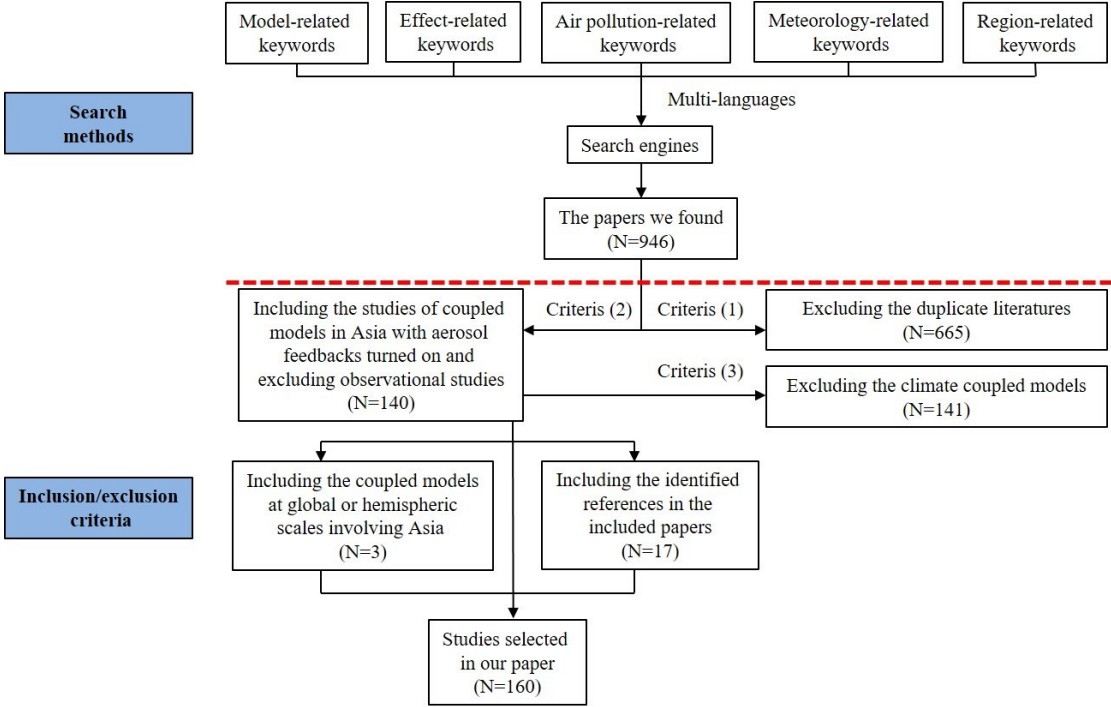

Figure A1. Flowchart of literature search and identification
**Appendix B**
Table B1. Lists of abbreviations and acronyms

| ACI | Aerosol-cloud interactions |
|---|---|
| AOD | Aerosol optical depth |
| AQCHEM | the CMAQ's standard aqueous chemistry module |
| ARB | Agriculture residue burning |
| ARB_BC | BC emitted from agriculture residue burning |
| ARB_CAs | Carbonaceous aerosols emitted from agriculture residue burning |
| ARI | Aerosol-radiation interactions |
| ATM | In the atmosphere |
| BB | Biomass burning |
| BC | Black carbon |
| BCs | Boundary conditions |
| BOT | At the bottom |

| | |
|---|---|
| BrC | Brown carbon |
| CA | Central Asia |
| CAMx | Comprehensive Air quality Model with extensions |
| CAs | Carbonaceous aerosols |
| CC | Central China |
| CCN | Cloud condensation nuclei |
| CDNC | Cloud droplet number concentration |
| CHIMERE | A multi-scale chemistry-transport model for atmospheric composition analysis and forecast |
| CMAQ | Community Multiscale Air Quality model |
| CO | Carbon monoxide |
| CRFs | Concentration-response functions |
| DRF | Direct radiative forcing |
| EA | East Asia |
| EC | East China |
| EQUISOLV II | the EQUIlibrium SOLVer version 2 |
| GATOR-GCMOM | Gas, aerosol, transport, radiation, general circulation, mesoscale, and ocean Model |
| GOCART | The Global Ozone Chemistry Aerosol Radiation and Transport |
| GPRAPES-CUACE | Global-regional assimilation and prediction system coupled with the Chinese Unified Atmospheric Chemistry Environment forecasting system |
| GSI | Gridpoint Statistical Interpolation |
| $H_2O_2$ | Hydrogen peroxide |
| $HNO_3$ | Nitric acid |
| $HO_2\cdot$ | Hydroperoxyl |
| ICs | Initial conditions |
| IN | Ice nuclei |
| INPs | Ice nucleation parameterizations |
| IPCC | Intergovernmental Panel on Climate Change |
| IPR | Ice particle radius |
| IWP | Ice water path |
| LWP | Liquid water path |
| LWRF | Longwave radiative forcing |
| MARS-A | the Model for an Aerosol Reacting System-version A |
| MB | Mean bias |
| ME | Middle East |
| MESA-MTEM | the Multicomponent Equilibrium Solver for Aerosols with the Multicomponent Taylor Expansion Method |
| MICS-Asia | Model Inter-Comparison Study for Asia |
| MOZART | Model for Ozone and Related Chemical Tracer |
| MRYR | Middle reaches of the Yangtze River |
| N | Nitrate |
| $N_2O_5$ | Nitrogen pentoxide |
| NAQPMS | Nested Air Quality Prediction Modeling System |
| NC | North China |
| NCP | North China Plain |
| NEA | Northeast Asia |
| NME | Normalized mean error |
| $NO_2$ | Nitrogen dioxide |
| NU-WRF | National aeronautics and space administration Unified Weather Research and Forecasting model |
| NWC | Northwest China |
| $O_3$ | Ozone |
| OA | Organic aerosols |
| OC | Organic carbon |
| ·OH | Hydroxyl radical |
| OPAC | Optical Properties of Aerosols and Clouds |
| PBL | Planetary boundary layer |
| PBLH | Planetary boundary layer height |
| $PM_{2.5}$ | Fine particulate matter |
| PRD | Pearl River Delta |
| PSI | Papers with statistical indices |
| R | Correlation coefficient |
| RADM | the Regional Acid Deposition Mode |
| RH2 | Relative humidity at 2 meters above the surface |
| RMSE | Root mean square error |
| RRTM | The Rapid Radiative Transfer Model |
| RRTMG | The Rapid Radiative Transfer Model for General Circulation Models |
| S | Sulfate |
| SA | South Asia |
| SC | South China |
| SEA | Southeast Asia |
| SEC | Southeast China |
| SH2 | Specific humidity at 2 meters above the surface |
| SI | Statistical indices |
| $SO_2$ | Sulfur dioxide |
| SOA | Secondary organic aerosol |
| SWC | Southwest China |
| SWRF | Shortwave radiative forcing |
| T2 | Air temperature at 2 meters above the surface |
| TOA | At the top of atmosphere |
| TP | Tibetan Plateau |
| US | the United States |
| VBS | Volatility basis set |

| | |
|---|---|
| WA | West Asia |
| WRF | Weather Research and Forecasting model |
| WRF-Chem | Weather Research and Forecasting model coupled with Chemistry |
| WRF-CHIMERE | Weather Research and Forecasting model coupled with a multi-scale Chemistry-Transport Model (CTM) for air quality forecasting and simulation |
| WRF-CMAQ | Weather Research and Forecasting model coupled with Community Multiscale Air Quality model |
| WRF-NAQPMS | Weather Research and Forecasting model coupled with the Nested Air Quality Prediction Modeling System |
| WS10 | Wind speed at 10 meters above the surface |
| YRD | Yangtze River Delta |

Table B2. The compiled number of publications (NP) and number of samples (NS) for papers that providing
statistical indices (SI) of meteorological variables.

| | Meteorological variables | | | | | | | | | | | | | | | |
|---|---|---|---|---|---|---|---|---|---|---|---|---|---|---|---|---|
| | T2 | | | | RH2 | | | | SH2 | | | | WS10 | | | |
| No.* | NP | NS | | | NP | NS | | | NP | NS | | | NP | NS | | |
| | | R | MB | RMSE | | R | MB | RMSE | | R | MB | RMSE | | R | MB | RMSE |
| 4 | 1 | 5 | 5 (4↑, 1↓) | 5 | 1 | 5 | 5 (1↑, 4↓) | 5 | | | | | | | | |
| 5 | | | | | 1 | | 3 (2↑, 1↓) | 3 | | | | | | | | |
| 7 | 1 | 4 | 4 (3↑, 1↓) | | | | | | | | | | | | | |
| 13 | 1 | | 1 (1↓) | | 1 | | 1 (1↑) | | | | | | | | | |
| 15 | 1 | 1 | | | 1 | 1 | | | | | | | 1 | 2 | | |
| 16 | 1 | 1 | | | | | | | | | | | | | | |
| 20 | 1 | 2 | 2 (1↑, 1↓) | 2 | 1 | 2 | 2 (1↑, 1↓) | 2 | | | | | 1 | 1 | 1 (1↑) | 1 |
| 21 | 1 | 0 | 2 (2↓) | 2 | | | | | | | | | 1 | | 2 (1↑, 1↓) | 2 |
| 22 | 1 | 1 | 1 (1↓) | 1 | 1 | 1 | 1 (1↑) | | | | | | 1 | 1 | 1 (1↓) | 1 |
| 23 | 1 | 1 | 1 (1↑) | | 1 | 1 | 1 (1↓) | | | | | | 1 | 1 | 1 (1↑) | |
| 24 | 1 | 1 | 1 (1↑) | | 1 | 1 | 1 (1↓) | | | | | | 1 | 1 | 1 (1↑) | |
| 25 | 1 | 1 | 1 (1↓) | | | | | | | | | | | | | |
| 28 | 1 | | 1 (1↑) | 1 | 1 | | 1 (1↓) | 1 | | | | | 1 | | 1 (1↑) | 1 |
| 29 | 1 | 9 | 9 (6↑, 3↓) | 9 | 1 | 8 | | 9 | | | | | 1 | 9 | 9 (9↑) | 9 |
| 33 | 1 | 6 | 6 (4↑, 2↓) | 6 | | | | | | | | | | | | |
| 34 | 1 | 2 | 2 (2↑) | 2 | | | | | | | | | 1 | 2 | 2 (2↓) | 2 |
| 35 | 1 | 2 | | 2 | 1 | 1 | | 1 | | | | | 1 | 1 | | 1 |
| 38 | 1 | | 4 (4↓) | 4 | 1 | | 4 (3↑, 1↓) | 4 | | | | | | | | |
| 50 | 1 | | 8 (8↓) | 8 | | | | | | | | | | | | |
| 56 | 1 | 1 | 1 (1↓) | 1 | 1 | 1 | 1 (1↓) | 1 | | | | | 1 | 1 | 1 (1↑) | 1 |
| 57 | 1 | 1 | | | 1 | 1 | | | | | | | 1 | 1 | | |
| 61 | 1 | 4 | 4 (4↓) | 4 | 1 | 4 | 4 (4↑) | 4 | | | | | 1 | 4 | 4 (4↑) | 4 |
| 62 | 1 | | 5 (5↓) | 5 | | | | | | | | | 1 | | 5 (4↑, 1↓) | 5 |
| 63 | 1 | 1 | | | | | | | | | | | | | | |
| 71 | 1 | 1 | | | | | | | | | | | | | | |
| 72 | 1 | 4 | 4 (3↑, 1↓) | 4 | 1 | 4 | 4 (3↑, 1↓) | 4 | | | | | | | | |
| 73 | 1 | 1 | 1 (1↓) | 1 | | | | | 1 | 1 | 1 (1↑) | 1 | 1 | 1 | 1 (1↑) | 1 |
| 75 | 1 | 4 | 4 (4↑) | | 1 | 4 | 4 (4↑) | | | | | 0 | 1 | 4 | 4 (1↑, 3↓) | |
| 77 | 1 | 4 | 4 (2↑, 2↓) | | | | | | 1 | 4 | 3 (3↑) | 4 | 1 | 4 | 4 (4↑) | 4 |
| 79 | 1 | | 8 (6↑, 2↓) | 8 | | | | | | | | | | | | |
| 80 | 1 | 8 | 8 (8↑) | 8 | 1 | 8 | 8 (8↓) | 8 | | | | | 1 | 8 | 8 (6↑, 2↓) | 8 |
| 85 | 1 | | 4 (1↑, 3↓) | 4 | 1 | | 4 (2↑, 2↓) | 4 | | | | | 1 | | 4 (4↑) | 4 |
| 87 | 1 | | 3 (2↑, 1↓) | 3 | | | | | | | | | 1 | | 3 (2↑, 1↓) | 3 |
| 88 | 1 | 3 | 3 (1↑, 2↓) | 3 | 1 | 3 | 3 (2↑, 1↓) | 3 | | | | | 1 | 3 | 3 (2↑, 1↓) | 3 |
| 90 | 1 | 4 | 4 (1↑, 3↓) | | | | | | 1 | 4 | 4 (4↑) | | 1 | 4 | 4 (4↑) | |
| 91 | 1 | 1 | 1 (1↓) | 1 | | | | | 1 | 1 | 1 (1↑) | 1 | 1 | 1 | 1 (1↑) | 1 |
| 94 | 1 | 6 | 6 (4↑, 2↓) | 6 | 1 | 6 | 6 (2↑, 4↓) | 6 | | | | | 1 | 6 | 6 (6↑) | 6 |
| 96 | 1 | 16 | 16 (11↑, 5↓) | | | | | | | | | | 1 | 16 | 16 (11↑, 5↓) | |
| 97 | 1 | 1 | 1 (1↓) | 1 | 1 | 1 | 1 (1↑) | 1 | | | | | 1 | 1 | 1 (1↑) | 1 |
| 106 | 1 | 6 | 6 (6↓) | | | | | | 1 | 6 | 5 (2↑, 3↓) | | 1 | 6 | 6 (6↑) | |
| 109 | 1 | 2 | 2 (2↓) | 2 | 1 | 3 | 3 (3↑) | 3 | | | | | 1 | 2 | 2 (2↓) | 2 |
| 112 | 1 | | 2 (2↓) | 2 | | | | | 1 | | 2 (2↓) | 2 | 1 | | 2 (2↑) | 2 |
| 116 | 1 | 2 | 2 (1↑, 1↓) | 0 | 1 | 2 | 2 (1↑, 1↓) | | | | | | | | | |
| 121 | 1 | 1 | 1 (1↓) | 1 | | | | | | | | | 1 | 1 | 1 (1↑) | 1 |
| 122 | 1 | | 2 (2↓) | 2 | 1 | | 2 (2↑) | 2 | | | | | 1 | | 2 (2↑) | 2 |
| 125 | 1 | 4 | 4 (4↓) | 4 | 1 | 4 | 4 (4↑) | 4 | | | | | 1 | 4 | 4 (4↓) | 4 |
| 126 | 1 | 4 | 4 (4↓) | | | | | | 1 | 4 | 4 (2↑, 2↓) | 4 | 1 | 4 | 4 (4↑) | 4 |
| 127 | 1 | | 2 (2↓) | 2 | | | | | | | | | 1 | | 2 (2↑) | 2 |
| 128 | 1 | 8 | 8 (8↓) | 8 | | | | | 1 | 8 | 8 (5↑, 3↓) | 8 | 1 | 8 | 8 (8↑) | 8 |
| 129 | 1 | 1 | 1 (1↓) | 1 | 1 | 1 | 1 (1↑) | 1 | | | | | 1 | 1 | 1 (1↑) | 1 |
| 133 | 1 | | 1 (1↓) | 0 | 1 | | 4 (4↑) | | | | | | 1 | | 4 (3↑, 1↓) | |
| 143 | 1 | | 4 | 4 | 1 | 4 | | 4 | | | | | 1 | | 4 | 4 |
| 147 | 1 | 2 | | 2 | 1 | 2 | | 2 | | | | | 1 | 2 | | 2 |
| 151 | 1 | 7 | 7 (7↓) | 7 | | | | | 1 | 7 | 7 (3↑, 4↓) | 7 | 1 | 7 | 7 (7↑) | 7 |
| Total | 53 | 137 | 167 (67↑, 100↓) | 130 | 30 | 68 | 70 (42↑, 28↓) | 73 | 9 | 35 | 35 (21↑, 14↓) | 27 | 40 | 111 | 126 (104↑, 22↓) | 97 |

Note that the No.* is consistent with the No. in Table 1, and ↑ and ↓ mark over- and underestimations of variables, respectively, along with
their number of samples.
Table B3. The compiled number of publications (NP) and number of samples (NS) for papers that providing
statistical indices (SI) of air quality variables.

| | Air quality variables | | | | | | | |
|---|---|---|---|---|---|---|---|---|
| | PM2.5 | | | | O3 | | | |
| No.* | NP | NS | | | NP | NS | | |
| | | R | MB | RMSE | | R | MB | RMSE |
| 4 | 1 | 5 | 5 (5↓) | 5 | | | | |
| 5 | 1 | | 1 (1↑) | 1 | 1 | | 1 (1↓) | 1 |
| 11 | 1 | 60 | | | | | | |
| 15 | 1 | 1 | | | | | | |
| 21 | 1 | | 2 (1↑, 1↓) | | | | | |
| 22 | 1 | 1 | 1 (1↑) | 1 | | | | |
| 23 | 1 | 1 | 1 (1↑) | | 1 | 1 | 1 (1↓) | |
| 24 | 1 | 1 | 1 (1↓) | | 1 | | 1 (1↓) | |
| 25 | 1 | 1 | 1 (1↑) | | 1 | 1 | 1 (1↑) | |
| 29 | 1 | 9 | 9 (6↑, 3↓) | 9 | | | | |

| No.* | NP | R | MB | RMSE | NP | R | MB | RMSE |
|---|---|---|---|---|---|---|---|---|
| 33 | 1 | 4 | 4 (4↓) | 4 | 1 | 4 | 4 (3↑, 1↓) | 4 |
| 34 | 1 | 2 | 2 (1↑, 1↓) | 2 | | | | |
| 35 | | | | | 1 | 1 | | 1 |
| 50 | 1 | | 4 (1↑, 3↓) | 4 | | | | |
| 56 | 1 | 1 | 1 (1↑) | 1 | | | | |
| 57 | 1 | 1 | | | | | | |
| 59 | 1 | 6 | 6 (6↓) | 6 | 1 | 6 | 6 (6↑) | 6 |
| 61 | 1 | 12 | 12 (12↑) | 12 | | | | |
| 67 | 1 | 10 | 2 (2↓) | 10 | | | | |
| 71 | 1 | 1 | | | | | | |
| 73 | 1 | 2 | 2 (1↑, 1↓) | | 1 | 4 | 4 (4↑) | |
| 77 | 1 | 4 | | | | | | |
| 85 | 1 | 3 | 3 (3↓) | | | | | |
| 86 | 1 | 4 | 4 (2↑, 2↓) | 4 | | | | |
| 88 | 1 | 3 | 3 (1↑, 2↓) | 3 | | | | |
| 90 | 1 | 8 | 8 (2↑, 6↓) | | 1 | 14 | 14 (14↑) | |
| 91 | 1 | 4 | 4 (1↑, 3↓) | 4 | 1 | 6 | 6 (4↑, 2↓) | 6 |
| 94 | 1 | 4 | 4 (3↑, 1↓) | 4 | | | | |
| 97 | 1 | 1 | 1 (1↓) | 1 | | | | |
| 100 | 1 | 1 | | | 1 | 1 | | |
| 106 | 1 | 6 | 6 (2↑, 4↓) | | 1 | 8 | 8 (4↑, 4↓) | |
| 112 | 1 | | | | | 1 | | |
| 121 | | | | | 1 | | | 5 |
| 122 | 1 | 4 | 4 (1↑, 3↓) | | | | | |
| 125 | 1 | 4 | 4 (2↑, 2↓) | 4 | 1 | 4 | 4 (4↑) | 4 |
| 126 | 1 | 4 | 4 (2↑, 2↓) | 4 | 1 | 4 | 4 (4↑) | 4 |
| 127 | 1 | | 1 (1↑) | 1 | | | | |
| 128 | 1 | 8 | 8 (3↑, 5↓) | 8 | | | | |
| 129 | 1 | 3 | 3 (2↑, 1↓) | 3 | 1 | 2 | 2 (1↑, 1↓) | 2 |
| 133 | | | | | 1 | 4 | 4 (3↑, 1↓) | 4 |
| 136 | 1 | 5 | 5 (5↓) | | | | | |
| 146 | 1 | 1 | | | 1 | 20 | | 20 |
| 147 | 1 | 2 | | 2 | | | | |
| 149 | 1 | 6 | | 6 | | | | |
| 150 | | | | | 1 | 21 | | 21 |
| 151 | 1 | 12 | 6 (6↑) | 6 | 1 | 24 | 12 (7↑, 5↓) | 12 |
| Total | 42 | 205 | 122 (55↑, 67↓) | 105 | 21 | 125 | 72 (55↑, 17↓) | 90 |

Note that the No.* is consistent with the No. in Table 1, and ↑ and ↓ mark over- and underestimations of variables, respectively, along with their number of samples.

Table B4. The compiled number of publications (NP) and number of samples (NS) for papers that simultaneously providing the statistical indices (SI) of meteorological variables simulated by coupled models (WRF-Chem, WRF-CMAQ, GRAPES-CUACE, WRF-NAQPMS and GATOR-GCMOM) with/out ARI.

| No.* | T2 | | | | RH2 | | | | SH2 | | | | WS10 | | | |
|---|---|---|---|---|---|---|---|---|---|---|---|---|---|---|---|---|
| | NP | NS R | NS MB | NS RMSE | NP | NS R | NS MB | NS RMSE | NP | NS R | NS MB | NS RMSE | NP | NS R | NS MB | NS RMSE |
| 32 | 1 | 3 | 3 (2↑, 1↓) | 3 | | | | | | | | | | | | |
| 78 | 1 | | 4 (3↑, 1↓) | 4 | | | | | | | | | | | | |
| 124 | 1 | 2 | 2 (2↓) | 2 | 1 | 2 | 2 (2↑) | 2 | | | | | 1 | 2 | 2 (2↓) | 2 |
| 125 | 1 | 2 | 2 (2↓) | 2 | | | | | 1 | 2 | 2 (1↑, 1↓) | 2 | 1 | 2 | 2 (2↑) | 2 |
| 126 | 1 | | 1 (1↓) | 1 | | | | | | | | | 1 | | 1 (1↑) | 1 |
| 127 | 1 | 4 | 4 (4↓) | 4 | | | | | 1 | 4 | 4 (3↑, 1↓) | 4 | 1 | 4 | 4 (4↑) | 4 |
| 146 | 1 | | 1 | 1 | 1 | 1 | | 1 | | | | | 1 | 1 | | 1 |
| Total | 7 | 12 | 16 (5↑, 11↓) | 17 | 2 | 3 | 2 (2↑) | 3 | 2 | 6 | 6 (4↑, 2↓) | 6 | 5 | 9 | 9 (7↑, 2↓) | 10 |

Note that the No.* is consistent with the No. in Table 1, and ↑ and ↓ mark over- and underestimations of variables, respectively, along with their number of samples.

Table B5. The compiled number of publications (NP) and number of samples (NS) for papers that simultaneously providing the statistical indices (SI) of air quality variables simulated by coupled models (WRF-Chem, WRF-CMAQ, GRAPES-CUACE, WRF-NAQPMS and GATOR-GCMOM) with/out ARI.

| No.* | PM$_{2.5}$ | | | | O$_3$ | | | |
|---|---|---|---|---|---|---|---|---|
| | NP | NS R | NS MB | NS RMSE | NP | NS R | NS MB | NS RMSE |
| 49 | 1 | | 2 (1↑, 1↓) | 2 | 1 | 10 | | 10 |
| 60 | 1 | 4 | 4 (4↑) | 4 | | | | |
| 124 | 1 | 2 | 2 (1↑, 1↓) | 2 | 1 | 2 | 2 (2↑) | 2 |
| 125 | 1 | 2 | 2 (1↑, 1↓) | 2 | 1 | 2 | 2 (2↑) | 2 |
| 127 | 1 | 4 | 4 (2↑, 2↓) | 4 | | | | |
| 146 | 1 | | 1 | 1 | | | | |
| Total | 5 | 13 | 14 (9↑, 5↓) | 15 | 3 | 14 | 4 (4↑) | 14 |

Note that the No.* is consistent with the No. in Table 1, and ↑ and ↓ mark over- and underestimations of variables, respectively, along with their number of samples.

Table B6. Description of refractive indices and radiation schemes used in the WRF-Chem and WRF-CMAQ in Asia.

| Model | Refractive indices of aerosol species groups | | Radiation scheme | |
|---|---|---|---|---|
| | SW | LW | SW scheme (Spectral intervals) | LW scheme (Spectral intervals) |
| WRF-Chem | 1. Water (1.35+1.524$^{-8}$i, | 1. Water (1.532+0.336i, | GODDARD (0.175-0.225, 0.225-0.245, 0.245- | RRTMG (10-350, 350-500, 500-630, 630-700, |

| | | | |
|---|---|---|---|
| | 1.34+2.494⁻⁹i, 1.33+1.638⁻⁹i, 1.33+3.128⁻⁶i) | 1.524+0.360i, 1.420+0.426i, 1.274+0.403i, 1.161+0.321i, | 0.260, 0.280-0.295, 0.295-0.310, 0.310-0.320, 0.325-0.400, 0.400-0.700, 0.700-1.220, 1.220- | 700-820, 820-980, 980-1080, 1080-1180, 1180-1390, 1390-1480, 1480-1800, 1800-2080, 2080- |

| Model | Column 2 | Column 3 | Column 4 | Column 5 |
|---|---|---|---|---|
| | 1.34+2.494$^{-9}$i, 1.33+1.638$^{-9}$i, 1.33+3.128$^{-6}$i)<br>2. Dust (1.55+0.003i, 1.550+0.003i, 1.550+0.003i, 1.550+0.003i)<br>3. BC (1.95+0.79i, 1.95+0.79i, 1.95+0.79i, 1.95+0.79i)<br>4. OC (1.45+0i, 1.45+0i, 1.45+0i, 1.45+0i)<br>5. Sea salt (1.51+8.66$^{-7}$i, 1.5+7.019$^{-8}$i, 1.5+1.184$^{-8}$i, 1.47+1.5$^{-4}$i)<br>6. Sulfate (1.52+1.00$^{-9}$i, 1.52+1.00$^{-9}$i, 1.52+1.00$^{-9}$i, 1.52+1.75$^{-6}$i) in term of 4 spectral intervals in 0.25-0.35, 0.35-0.45, 0.55-0.65, 0.998-1.000 μm | 1.524+0.360i, 1.420+0.426i, 1.274+0.403i, 1.161+0.321i, 1.142+0.115i, 1.232+0.0471i, 1.266+0.039i, 1.296+0.034i, 1.321+0.0344i, 1.342+0.092i, 1.315+0.012i, 1.330+0.013i, 1.339+0.01i, 1.350+0.0049i, 1.408+0.0142i)<br>2. Dust (2.34+0.7i, 2.904+0.857i, 1.748+0.462i, 1.508+0.263i, 1.911+0.319i, 1.822+0.26i, 2.917+0.65i, 1.557+0.373i, 1.242+0.093i, 1.447+0.105i, 1.432+0.061i, 1.473+0.0245i, 1.495+0.011i, 1.5+0.008i)<br>3. BC (1.95+0.79i, 1.95+0.79i, 1.95+0.79i, 1.95+0.79i, 1.95+0.79i, 1.95+0.79i, 1.95+0.79i, 1.95+0.79i, 1.95+0.79i, 1.95+0.79i, 1.95+0.79i, 1.95+0.79i, 1.95+0.79i, 1.95+0.79i,)<br>4. OC (1.86+0.5i, 1.91+0.268i, 1.988+0.185i, 1.439+0.198i, 1.606+0.059i, 1.7+0.0488i, 1.888+0.11i, 2.489+0.3345i, 1.219+0.065i, 1.419+0.058i, 1.426+0.0261i, 1.446+0.0142i, 1.457+0.013i, 1.458+0.01i)<br>5. Sea salt (1.74+0.1978i, 1.76+0.1978i, 1.78+0.129i, 1.456+0.038i, 1.41+0.019i, 1.48+0.014i, 1.56+0.016i, 1.63+0.03i, 1.4+0.012i, 1.43+0.0064i, 1.56+0.0196i, 1.45+0.0029i, 1.485+0.0017i, 1.486+0.0014i)<br>6. Sulfate (1.89+0.22i, 1.91+0.152i, 1.93+0.0846i, 1.586+0.2225i, 1.678+0.195i, 1.758+0.441i, 1.855+0.696i, 1.597+0.695i, 1.15+0.459i, 1.26+0.161i, 1.42+0.172i, 1.35+0.14i, 1.379+0.12i, 1.385+0.122i) in term of 16 spectral intervals in 10-350, 350-500, 500-630, 630-700, 700-820, 820-980, 980-1080, 1080-1180, 1180-1390, 1390-1480, 1480-1800, 1800-2080, 2080-2250, 2250-2390, 2390-2600, 2600-3250 cm$^{-1}$ | 0.260, 0.280-0.295, 0.295-0.310, 0.310-0.320, 0.325-0.400, 0.400-0.700, 0.700-1.220, 1.220-2.270, 2.270-10.00 μm)<br>RRTMG (3.077-3.846, 2.500-3.077, 2.150-2.500, 1.942-2.150, 1.626-1.942, 1.299-1.626, 1.242-1.299, 0.778-1.242, 0.625-0.778, 0.442-0.625, 0.345-0.442, 0.263-0.345, 0.200-0.263, 3.846-12.195 μm) | 700-820, 820-980, 980-1080, 1080-1180, 1180-1390, 1390-1480, 1480-1800, 1800-2080, 2080-2250, 2250-2390, 2390-2600, 2600-3250 cm$^{-1}$) |
| WRF-CMAQ | 1. Water (1.408+1.420$^{-2}$i, 1.324+1.577$^{-1}$i, 1.277+1.516$^{-3}$i, 1.302+1.159$^{-3}$i, 1.312+2.360$^{-4}$i, 1.321+1.713$^{-4}$i, 1.323+2.425$^{-5}$i, 1.327+3.125$^{-6}$i, 1.331+3.405$^{-8}$i, 1.334+1.639$^{-9}$i, 1.340+2.955$^{-9}$i, 1.349+1.635$^{-8}$i, 1.362+3.350$^{-8}$i, 1.260+6.220$^{-2}$i)<br>2. Water-soluble (1.443+5.718$^{-3}$i, 1.420+1.777$^{-2}$i, 1.420+1.060$^{-2}$i, 1.420+8.368$^{-3}$i, 1.463+1.621$^{-2}$i, 1.510+2.198$^{-2}$i, 1.510+1.929$^{-2}$i, 1.520+1.564$^{-2}$i, 1.530+7.000$^{-3}$i, 1.530+5.666$^{-3}$i, 1.530+5.000$^{-3}$i, 1.530+8.440$^{-3}$i, 1.530+3.000$^{-2}$i, 1.710+1.100$^{-1}$i)<br>3. BC (2.089+1.070i, 2.014+0.939i, 1.962+0.843i, 1.950+0.784i, 1.940+0.760i, 1.930+0.749i, 1.905+0.737i, 1.870+0.726i, 1.850+0.710i, 1.850+0.710i, 1.850+0.710i, 1.850+0.710i, 1.850+0.710i, 2.589+1.771i) | 1. Water (1.160+0.321i, 1.140+0.117i, 1.232+0.047i, 1.266+0.038i, 1.300+0.034i)<br>2. Water-soluble (1.570+0.069i, 1.700+0.055i, 1.890+0.128i, 2.233+0.334i, 1.220+0.066i)<br>3. BC (1.570+2.200i, 1.700+2.200i, 1.890+2.200i, 2.233+2.200i, 1.220+2.200i)<br>4. Insoluble (1.482+0.096i, 1.600+0.107i, 1.739+0.162i, 1.508+0.117i, 1.175+0.042i)<br>5. Sea-salt (1.410+0.019i, 1.490+0.014i, 1.560+0.017i, 1.600+0.029i, 1.402+0.012i) in term of 5 thermal windows at 13.240, 11.20, 9.73, 8.870, 7.830 μm | RRTMG (3.077-3.846, 2.500-3.077, 2.150-2.500, 1.942-2.150, 1.626-1.942, 1.299-1.626, 1.242-1.299, 0.778-1.242, 0.625-0.778, 0.442-0.625, 0.345-0.442, 0.263-0.345, 0.200-0.263, 3.846-12.195 μm) | RRTMG (10-350, 350-500, 500-630, 630-700, 700-820, 820-980, 980-1080, 1080-1180, 1180-1390, 1390-1480, 1480-1800, 1800-2080, 2080-2250, 2250-2390, 2390-2600, 2600-3250 cm$^{-1}$) |

4. Insoluble (1.272+1.165$^{-2}$i,
1.168+1.073$^{-2}$i, 1.208+8.650$^{-3}$i, 1.253+8.092$^{-3}$i,
1.329+8.000$^{-3}$i, 1.418+8.000$^{-3}$i,
1.456+8.000$^{-3}$i,
1.518+8.000$^{-3}$i, 1.530+8.000$^{-3}$i,
1.530+8.000$^{-3}$i,
1.530+8.000$^{-3}$i, 1.530+8.440$^{-3}$i, 1.530+3.000$^{-2}$i,
1.470+9.000$^{-2}$i)

5. Sea-salt (1.480+1.758$^{-3}$i,
1.534+7.462$^{-3}$i, 1.437+2.950$^{-3}$i,
1.448+1.276$^{-3}$i,
1.450+7.944$^{-4}$i, 1.462+5.382$^{-4}$i,
1.469+3.754$^{-4}$i,
1.470+1.498$^{-4}$i, 1.490+2.050$^{-7}$i,
1.500+1.184$^{-8}$i,
1.502+9.938$^{-8}$i, 1.510+2.060$^{-6}$i, 1.510+5.000$^{-6}$i,
1.510+1.000$^{-2}$i) in term of 14
wavelengths at 3.4615,
2.7885, 2.325, 2.046, 1.784,
1.4625, 1.2705, 1.0101,
0.7016, 0.53325, 0.38815,
0.299, 0.2316, 8.24 μm


Table B7. Summary of normalized mean error (NME) (%) of surface meteorological and air quality variables using
two-way coupled models (WRF-Chem and WRF-CMAQ).

| T2 | SH2 | RH2 | WS10 | PM$_{2.5}$ | O$_3$ | PM$_{2.5}$ with ARI (ARI) or without ARI (NO) | O$_3$ with ARI (ARI) or without ARI (NO) | Model | Region | Reference |
|---|---|---|---|---|---|---|---|---|---|---|
| | | | | | 23.60, 38.50, 55.70, 39.80 | | | WRF-Chem | EA | Liu X. et al. (2016) |
| 0.80, 0.60, 0.60, 0.60 | | 19.10, 16.50, 10.00, 10.10 | 58.90, 41.60, 44.90, 49.50 | 37.31, 37.61, 35.77, 34.69, 35.34, 35.41, 45.22, 44.33, 43.09, 39.29, 39.49, 39.07 | | 37.61, 35.34, 44.33, 39.49 (ARI) 35.77, 35.41, 43.09, 39.07 (NO) | | WRF-Chem | China | Zhang et al. (2018) |
| 270.20, 22.30, 12.50, 17.60 | | | | | | | | WRF-Chem | EA | Zhang Yang et al. (2016a) |
| | | | | 44.99, 29.55, 37.28 | | | | WRF-Chem | NCP | Yang et al. (2015) |
| 15.50, 15.80, 13.90, 9.90 | 10.40, 10.40, 9.90, 9.90 | | 31.30, 31.30, 32.50, 32.50 | 49.80, 65.30, 49.80, 65.60, 88.30, 56.90, 88.40, 57.00 | 127.00, 32.20, 25.40, 126.10, 32.10, 25.00, 79.90, 25.80, 21.40, 45.80, 77.90, 25.60, 21.10, 39.50 | | | WRF-Chem | EA | Zhang Y. et al. (2015a) |
| 14 | 11 | | 32 | 52.70, 58.00, 104.70, 62.00 | 87.50, 28.60, 23.30, 52.90, 32.40, 28.20 | | | WRF-Chem | EA | Chen Y. et al. (2015) |
| -0.48, 0.19, 0.21, 0.05, 0.08, 0.13, 0.05, 0.04, 0.04, 0.05, 0.02, 0.02, 0.06, 0.05, 0.04, 0.02 | | | 0.33, 1.92, 0.71, 0.78, 0.28, 1.72, 0.61, 0.64, 0.24, 1.76, 0.00, 0.45, 0.34, 1.29, 0.44, 0.56 | | | | | WRF-Chem | NCP | Chen D. et al. (2015) |
| 16.60, 10.50, 8.90, 12.90, 10.50, 10.20 | | | | | | | | WRF-Chem | EA | Wang K. et al. (2018) |
| 6.52, 6.58 | | 15.76, 12.15 | 112.28, 97.26 | | | | | WRF-Chem | NEA | Park et al. (2018) |
| | | | | 36.00, 33.00 | 31.00, 22.00 | | | WRF-Chem | China | Zhao et al. (2017) |
| | | | | 44.00, 44.60, 40.10, 54.30 | | | | WRF-Chem | NCP | Gao M. et al. (2015) |
| | | | | 41.48, 41.00, 51.77, 55.70 | 26.68, 26.71, 34.43, 34.64 | 41.00, 55.70 (ARI) 41.48, 51.77 (NO) | 26.71, 34.64 (ARI) 26.68, 34.43 (NO) | WRF-CMAQ | SEA | Nguyen et al. (2019b) |
| | | | | 37.99, 35.06, 38.59, 35.44, 34.39 | | | | WRF-CMAQ | China | Chang (2018) |


**Appendix C**
**C1 Comparisons of SI at different temporal scales for meteorology**
To probe the model performance of simulated T2, RH2, SH2 and WS2 at different temporal
scales, the SI of these meteorological variables from PSI were grouped according to the simulation
time (yearly, seasonal, monthly and daily) and plotted in Figure C1. Note that the seasonal results
contained SI values from simulations lasting more than one month and less than or equal to 3 months.
Here in Figure C1, NP and NS were the number of PSI and samples with SI at different time scales,
respectively, and also their total values were the same as the ones listed in Table S2. The correlation
between simulated and observed T2 (Figure C1a) at the seasonal (mean R= 0.97 with the smallest
sample size), yearly (0.91) and monthly (0.90) scales were stronger than that at the daily scale (0.87),
indicating that long-term simulations of T2 were well reproduced by coupled models. As shown in
Figure C1e, T2 underestimation mentioned above (Fig. 3a) appeared also in the seasonal, monthly
and yearly simulations (average MB = -0.87 °C, -0.15 °C and -0.34 °C, respectively), but the daily

T2 were overestimated (average MB = 0.07 °C). It should be noted that T2 at the monthly scale was underpredicted mainly during winter months (16 samples). Regarding the mean RMSE, its value (Figure C1i) at the daily scale was the largest (0.97 °C) in comparison with that at the other temporal scales.

Given that no SI was available for RH2 at the seasonal scale, results at other time scales were discussed here. Figure C1b presented that simulated RH2 at the daily scale had the best correlation coefficient (mean R= 0.74), followed by those at the monthly (0.73) and yearly (0.71) scales. Except overestimation (average MB = 3.6 %) at the yearly scale (Figure C1f), modeled RH2 were underestimated at the monthly (average MB = -1.1 %) and daily (average MB = -0.2 %) scales, respectively. Therefore, coupled models calculated RH2 reasonably well in short-term simulations. However, at the daily scale, RMSE of modeled RH2 (Figure C1j) was relatively large fluctuation ranging from 6.2 % to 21.3 %.

Lacking of SI for SH2 at the daily scale, only those at other time scales were compared. Even though NP and NS were very limited, the modeled SH2 (Figure C1c) exhibited especially good correlation with observations with the mean R values exceeding 0.95 at the yearly, seasonal and monthly scales (0.99, 0.97 and 0.96, respectively) but had the largest mean RMSE (2.09 g·kg$^{-1}$) at the yearly scale (Figure C1k). Also, both over- and under-estimations of modeled SH2 (Fig. C1g) were reported at different time scales with average MB values as 0.15 g·kg$^{-1}$, -0.02 g·kg$^{-1}$, and -0.14 g·kg$^{-1}$ for yearly, seasonal and monthly simulations, respectively. Generally, the long-term simulations of SH2 agreed better with observations than the short-term ones.

As seen in Figure C1d, the modeled WS10 at the monthly scale (mean R = 0.68) correlated with observations better than that at the daily, yearly and seasonal scales (mean R = 0.62, 0.48 and 0.46, respectively). The simulations at all temporal scales tended to overestimate WS10 comparing against observations (Figure C1h) and their average MB were 0.80 m·s$^{-1}$ (seasonal), 0.86 m·s$^{-1}$ (monthly), 0.64 m·s$^{-1}$ (yearly) and 0.62 m·s$^{-1}$ (daily), respectively. The short-term simulations of WS10 better matched with observations compared to the long-term ones. At the same time, the largest mean RMSE (1.79 m·s$^{-1}$) of simulated WS10 (Figure C1l) appeared at the seasonal scale.

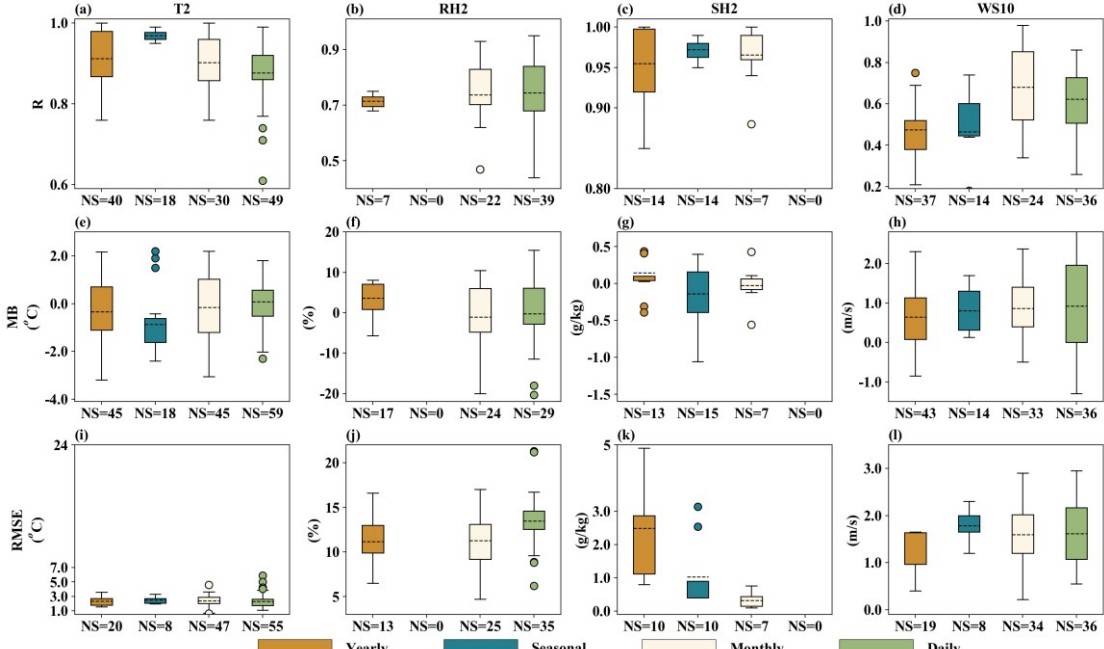

Figure C1. The statistical indices of modeled meteorological variables at different temporal scales (Yearly, Seasonal, Monthly and Daily) from past studies in Asia.

## C2 Comparisons of SI at different temporal scales for air quality

Figure C2 depicted the SI of simulated PM$_{2.5}$ and O$_3$ at yearly, seasonal, monthly and daily scales. The correlation between simulated and observed PM$_{2.5}$ (Figure C2a) at the monthly scale (mean R= 0.68) was largest compared to those at the yearly (0.64), seasonal (0.59), daily (0.57) scales. All the simulated PM$_{2.5}$ were underestimated, with the average daily, monthly, seasonal, and

yearly MB as -4.13, -1.46, -0.28, and -1.89 $\mu g \cdot m^{-3}$, respectively (Figure C2c). As displayed in Figure
C2e, the mean RMSE at the monthly scale was the largest (61.57 $\mu g \cdot m^{-3}$).
Regarding to correlation between simulated and observed $O_3$ (Figure C2b), it was the best at
the daily scale (mean R= 0.77). Modeled $O_3$ were overestimated at the seasonal (average MB =
+4.12 $\mu g \cdot m^{-3}$), monthly (average MB = +6.11 $\mu g \cdot m^{-3}$) and yearly (average MB = +11.71 $\mu g \cdot m^{-3}$)
scales, but underestimated at the daily scale (average MB =-8.89 $\mu g \cdot m^{-3}$) (Figure C2d). Note that no
RMSE for $O_3$ simulation was available at the daily scale, and the RMSE at the yearly scale (Figure
C2f) had relatively large fluctuation ranging from 0.21 to 71 $\mu g \cdot m^{-3}$. Therefore, coupled models
calculated $O_3$ matched well with observation in short-term simulations.

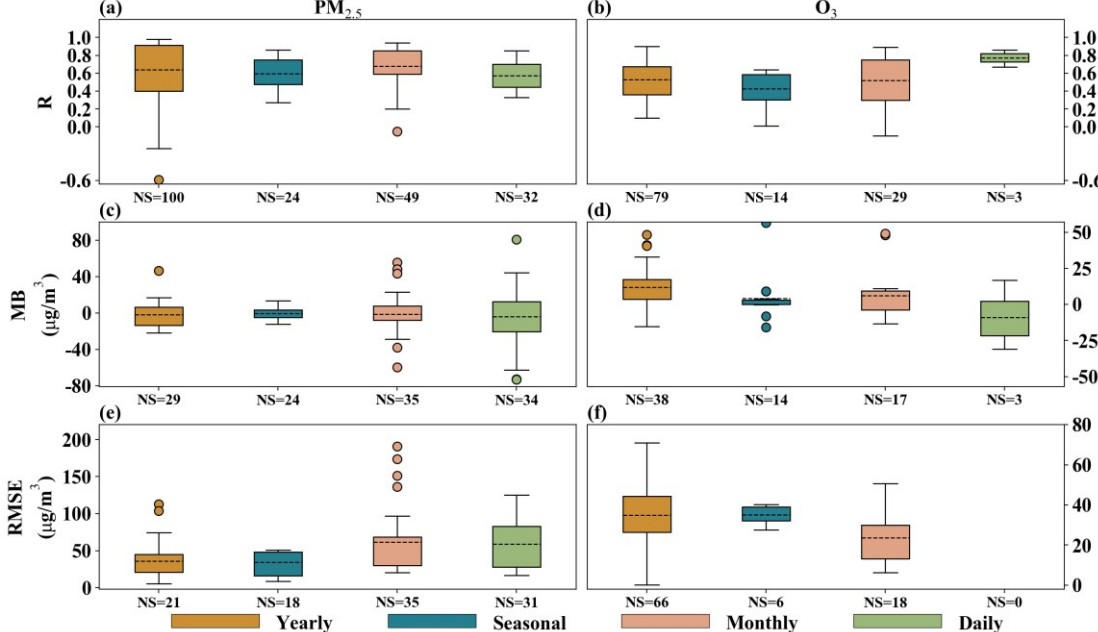

Figure C2. The quantile distributions of simulated $PM_{2.5}$ and $O_3$ performance metrics at different temporal scales
from past studies in Asia.

**Data availability**
The related dataset can be downloaded from https://doi.org/10.5281/zenodo.5571076 (Gao et
al., 2021), and this dataset includes basic information (Table S1), performance metrics (Table S2),
quantitative effects of aerosol feedbacks on meteorological and air quality variables (Table S3),
model configuration and setup (Table S4) and aerosol-induced variations of simulated shortwave
and longwave radiative forcing (Table S5) extracted from collected studies of applications of two-
way coupled meteorology and air quality models in Asia.

**Author contribution**
Chao Gao, Aijun Xiu, Xuelei Zhang and Qingqing Tong carried out the data collection, related
analysis, figure plotting, and manuscript writing; Hongmei Zhao, Shichun Zhang, Guangyi Yang
and Mengduo Zhang involved with the original research plan and made suggestions to the
manuscript writing.

**Competing interest**
The authors declare that they have no conflict of interest.

**Acknowledgement**
This study was financially sponsored by National Key Research and Development Program of
China (No. 2017YFC0212304 and No. 2019YFE0194500), Talent Program of Chinese Academy of
Sciences, and National Natural Science Foundation of China (No. 42171142, No. 41771071 and No.
41571063). The authors are very grateful to many researchers who provided detailed information
on the two-way coupled models and related research work. The list includes but is not limited to
Xueshun Chen, Zifa Wang, Yi Gao, Meigen Zhang and Baozhu Ge (Institute of Atmospheric Physics,
Chinese Academy of Sciences), Chunhong Zhou (Chinese Academy of Meteorological Sciences),
Yang Zhang (Northeastern University), Mark Zachary Jacobson (Stanford University), Tianliang
Zhao (Nanjing University of Information Science & Technology), Xin Huang (Nanjing University),
Chun Zhao (University of Science and Technology of China), Junhua Yang and Shichang Kang
(Northwest Institute of Eco-Environment and Resources, Chinese Academy of Sciences), Sachin
Ghude (Ministry of Earth Sciences Government of India) and Luke Conibear (University of Leeds).
We would also like to express our deepest appreciation to the editor James Allan and two anonymous
reviewers for their constructive comments and suggestions, which helped to improve the quality and
readability of this article.

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
