# Peer review of "Two-way coupled meteorology and air quality models in Asia: a systematic review"

_Atmospheric Chemistry and Physics, 2021_

## Author Comment (AC1)

We really appreciate the insightful and constructive comments by the Reviewer 1 regarding our manuscript. On behalf of all the co-authors, we made every effort to address these comments and revised the manuscript accordingly to improve its quality. Following the Reviewer's comments in black, please find our point-to-point responses in blue. Hereafter, all new added or modified sentences are marked in blue and italic in this response.

General Comments:

1.The paper does a thorough job of reviewing the studies involving coupled Met-AQ modeling with aerosol feedback effects, but it does not provide summary of the methods used to represent ARI and ACI or any assessment of the realism of the different models. It seems important to explain various the methods used to represent ARI and ACI and give some information on their accuracy.

Response: We agreed that it is useful to provide more detailed information about how ARI and ACI are treated in the five two-way coupled models applied the most in Asia. Therefore in the revised manuscript, we summarized the aspects for calculating ARI (including aerosol species groups, aerosol size distribution in different aerosol mechanisms, mixing states, and short- and long-wave radiation schemes) and ACI (including CCN and IN activation methods in microphysics schemes) in WRF-Chem, WRF-CMAQ, GRAPES-CUACE, WRF-NAQPMS and GATOR-GCMOM in Table 4. Please note that according to the Reviewer 2's suggestion, relevant information of GATOR-GCMOM was extracted and added in Table 4 as well. Table B6 in Appendix B of the revised manuscript further presents description of refractive indices of different aerosol species groups used in short- and long-wave radiation schemes in WRF-Chem and WRF-CMAQ. Due to unavailability of source codes, relevant information in other three coupled models (GRAPES-CUACE, WRF-NAQPMS and GATOR-GCMOM) is not presented in this table.

*Table 4. Summary of relevant information regarding calculations of aerosol-radiation interactions (ARI) and aerosol-cloud interactions (ACI) in two-way coupled models (WRF-Chem, WRF-CMAQ, GRAPES-CUACE, WRF-NAQPMS and GATOR-GCMOM) applied in Asia.*

| Model | ARI | | | | | ACI | |
|---|---|---|---|---|---|---|---|
| | Aerosol species groups | Aerosol size distribution (Aerosol mechanism) | Mixing state‡ | SW scheme (# of spectral intervals) | LW scheme (# of spectral intervals) | CCN (Microphysics scheme) | IN (Microphysics scheme) |
| WRF-Chem | 1. Water 2. Dust 3. BC 4. OC 5. Sea-salt 6. Sulfate | 1. Bulk (GOCART) 2. Modal (MADE/SORGAM, AERO5, MAM3 and MAM7) 3. Sectional (MOSAIC (4bins and 8 bins) and MADRID (8bins)) | Internal mixing (Volume averaging, Core-shell, and Maxwell-Garnett) | 1. Goddard (11) 2. RRTMG (14) | RRTMG (16) | Activation under a certain supersaturation in an air parcel based on Köhler theory (Morrison, Lin, Thompson, WSM 6/5/3 class and Milbrandt-Yau) | Ice heterogeneous nucleation of mineral dust aerosols in based on classical nucleation theory (Milbrandt-Yau and Morrison)‡ |
| WRF-CMAQ | 1. Water 2. Water-soluble 3. BC 4. Insoluble 5. Sea-salt | Modal (AERO5, AERO6 and AERO7) | Internal mixing (Core-shell) | RRTMG (14) | RRTMG (16) | None | None |
| GRAPES-CUACE | 1. Nitrate 2. Dust 3. BC 4. OC 5. Sea-salt 6. Sulfate 7. Ammonium | Sectional (CUACE (12 bins)) | External mixing | Goddard (11) | Goddard (10) | Activation under a certain supersaturation in an air parcel based on Köhler theory (WSM 6-class) | None |
| WRF-NAQPMS | 1. Nitrate 2. Dust 3. BC 4. OC 5. Sea-salt 6. Sulfate 7. Ammonium 8. Other primary particles | Modal (AERO5) | External mixing | Goddard (11) | RRTM (16) | Activation under a certain supersaturation in an air parcel based on Köhler theory (Lin) | None |
| GATOR-GCMOM | 1. Water 2. Dust 3. BC 4. HCO₃ 5. SOA 6. Sulfate … 42. MgCO₃(s) | Sectional (GATOR2012* (17-30 bins)) | Internal mixing (Core-shell‡) | Toon* (318) | Toon* (376) | Activation under a certain supersaturation in an air parcel based on Köhler theory (GATOR2012*) | Ice heterogeneous and homogeneous nucleation (GATOR2012*) |

‡ *Specific version of WRF-Chem, WRF-NAQPMS and GOTAR-GCMOM have the ability of simulating aerosol aging (Zhang et al.,*

2014;Chen et al., 2017; Li et al., 2018; Jacobson, 2012).

[†] *Some specific versions of WRF-Chem consider IN (Keita et al., 2020; Lee et al., 2020).*

[※] *The short- and long-wave radiation calculations in GATOR-GCMOM are based on the algorithm of Toon et al. (1989).*

[∗] *GATOR2012 refers to either the aerosol or cloud microphysics scheme used in Jacobson (2012).*

**Table B6. Description of refractive indices and radiation schemes used in the WRF-Chem and WRF-CMAQ models applied in Asia.**

| Model | Refractive indices of aerosol species groups | | Radiation scheme | |
|---|---|---|---|---|
| | SW | LW | SW scheme (Spectral intervals) | LW scheme (Spectral intervals) |
| WRF-Chem | 1. Water (1.35+1.524[-8]i, 1.34+2.494[-9]i, 1.33+1.638[-9]i, 1.33+3.128[-6]i) | 1. Water (1.532+0.336i, 1.524+0.360i, 1.420+0.426i, 1.274+0.403i, 1.161+0.321i, 1.142+0.115i, 1.232+0.0471i, 1.266+0.039i, 1.296+0.034i, 1.321+0.0344i, 1.342+0.092i, 1.315+0.012i, 1.330+0.013i, 1.339+0.01i, 1.350+0.0049i, 1.408+0.0142i) | GODDARD (0.175-0.225, 0.225-0.245, 0.245-0.260, 0.280-0.295, 0.295-0.310, 0.310-0.320, 0.325-0.400, 0.400-0.700, 0.700-1.220, 1.220-2.270, 2.270-10.00 μm) RRTMG (3.077-3.846, 2.500-3.077, 2.150-2.500, 1.942-2.150, 1.626-1.942, 1.299-1.626, 1.242-1.299, 0.778-1.242, 0.625-0.778, 0.442-0.625, 0.345-0.442, 0.263-0.345, 0.200-0.263, 3.846-12.195 μm) | RRTMG (10-350, 350-500, 500-630, 630-700, 700-820, 820-980, 980-1080, 1080-1180, 1180-1390, 1390-1480, 1480-1800, 1800-2080, 2080-2250, 2250-2390, 2390-2600, 2600-3250 cm[-1]) |
| | 2. Dust (1.55+0.003i, 1.550+0.003i, 1.550+0.003i, 1.550+0.003i) | | | |
| | 3. BC (1.95+0.79i, 1.95+0.79i, 1.95+0.79i, 1.95+0.79i) | | | |
| | 4. OC (1.45+0i, 1.45+0i, 1.45+0i, 1.45+0i) | | | |
| | 5. Sea salt (1.51+8.66[-7]i, 1.5+7.019[-8]i, 1.5+1.184[-8]i, 1.47+1.5[-4]i) | | | |
| | 6. Sulfate (1.52+1.00[-9]i, 1.52+1.00[-9]i, 1.52+1.00[-9]i, 1.52+1.75[-6]i) in term of 4 spectral intervals in 0.25-0.35, 0.35-0.45, 0.55-0.65, 0.998-1.000 μm | | | |
| | | 2. Dust (2.34+0.7i, 2.904+0.857i, 1.748+0.462i, 1.508+0.263i, 1.911+0.319i, 1.822+0.26i, 2.917+0.65i, 1.557+0.373i, 1.242+0.093i, 1.447+0.105i, 1.432+0.061i, 1.473+0.0245i, 1.495+0.011i, 1.5+0.008i) | | |
| | | 3. BC (1.95+0.79i, 1.95+0.79i, 1.95+0.79i, 1.95+0.79i, 1.95+0.79i, 1.95+0.79i, 1.95+0.79i, 1.95+0.79i, 1.95+0.79i, 1.95+0.79i, 1.95+0.79i, 1.95+0.79i, 1.95+0.79i, 1.95+0.79i,) | | |
| | | 4. OC (1.86+0.5i, 1.91+0.268i, 1.988+0.185i, 1.439+0.198i, 1.606+0.059i, 1.7+0.0488i, 1.888+0.11i, 2.489+0.3345i, 1.219+0.065i, 1.419+0.058i, 1.426+0.0261i, 1.446+0.0142i, 1.457+0.013i, 1.458+0.01i) | | |
| | | 5. Sea salt (1.74+0.1978i, 1.76+0.1978i, 1.78+0.129i, 1.456+0.038i, 1.41+0.019i, 1.48+0.014i, 1.56+0.016i, 1.63+0.03i, 1.4+0.012i, 1.43+0.0064i, 1.56+0.0196i, 1.45+0.0029i, 1.485+0.0017i, 1.486+0.0014i) | | |
| | | 6. Sulfate (1.89+0.22i, 1.91+0.152i, 1.93+0.0846i, | | |

| | | | |
|---|---|---|---|
| | 1.586+0.2225i, 1.678+0.195i, 1.758+0.441i, 1.855+0.696i, 1.597+0.695i, 1.15+0.459i, 1.26+0.161i, 1.42+0.172i, 1.35+0.14i, 1.379+0.12i, 1.385+0.122i) in term of 16 spectral intervals in 10-350, 350-500, 500-630, 630-700, 700-820, 820-980, 980-1080, 1080-1180, 1180-1390, 1390-1480, 1480-1800, 1800-2080, 2080-2250, 2250-2390, 2390-2600, 2600-3250 $cm^{-1}$ | | |
| WRF-CMAQ | 1. Water (1.408+1.420$^{-2}$i, 1.324+1.577$^{-1}$i, 1.277+1.516$^{-1}$i, 1.302+1.159$^{-1}$i, 1.312+2.360$^{-1}$i, 1.321+1.713$^{-1}$i, 1.323+2.425$^{-1}$i, 1.327+3.125$^{-6}$i, 1.331+3.405$^{-8}$i, 1.334+1.639$^{-9}$i, 1.340+2.955$^{-9}$i, 1.349+1.635$^{-8}$i, 1.362+3.350$^{-8}$i, 1.260+6.220$^{-2}$i) 2. Water-soluble (1.443+5.718$^{-3}$i, 1.420+1.777$^{-2}$i, 1.420+1.060$^{-2}$i, 1.420+8.368$^{-3}$i, 1.463+1.621$^{-2}$i, 1.510+2.198$^{-2}$i, 1.510+1.929$^{-2}$i, 1.520+1.564$^{-2}$i, 1.530+7.000$^{-3}$i, 1.530+5.666$^{-3}$i, 1.530+5.000$^{-3}$i, 1.530+8.440$^{-3}$i, 1.530+3.000$^{-2}$i, 1.710+1.100$^{-1}$i) 3. BC (2.089+1.070i, 2.014+0.939i, 1.962+0.843i, 1.950+0.784i, 1.940+0.760i, 1.930+0.749i, 1.905+0.737i, 1.870+0.726i, 1.850+0.710i, 1.850+0.710i, 1.850+0.710i, 1.850+0.710i, 1.850+0.710i, 2.589+1.771i) 4. Insoluble (1.272+1.165$^{-2}$i, 1.168+1.073$^{-2}$i, 1.208+8.650$^{-3}$i, 1.253+8.092$^{-3}$i, 1.329+8.000$^{-3}$i, 1.418+8.000$^{-3}$i, 1.456+8.000$^{-3}$i, 1.518+8.000$^{-3}$i, 1.530+8.000$^{-3}$i, 1.530+8.000$^{-3}$i, 1.530+8.000$^{-3}$i, 1.530+8.440$^{-3}$i, 1.530+3.000$^{-2}$i, 1.470+9.000$^{-2}$i) 5. Sea-salt (1.480+1.758$^{-3}$i, 1.534+7.462$^{-3}$i, 1.437+2.950$^{-3}$i, 1.448+1.276$^{-3}$i, 1.450+7.944$^{-4}$i, 1.462+5.382$^{-4}$i, 1.469+3.754$^{-4}$i, 1.470+1.498$^{-4}$i, 1.490+2.050$^{-7}$i, 1.500+1.184$^{-6}$i, 1.502+9.938$^{-8}$i, 1.510+2.060$^{-6}$i, 1.510+5.000$^{-6}$i, 1.510+1.000$^{-2}$i) in term of 14 wavelengths at 3.4615, 2.7885, 2.325, 2.046, 1.784, 1.4625, 1.2705, 1.0101, 0.7016, 0.53325, 0.38815, 0.299, 0.2316, 8.24 $\mu m$ | 1. Water (1.160+0.321i, 1.140+0.117i, 1.232+0.047i, 1.266+0.038i, 1.300+0.034i) 2. Water-soluble (1.570+0.069i, 1.700+0.055i, 1.890+0.128i, 2.233+0.334i, 1.220+0.066i) 3. BC (1.570+2.200i, 1.700+2.200i, 1.890+2.200i, 2.233+2.200i, 1.220+2.200i) 4. Insoluble (1.482+0.096i, 1.600+0.107i, 1.739+0.162i, 1.508+0.117i, 1.175+0.042i) 5. Sea-salt (1.410+0.019i, 1.490+0.014i, 1.560+0.017i, 1.600+0.029i, 1.402+0.012i) in term of 5 thermal windows at 13.240, 11.20, 9.73, 8.870, 7.830 $\mu m$ | RRTMG (3.077-3.846, 2.500-3.077, 2.150-2.500, 1.942-2.150, 1.626-1.942, 1.299-1.626, 1.242-1.299, 0.778-1.242, 0.625-0.778, 0.442-0.625, 0.345-0.442, 0.263-0.345, 0.200-0.263, 3.846-12.195 $\mu m$) | RRTMG (10-350, 350-500, 500-630, 630-700, 700-820, 820-980, 980-1080, 1080-1180, 1180-1390, 1390-1480, 1480-1800, 1800-2080, 2080-2250, 2250-2390, 2390-2600, 2600-3250 $cm^{-1}$) |

The following two paragraphs and Table 4 are added into a newly added Section 3.3 (Summary of modeling methodologies) in the revised manuscript. We also changed the title of Section 3 to "Basic overview" to reflect these changes.

*"Table 4 further lists various aspects with regards to how ARI and ACI being calculated in the five two-way coupled models (WRF-Chem, WRF-CMAQ, GRAPES-CUACE, WRF-NAQPMS, and GATOR-GCMOM) applied in Asia. Note that the information in this table was extracted from the latest released version of WRF-Chem (version 4.3.3) and WRF-CMAQ (based on WRF v4.3 and CMAQ v5.3.3) as well as relevant references for GRAPES-CUACE (Wang et al., 2015), WRF-NAQPMS (Wang et al., 2014) and GATOR-GCMOM (Jacobson et al., 2010; 2012). These models all use the Mie theory to compute ARI effects but differ in representations of aerosol optical properties and radiation schemes. To simplify the calculation, aerosol species simulated*

*by the chemistry module/model are put into different groups (Table 4) and the refractive indices of these groups are directly from the Optical Properties of Aerosols and Clouds (OPAC) database (Hess et al., 1998) in WRF-Chem and WRF-CMAQ (Table B6 in Appendix B). In WRF-Chem, the aerosol optical properties (AOD, extinction/scattering/absorption coefficient, single scattering albedo and asymmetry factor) are calculated in terms of four spectral intervals (listed in Table B6 in Appendix B) and then inter/extrapolated to 11 (14) SW intervals defined in the GODDARD (RRTMG) scheme. For SW and LW radiation in both WRF-CMAQ and WRF-Chem, these optical parameters are computed at each of corresponding spectral intervals in the RRTMG scheme. The aerosol optical property for LW radiation is considered only at 5 thermal windows (listed in Table B6) in WRF-CMAQ. No detailed information regarding how aerosol optical property and relevant parameters being calculated in GRAPES-CUACE and WRF-NAQPMS can be found from the relevant references.*

*With respect to ACI effects, the simulated aerosol characteristics (such as mass, size distribution and species) are utilized for the calculation of cloud droplet activation and aerosol resuspension based on the Köhler theory (Abdul-Razzak and Ghan, 2002) in several (one) microphysics schemes (scheme) in WRF-Chem (GRAPES-CUACE). GATOR-GCMOM is the first two-way coupled model adding IN activation processes including heterogeneous and homogeneous freezing (Jacobson et al., 2003). None of the other four two-way coupled models considers the IN formation processes (including immersion freezing, deposition freezing, contact freezing, and condensation freezing) but they have been included in some specific versions of WRF-Chem (Keita et al., 2020; Lee et al., 2020), which are not yet in the latest release version 4.3.3 of WRF-Chem."*

Hitherto in Asia, there are no assessment studies targeting how the various aspects of ARI and ACI calculations in two-way coupled models affect the accuracies of model simulations and rather limited studies in US and Europe. Baró et al. (2015) evaluated the impacts of two microphysics schemes (Morrison and Lin) on WRF-Chem simulations for a European domain and found out that no conclusive results indicating which scheme was more accurate, even though WRF-Chem with these two schemes did produce different cloud properties in various areas and seasons. Three combinations of gaseous and aerosol mechanisms (CBMZ-MOSAIC, MOZART-MOZAIC and RADM2-MAD/SORGAM) in WRF-Chem were compared over the Eastern Mediterranean by Georgiou et al. (2018) and the WRF-Chem with RADM2-MADE/SORGAM simulated $O_3$ and $PM_{2.5}$ slightly better than the other two mechanisms. Targeting a summertime aerosol pollution episode occurring in central Europe, Palacios-Peña et al. (2020) tweaked parameters set in the bulk size distribution and GOCART mechanism in WRF-Chem and investigated the sensitivities of AOD to different parameters defining aerosol size distribution in various modes.

2. The paper is very long, and I found it very difficult to read through the seemingly endless recitation of statistics that have very wide ranges without any explanation for the different results. The variety of modeling techniques, domains, resolutions, data assimilation, ICs and BCs, emissions, etc, should be considered in these comparisons. Why such wide ranges of results? Perhaps investigate the extremes to find out and maybe exclude studies with serious issues.
Response: To improve the paper's readability, we moved Section 5.1.2 and Section 5.2.2 to Appendix C in the revised manuscript. We thank the Reviewer 1 for pointing out that we should also outline the various aspects of how modeling studies being set up, which can affect the results of simulations and statistical analyses. A new Table S4

in Supplement of our revised manuscript illustrates the relevant information, and it is organized in the same order as Table 1 of the revised manuscript and contains extra/auxiliary information about model setup in the two-way coupled model applications in Asia.

*Table S4. Basic information of model setup for two-way coupled model applications in Asia.*

| No. | Grid resolution (km) | Vertical layer | Aerosol mechanism | Gas phase chemical mechanism | PBL scheme | Meteorological ICs and BCs | Chemical ICs and BCs | Anthropogenic emission | Natural emission | Reference |
|---|---|---|---|---|---|---|---|---|---|---|
| 1 | † | † | † | † | YSU | † | † | † | † | Singh et al. (2020)* |
| 2 | 30 | 28 | MADE/SORGAM | RADM2 | YSU | † | † | † | † | Bharali et al. (2019) |
| 3 | † | † | MOSACI | CMBZ | † | † | † | † | † | Shahid et al. (2019) |
| 4 | 18, 6 | 42 | MOSAIC (8 bins) | CBMZ | YSU | FNL | MOZART | 2010 MEIC | † | Wang et al. (2019) |
| 5 | 12 | 35 | AERO5 | SAPRC99 | MYJ | FNL | MOZART | 2012 MEIC | MEGAN | Wu et al. (2019a) |
| 6 | 12 | 35 | AERO5 | SAPRC99 | MYJ | FNL | MOZART | 2012 MEIC | MEGAN | Wu et al. (2019b) |
| 7 | 36 | 35 | MADE/SOGARM | RADM2 | YSU | FNL | † | † | † | Yuan et al. (2019) |
| 8 | 27, 9 | † | MOSAIC | CMBZ | YSU | FNL | † | † | † | Zhang et al. (2019) |
| 9 | 36 | 37 | MOSAIC (4 bins) | CBMZ | YSU | FNL | MOZART | 2012 MEIC/2010 MIX | MEGAN/Dust | Zhou et al. (2019) |
| 10 | 50 | 29 | MOSAIC | † | YSU | † | † | † | † | Bran et al. (2018) |
| 11 | 60 | 28 | MOSAIC (8 bins) | CBMZ | YSU | FNL | Default profile | MIX | MEGAN/GFED/Dust | Gao et al. (2018b) |
| 12 | 12, 4 | 24 | MOSAIC | CBMZ | YSU | † | † | † | † | Li M. M. et al. (2018) |
| 13 | 20 | 42 | † | † | † | † | † | † | † | Li and Sokolik (2018) |
| 14 | 9 | 40 | MOSAIC(4 bins) | CMBZ | MYJ TKE | FNL | † | † | † | Liu et al. (2018) |
| 15 | 15 | 21 | MADE/SOGARM | RADM2 | YSU | † | † | † | † | Miao et al. (2018) |
| 16 | 30 | 27 | MADE/SORGAM | RADM2 | † | † | † | † | † | Soni et al. (2018) |
| 17 | 36, 12 | 23 | MOSAIC | CBMZ | YSU | † | † | † | † | Wang L. T. et al. (2018) |
| 18 | 4 | 100 | MOSAIC | CMBZ | YSU | † | † | † | † | Wang Z. L. et al. (2018) |
| 19 | 25 | 30 | MOSAIC (4 bins) | GOCART | MYJ | FNL | † | † | † | Yang et al. (2018) |
| 20 | 20 | 30 | MOSAIC | CMBZ | YSU | FNL | † | MEIC | MEGAN | Zhou et al. (2018) |
| 21 | 81, 27 | † | MOSAIC(8 bins) | CBMZ | YSU | ECMWF | † | 2010 MIX | MEGAN/Dust | Gao et al. (2017c) |
| 22 | 81, 27, 9, 3 | 24 | MOSAIC(8 bins) | CBMZ | YSU | FNL | † | 2012 MEIC | MEGAN/Dust | Li et al. (2017a) |
| 23 | 81, 27, 9, 3 | 21 | MOSAIC | CBMZ | YSU | † | † | 2012 MEIC | MEGAN | Li et al. (2017b) |
| 24 | 90, 30, 10 | 33 | MOSAIC(8 bins) | CBMZ | YSU | FNL | MOZART | 2010 MIX | MEGAN/FINN/Dust/Sea salt | Qiu et al. (2017) |
| 25 | 27, 9, 3 | 41 | MOSAIC (4 bins) | CBMZ | MYJ | CFSR | MOZART | † | MEGAN/GFED | Yang and Liu (2017a) |
| 26 | 27, 9, 3 | 41 | MOSAIC (4 bins) | CBMZ | MYJ | CFSR | MOZART | † | MEGAN/GFED | Yang and Liu (2017b) |
| 27 | 75, 25 | 25 | MOSAIC (4 bins) | † | YSU | † | † | † | † | Yao et al. (2017) |
| 28 | 81, 27, 9 | 27 | MOSAIC | CBMZ | ACM2 | † | † | † | † | Zhan et al. (2017) |
| 29 | 12 | 27 | GOCART | MOZART | MYJ | † | † | † | † | Feng et al. (2016) |
| 30 | 81, 27 | 27 | MOSAIC(8 bins) | CBMZ | YSU | FNL | MOZART | MEIC | MEGAN | Gao et al. (2016b) |
| 31 | 36 | 23 | MADRID(8 bins) | CB05 | † | FNL | GEOS-Chem | 2006 INTEX-B | † | Liu et al. (2016) |
| 32 | 20 | 30 | MOSAIC | † | YSU | FNL | † | † | Dust | Liu et al. (2016) |
| 33 | 13.5, 4.5 | 48 | MADE/SOGARM | RADM2 | YSU | † | † | † | † | Miao et al. (2016) |
| 34 | 36 | 32 | MOSAIC | CBMZ | QNSE | FNL | MOZART | 2006 INTEX-B | † | Wang et al. (2016) |
| 35 | 3 | 40 | MOSAIC | CBMZ | YSU | † | † | † | † | Yang et al. (2016) |
| 36 | 20 | 31 | MADE/SORGAM | RADM2 | † | † | † | † | † | Zhong et al. (2016) |
| 37 | 12 | † | GOCART | MOZART | MYJ | † | † | † | † | Govardhan et al. (2015) |
| 38 | 50 | 15 | MOSAIC(4 bins) | CBMZ | YSU | FNL | † | 2006 INTEX-B | MEGAN/FINN/Dust/Sea salt | Huang et al. (2015) |
| 39 | 54 | 27 | MOSAIC | CBMZ | YSU | FNL | † | 2006 INTEX-B | MEGAN | Chen et al. (2014) |

| | | | | | | | | | | |
|---|---|---|---|---|---|---|---|---|---|---|
| 40 | 36 | 35 | MADE/SOGARM | RADM2 | † | FNL | † | 2006 INTEX-B | MEGAN/GFED | Gao et al. (2014) |
| 41 | 81, 27 | 27 | MADE/SOGARM | RADM2 | YSU | FNL | † | FLAMBE | † | Ge et al. (2014) |
| 42 | 30 | 51 | MOZART-4 | GOCART | † | FNL | † | † | † | Kumar et al. (2014) |
| 43 | 60 | 31 | MOSAIC(8 bins) | CBMZ | YSU | † | † | † | † | Li et al. (2014) |
| 44 | 27 | 35 | MADE/SOGARM | RADM2 | MYJ | FNL | † | 2006 INTEX-B | FINN/Dust | Lin et al. (2014) |
| 45 | 27, 9 | 50 | † | † | MYJ | † | † | † | † | Chen et al. (2013) |
| 46 | 27 | 50 | GOCART | † | BouLac | † | † | † | † | Dipu et al. (2013) |
| 47 | 45 | 51 | RADE/SOGARM | † | MYJ | † | † | † | † | Kumar et al. (2012a) |
| 48 | 45 | 51 | RADE/SOGARM | † | MYJ | † | † | † | † | Kumar et al. (2012b) |
| 49 | 25 | 19 | MOSAIC(8 bins) | CBMZ | † | FNL | † | † | † | Seethala et al. (2011) |
| 50 | 75 | 18 | † | † | † | FNL | † | † | † | Zhuang et al. (2011) |
| 51 | 20, 4 | 41 | MOSAIC (4 bins) | CBMZ | YSU | † | † | † | † | Liu et al. (2020)* |
| 52 | 5 | 33 | MADE/SOGARM | RADM2 | QNSE | FNL | † | 2006 INTEX-B | † | Jia et al. (2019) |
| 53 | 20 | 28, 40, 60 | MOSACI | CMBZ | YSU | † | † | † | † | Wang et al. (2019) |
| 54 | 27 | 51 | MOSACI | CMBZ | YSU | † | † | † | † | Nicholls et al. (2019) |
| 55 | 25 | † | MOSAIC (4 bins) | CBMZ | YSU | FNL | † | 2016 MEIC | MEGAN | Li et al. (2019) |
| 56 | 75, 25 | 72 | MADE/SORGAM | RADM2 | YSU | † | † | † | † | Kedia et al. (2019a) |
| 57 | 50 | 37 | MADE/SORGAM | RADM2 | YSU | FNL | † | EDGAR | MEGAN/MODIS_Fire | Kedia et al. (2019b) |
| 58 | 45 | † | MADE/SORGAM | RADM2 | YSU | † | † | † | † | Huang et al. (2019) |
| 59 | 15 | 26 | MOSAIC(4 bins) | MOZART | YSU | † | † | † | † | Ding et al. (2019) |
| 60 | 27, 9 | 29 | MOSAIC (8 bins) | CBMZ | YSU | FNL | MOZART | MIX | MEGAN/GFED/Dust/Sea salt | Chen et al. (2019b) |
| 61 | 35 | 12 | † | † | MYJ | † | † | † | † | An et al. (2019) |
| 62 | 27, 9 | 28 | MADE/SORGAM | RADM2 | YSU | † | † | † | † | Liu et al. (2018) |
| 63 | 27, 9, 3 | 35 | MADE/SOGARM | CB05 | YSU | FNL | MOZART | † | MEGAN/FINN/Dust/Sea salt | Liu et al. (2018) |
| 64 | 36 | 46 | MOSAIC (4 bins) | CBMZ | YSU | FNL | MOZART | MEIC | MEGAN/Dust | Zhang et al. (2018) |
| 65 | 36, 12 | 38 | MOSAIC | CMBZ | YSU | † | † | † | † | Gao et al. (2018) |
| 66 | 36 | 23 | MAM3 | CBMZ | † | † | † | † | † | Zhang et al. (2017) |
| 67 | 12 | 24 | MOSAIC (4 bins) | CBMZ | YSU | † | † | † | † | Wu et al. (2017) |
| 68 | 27, 9, 3 | 25 | MADE/SOGARM | RADM2 | YSU | † | † | † | † | Sun et al. (2017) |
| 69 | 3 | 50 | MADE/SORGAM | RADM2 | MYJ | FNL | Quasi-global WRF-Chem simulation | 2006 INTEX-B | MEGAN/FINN/Dust/Sea salt | Zhong et al. (2017) |
| 70 | 81, 27 | 27 | MOSAIC (8 bins) | CBMZ | YSU | FNL | MOZART | 2012 MEIC | MEGAN | Gao et al. (2017a) |
| 71 | 81, 27 | † | MOSAIC(8 bins) | CBMZ | YSU | ECMWF | † | 2010 MIX | MEGAN/Dust | Gao et al. (2017b) |
| 72 | 54 | 27 | † | † | † | † | † | † | † | Ma et al. (2017) |
| 73 | 27, 9 | 61 | † | † | † | † | † | † | † | Lau et al. (2017) |
| 74 | 20, 9 | 35 | MADE/SOGARM | RADM2 | MYJ | † | † | REAS v2/GFED v3.1 | Dust/Sea salt | Kajino et al. (2017) |
| 75 | 15 | 30 | MOSAIC | † | MYJ | FNL | MOZART | 2006 INTEX-B | MEGAN | Yang et al. (2017) |
| 76 | 36 | 23 | MAM3 | CBMZ | † | † | † | † | † | He et al. (2017) |
| 77 | 36, 12, 4 | † | † | † | † | † | † | † | † | Campbell et al. (2017) |
| 78 | 27, 9 | 30 | MADE/SORGAM | RADM2 | QNSE | FNL | † | 2012 MEIC | † | Zhang et al. (2016) |
| 79 | 54 | 30 | MOSAIC | CBMZ | YSU | FNL | † | † | † | Ma et al. (2016) |
| 80 | 36 | 23 | MOSAIC | CBMZ | YSU | † | † | † | † | Zhang et al. (2016a) |
| 81 | 36 | 23 | MOSAIC | CBMZ | YSU | † | † | † | † | Zhang et al. (2016b) |
| 82 | 20, 4 | 31 | MOSAIC | CBMZ | YSU | FNL | † | MEIC | MODIS_Fire | Huang et al. (2016) |
| 83 | 81, 27, 9 | 36 | MOSAIC | CBMZ | MYJ | † | † | † | † | Xie et al. (2016) |
| 84 | 45, 15, 5, 1.67 | 27 | MOSAIC(4 bins) | CBMZ | YSU | † | † | † | † | Srinivas et al. (2016) |
| 85 | 25 | 28 | MADE/SOGARM | RADM2 | YSU | † | † | † | † | Kedia et al. (2016) |
| 86 | 54 | 30 | MADE/SOGARM | † | YSU | † | † | † | † | Jin et al. (2016a) |

| # | Res | Lev | Aerosol | Chemistry | PBL | Met | Chem BC | Emission | Biogenic | Reference |
|---|---|---|---|---|---|---|---|---|---|---|
| 87 | 54 | 30 | MADE/SOGARM | † | YSU | † | † | † | † | Jin et al. (2016b) |
| 88 | 81, 27, 9 | 27 | MOSAIC(8 bins) | CBMZ | YSU | FNL | MOZART | MEIC | MEGAN | Gao et al. (2016a) |
| 89 | 36 | 35 | MADE/SOGARM | RADM2 | † | † | † | † | † | Gao et al. (2016) |
| 90 | 36 | 25 | MOSAIC | CMBZ | YSU | FNL | † | MEIC | MEGAN | Ding et al. (2016) |
| 91 | 27, 9 | 42 | MADE/SOGARM, MADE/SORGAM_aq, MOSAIC(8 bins) & MADE/SORGAM | RADM2, RADM2, CBMZ & CBMZ | YSU | FNL | † | 2010 MEIC | MEGAN/FINN/Dust/Sea salt | Yang et al. (2015) |
| 92 | 54 | 28 | MADE/SORGAM | RADM2 | † | FNL | † | † | MEGAN/FINN/Dust/Sea salt | Shen et al. (2015) |
| 93 | 36 | 23 | MAM3 | CBMZ | UW | FNL | † | REAS v2.1 | † | Zhang et al. (2015a) |
| 94 | 36 | 23 | MAM3 | CBMZ | UW | FNL | CMAQ/GEOS-Chem | MEIC/INTEX-B | MEGAN/Dust/Sea salt | Chen et al. (2015) |
| 95 | 36, 12, 4 | 35 | MADE/SOGARM | RADM2 | YSU | FNL | Quasi-global WRF-Chem simulation | 2006 INTEX-B | MEGAN/FINN/Dust/Sea salt | Zhong et al. (2015) |
| 96 | 54 | 30 | MADE/SOGARM | † | YSU | † | † | † | † | Jin et al. (2015) |
| 97 | 36 | † | GOCART | MOZART-4 | BouLac | † | † | † | † | Jena et al. (2015) |
| 98 | 27 | 51 | MOSAIC(8 bins) | CBMZ | † | FNL | MOZART | † | † | Gao Y. et al. (2015) |
| 99 | † | 40 | MOSAIC | CBMZ | † | † | † | 2006 INTEX-B | MEGAN/FINN/Dust/Sea salt | Fan et al. (2015) |
| 100 | 54 | 27 | MOSAIC | CBMZ | YSU | FNL | † | 2006 INTEX-B | MEGAN/FINN/Dust/Sea salt | Chen et al. (2015) |
| 101 | 27 | 28 | MOSAIC | CBMZ | YSU | FNL | MOZART | MEIC | MEGAN | Zhang et al. (2015) |
| 102 | 36 | † | MADE/SOGARM | † | YSU | † | † | † | † | Wu et al. (2013) |
| 103 | 45, 15, 5, 1.67 | 27 | MOSAIC(4 bins) | CBMZ | YSU | † | † | † | † | Beig et al. (2013) |
| 104 | 5 | 33 | MOSAIC (4 bins) | CBM-IV | QNSE | † | † | † | † | Jia et al. (2012) |
| 105 | † | 27 | MADRID | CB05 | YSU | † | † | † | † | Zhang et al. (2012) |
| 106 | 36 | 35 | MADE/SOGARM | RADM2 | MYJ | † | † | † | † | Gao et al. (2012) |
| 107 | 27, 9, 3 | 28 | MOSAIC (4 bins) | CBMZ | YSU | FNL | † | 2016 MIX | † | Bai et al. (2020)* |
| 108 | 81, 27, 9, 3 | 24 | MOSAIC | CBMZ | MYJ | † | † | † | † | Liu et al. (2019) |
| 109 | 36, 12, 4 | 23 | MAM3 | CBMZ | UW | † | † | † | † | Wang K. et al. (2018) |
| 110 | 27, 9 | 40 | GOCART | † | MYJ | † | † | † | † | Su et al. (2018a) |
| 111 | 27, 9 | 40 | GOCART | † | MYJ | † | † | † | † | Su et al. (2018b) |
| 112 | 27 | 15 | MADE/SOGARM | RACM | YSU | FNL | MOZART | 2015 MAPS-Seoul campaign emission | MEGAN | Park et al. (2018) |
| 113 | 36 | 35 | MOSAIC | CBMZ | † | † | † | † | † | Gao and Zhang (2018) |
| 114 | 18, 6 | 45 | MADE/SOGARM | RADM2 | YSU | FNL | † | 2006 INTEX-B | MEGAN | Shen et al. (2017) |
| 115 | 36 | 24 | MOSAIC | CBMZ | YSU | FNL | Default profile | 2010 MIX | MEGAN/Dust | Zhao et al. (2017) |
| 116 | 4.5 | † | † | † | † | † | † | † | † | Bhattacharya et al. (2017) |
| 117 | 36, 12, 4 | 31 | MADE/SOGARM | RADM2 | YSU | † | † | † | † | Jiang et al. (2016) |
| 118 | 36 | 23 | MAM3 | CBMZ | UW | FNL | † | † | † | Zhang et al. (2015b) |
| 119 | 27, 9, 3 | 34 | MOSAIC(4 bins) | CBMZ | MYJ | FNL | † | † | † | Sarangi et al. (2015) |
| 120 | 36 | 23 | † | † | † | † | † | † | † | Zhang et al. (2014) |
| 121 | 36 | 45 | MAM3 | † | YSU | † | † | † | † | Lin et al. (2014) |
| 122 | † | † | † | † | † | † | † | † | † | Bennartz et al. (2011) |
| 123 | 20 | † | MOSAIC | CBMZ | † | FNL | AM3 | 2008 MEIC/REAS/EDGAR v4.2 | MEGAN/FINN | Zhong et al. (2019) |
| 124 | 30 | 27 | MOSAIC (4 bins) | MOZART-4 using KPP | MYNN2 | FNL | MOZART | † | Dust | Conibear et al. (2018a) |
| 125 | 30 | 27 | MOSAIC (4 bins) | MOZART-4 using KPP | MYNN2 | FNL | MOZART | † | Dust | Conibear et al. (2018b) |
| 126 | 36 | † | GOCART | MOZART-4 | BouLac | † | † | † | † | Ghude et al. (2016) |
| 127 | 81, 27, 9 | † | MOSAIC(8 bins) | CBMZ | † | FNL | MOZART | 2010 MEIC | MEGAN | Gao M. et al. (2015) |
| 128 | 36 | 34 | AERO6 | CB05 | † | † | † | † | † | Dong et al. (2019) |
| 129 | 27 | † | AERO6 | CB05 | ACM2 | FNL | Default profile | † | † | Jung et al. (2019) |
| 130 | 45 | 30 | AERO6 | CB05 | ACM2 | FNL | MOZART | JEI-DB/INTEX-B | MEGAN/FINN | Nguyen et al. (2019a) |

| | | | | | | | | | | |
|---|---|---|---|---|---|---|---|---|---|---|
| 131 | 72, 24 | 30 | AERO6 | CB05 | ACM2 | FNL | † | HTAP v2/MEIC v1.2 | † | Nguyen et al. (2019b) |
| 132 | 12 | 30 | AERO6 | CB05 | ACM2 | FNL | † | MEIC | † | Yoo et al. (2019) |
| 133 | 45 | 30 | AERO6 | CB05 | ACM2 | FNL | MOZART | JEI-DB/INTEX-B | MEGAN/FINN | Sekiguchi et al. (2018) |
| 134 | 36 | 23 | AERO6 | CB05 | ACM2 | FNL | CESM | 2008 MIX | BEIS3/Dust | Hong et al. (2017) |
| 135 | 36 | 23 | AERO6 | CB05 | ACM2 | FNL | † | MEIC | † | Xing et al. (2017) |
| 136 | 108 | 44 | AERO6 | CB05 | ACM2 | FNL | † | EDGAR | † | Xing et al. (2016) |
| 137 | 108 | 44 | AERO6 | CB05 | ACM2 | FNL | † | EDGAR | † | Xing et al. (2015a) |
| 138 | 108 | 44 | AERO6 | CB05 | ACM2 | FNL | † | † | † | Xing et al. (2015b) |
| 139 | 108 | 44 | AERO6 | CB05 | ACM2 | FNL | † | EDGAR | MEGAN/Dust/Sea salt | Xing et al. (2015c) |
| 140 | 36 | 44 | AERO6 | CB05 | ACM2 | FNL | † | † | † | Wang et al. (2014) |
| 141 | 12, 4 | 29 | AERO6 | CB05 | ACM2 | FNL | CESM | 2008 MIX | BEIS3/Dust | Chang et al. (2018) |
| 142 | 36 | 23 | AERO6 | CB05 | ACM2 | FNL | CESM | 2008 MIX | BEIS3/Dust | Hong et al. (2019) |
| 143 | 108 | 44 | AERO6 | CB05 | ACM2 | † | † | † | † | Wang et al. (2017) |
| 144 | † | † | CUACE | RADM2 | MRF | † | † | † | † | Wang H. et al. (2018) |
| 145 | † | † | CUACE | RADM2 | † | † | † | † | † | Wang et al. (2015) |
| 146 | † | † | † | † | † | † | † | † | † | Wang et al. (2013a) |
| 147 | † | † | † | † | † | † | † | † | † | Wang et al. (2013b) |
| 148 | 54 | 24 | CUACE | RADM2 | † | † | † | † | † | Zhou et al. (2012) |
| 149 | † | † | † | † | † | † | † | † | † | Wang et al. (2010) |
| 150 | † | † | † | † | † | † | † | † | † | Zhou et al. (2016) |
| 151 | 45 | 20 | † | CMBZ | † | † | † | † | † | Li J. et al. (2018) |
| 152 | 45, 15, 5 | 28 | † | CBMZ | MYJ | † | MOZART | REAS v2.1 | † | Wang et al. (2014) |
| 153 | 80, 20 | 20 | † | CBMZ | MYJ | † | MOZART | REAS v2.1 | GEIA | Wang et al. (2014) |
| 154 | † | † | † | GATOR | GATOR | † | † | † | † | Ten et al. (2012) |
| 155 | † | † | † | GATOR | GATOR | † | † | † | † | Jacobson et al. (2019) |
| 156 | † | † | † | GATOR | GATOR | † | † | † | † | Jacobson et al. (2015) |
| 157 | † | † | † | † | † | † | † | † | † | Chen et al. (2019a) |
| 158 | † | † | † | † | † | † | † | † | † | Li et al. (2019) |
| 159 | † | † | † | † | † | † | † | † | † | Gao et al. (2018a) |
| 160 | † | † | † | † | † | † | † | † | † | Govardhan et al. (2016) |

†: Unclear; *: A preprint version of this study was available online on October 31, 2019, and was formally published on January 1, 2020.

The following paragraph is added into the newly added Section 3.3 of the revised manuscript.

*"Not only the choice of methodologies for ARI and ACI calculations can impact simulation results, but also the various aspects regarding the setup of modeling studies by applying two-way coupled models. The extra/auxiliary information about model configuration, including horizontal and vertical resolutions, aerosol and gas phase chemical mechanisms, PBL schemes, meteorological and chemical initial conditions (ICs) and boundary conditions (BCs), anthropogenic and natural emissions, were extracted from the 160 papers and presented in Table S4 in Supplement, which is organized in the same order as Table 1.*

*For two-way coupled model applications in Asia, horizontal resolutions were from a few to a hundred kilometers, sometimes with nests, and vertical resolutions from 15 to about 50-70 levels, with one study performed at 100 levels for studying a fog case (Wang Z. L. et al., 2018). Wang K. et al. (2018) evaluated the impacts of horizontal resolutions on simulation results and found out surface meteorological variables were better modeled at finer resolution but no significant improvements of ACI related meteorological variables and certain chemical species between different grid*

*resolutions. Through applying a single column model and then WRF-Chem with ARI, Wang et al. (2019) unraveled that better representation of PBL structure and relevant variables with finer vertical resolution from the surface to PBL top could reduce model biases noticeably, but balancing between vertical resolution and computational resource was important as well. Among the 160 applications of two-way coupled models in Asia, the frequently used aerosol module and gas-phase chemistry mechanism in WRF-CMAQ (WRF-Chem) were AERO6 (MOSAIC and MADE/SOGARM) and CB05 (CBMZ and RADM2), respectively. For PBL schemes, most studies selected YSU in WRF-Chem and ACM2 in WRF-CMAQ. Regarding to meteorological ICs and BCs, the FNL data were the first choice, and outputs from the Model for Ozone and Related Chemical Tracer (MOZART) were used to generate chemical ICs and BCs by most researchers. Georgiou et al. (2018) also unraveled that boundary conditions of dust and $O_3$ played an important role in WRF-Chem simulations. The modeling applications in Asia utilized global (EDGAR), regional (e.g., MIX, INTEX-B, and REAS), and national (e.g., MEIC and JEI-DB) anthropogenic emission inventories. Natural emission sources, such as mineral dust (Shao, 2004), biomass burning (FINN (Wiedinmyer et al., 2011) and GFED (Guido et al., 2010)), biogenic VOCs (MEGAN (Guenther et al., 2006)), and sea salt (Gong et al., 1997) were also considered. It should be noted that only one paper by Gao et al. (2017) reported that the WRF-Chem model with the Gridpoint Statistical Interpolation (GSI) data assimilation could improve the simulation accuracy during a wintertime pollution period."*

Since no study assessing the accuracies of different methodologies in terms of ARI and ACI calculations in two-way coupled models has been conducted in Asia, we added a sentence "*Special attention needs to be paid to assess the accuracies of different methodologies in terms of ARI and ACI calculations in two-way coupled models in Asia and other regions.*" in the Conclusion section of the revised manuscript.

Specific Comments:
(1) Lines 103-108: This sentence is confusing. Are those names of 5 models in the parentheses?
Response: The names in the parentheses are the 5 models reviewed by Zhang (2008). To make the sentence more readable, we deleted the parentheses in this sentence. Now the sentence is "*Zhang (2008) overviewed the developments and applications of five coupled models in the United States (US) and the treatments of chemical and physical processes in these coupled models with emphasis on the ACI related processes.*".

(2) Lines 145-146: This is misleading. While the current versions of WRF is 4.3 and CMAQ 5.3.2, these were not the version used by Wong et al 2012. Those were WRFv3.0 and CMAQv4.7.1.
Response: We deleted the reference and the sentence is revised to "*Different from current released version of WRF-CMAQ model (based on WRF version 4.3 and CMAQ version 5.3.3) that only includes ARI, WRF-Chem with ACI (starting from WRF-Chem version 3.0, Chapman et al., 2009) has been implemented for analyzing the complicated aerosol effects that lead to variations of cloud properties, precipitations and PM2.5 concentrations (Bai et al., 2020; Liu Z. et al., 2018; Park et al., 2018; Zhao et al., 2017).*".

(3) Lines 410-413: I don't understand this sentence. What is accounting for 80% of what? Please clarify.
Response: We rewrote the sentence in Lines 410-413 as follows:

*"Besides the ARI effects of dust, 80 % of the net reductions of $O_3$, $NO_2$, $NO_3$, $N_2O_5$, $HNO_3$, $\cdot OH$, $HO_2\cdot$ and $H_2O_2$ were attributed to the heterogeneous chemistry on dust particles' surface added in WRF-Chem when a springtime dust storm striking the Nanjing megacity of EC (Li M. M. et al., 2017a)."*

(4) Lines 428-432: This sentence is too long and complicated to follow. For example, "enhanced (reduced) radiative forcing at the TOA". The bit in parentheses generally refers to the opposite effect on something. What that something is, is not clear here. Is it reduced atmospheric stability and all the things in the parentheses?

Response: We deleted all the parentheses and now the sentence is "*In the Maritime SEA region, peat and forest fire triggered by El Niño induced drought conditions released huge amount of smoke particles, which promoted dire air pollution problems in the downstream areas, and their ARI effects simulated by WRF-Chem enhanced radiative forcing at the TOA and the atmospheric stability (Ge et al., 2014).*"

(5) Lines 493-496: this sentence does not make sense.

Response: This sentence has now been re-written as follows:

"*As the most important absorbing aerosol, BC induced the largest positive, positive and negative mean DRF at the TOA, in the ATM, and at the surface, respectively, over China during 2006 (Huang et al., 2015).*"

(6) Line 498: "prohibited" is not the right word. Suppressed might be better.

Response: Thanks for the suggestion and "prohibited" is replaced by "suppressed". Now the sentence is "*Ding et al. (2016) and Wang Z. et al. (2018) further applied WRF-Chem with feedbacks to investigate how aerosol-PBL interactions involving BC suppressed the PBL development, which deteriorated air quality in Chinese cities and was described as "dome effect" (namely BC warms the atmosphere and cools the surface, suppresses the PBL development and eventually results in more accumulation of pollutants).*"

(7) Line 545: CA is use here as carbonaceous aerosols and further back as central Asia.

Response: Now we use "CAs" as the abbreviation for carbonaceous aerosols and keep CA for central Asia throughout the revised manuscript.

(8) Line 617-621: This sentence seems self-contradictory. Please clarify.

Response: This sentence is modified to "*With the process analysis methodology in WRF-Chem, Gao J. et al. (2018) indicated that comparing to simulations without BC, the BC and PBL interaction slowed the $O_3$ growth from late morning to early afternoon somewhat before $O_3$ reaching its maximum value at noon due to less vertical mixing in PBL.*"

(9) Line 639: Pool should be Poor.

Response: We have fixed the typo and now the sentence is "*Poor air quality posts risks to human health (Brunekreef and Holgate, 2002; Manisalidis et al., 2020), therefore, in the past several decades, air quality models had been used in epidemiology related research to establish quantitative relationships between concentrations of various pollutants and burden of disease (including mortality or/and morbidity) as well as associated economic loss (Conti et al., 2017).*"

(10) Line 684-686: This sentence is badly worded.

Response: We rewrote this sentence as "*This section provides a summary of model performance by presenting the SI of meteorology and air quality variables as shown in Table S2. These SI were collected from the selected papers that supplying these indices and being defined as papers with SI (PSI) (listed in Tables B2-B3 of Appendix B).*"

(11) Figure 3: Why are there so many more samples for PSI than for ARI and no-ARI?

Response: Samples for PSI included all the relevant statistical indices we found from the selected papers, which could include the evaluations of model simulations with ARI or/and ACI. But the sample size for statistical analysis of model simulations with ARI and without ARI were limited, due to many papers did not report their results differentiating between with and without ARI.

(12) Lines 734-735: It seems from Figure 3 that RH2 has 2 but the SH2 has 6 not 1 PSI with ARI/no-ARI.

Response: In the original manuscript, we deleted the sentence "It should be noted that only 2 or 1 PSI supplying statistical analysis of modeled RH2 and SH2 with/without ARI effects may not be enough to make these comparisons statistically significant and further investigations are much needed." in Lines 734-735 and also deleted "very" in Line 738 to reflect the limited numbers of PSI supplying statistical analysis of modeled RH2 and SH2 with/without ARI effects. Now, it is revised as "*Overall, the modeled RH2 and SH2 were in good agreement with observations with slight over- and under-estimations, respectively, and the limited studies showed that RH2 and SH2 simulated by models with ARI turned on had marginally larger positive biases relative to the results without ARI.*"

(13) Line 742: should be that rather than the

Response: The sentence is modified as "*The meta-analysis also indicated that the most modeled WS10 tended to be overestimated (81 % of the samples) with the average MB value of 0.79 $m·s^{-1}$, and the mean RMSE value was 2.76 $m·s^{-1}$.*"

(14) Line 747: Figure 3 say 9 and 10 PSI with ARI/no-ARI, not 5.

Response: The sentence now reads as "*The PSI with ARI effects suggested that the correlation of wind speed was slightly improved (mean R from 0.56 to 0.57) and the average RMSE and positive MB decreased by 0.003 $m·s^{-1}$ and 0.051 $m·s^{-1}$, respectively (Fig. 3h).*"

(15) Section 5.1.2: I think this analysis needs more explanation. Were these different studies of different lengths where the PSI were grouped according temporal scale? Is daily scale, PSI simulations that only lasted one day? I don't see the significance of this analysis.

Response: The model simulations and statistical indices from the PSI were on different time scales so that we did the meta-analysis and grouped SI according to annual, seasonal, monthly, and daily scales. Even though some model simulations lasted more than one day, we classified the statistical indices as daily scale as long as they were reported daily from the relevant PSI. As mentioned before, we move Section 5.1.2 and Section 5.2.2 to Appendix C of the revised manuscript to improve the paper's readability, but intend to provide more detailed information about the model performances at different temporal scales.

(16) Section 5.1.3: This section is also of questionable value. The meteorological performance of these models is more related to the physics options, FDDA, initial and boundary conditions, resolution, domain, time period, etc, of the WRF setup than whether it is WRF-Chem or WRF-CMAQ. The meteorology performance is due to WRF not Chem or CMAQ parts.

Response: We agree that many factors can affect meteorological performance of two-way coupled models and add Table S4 in Supplement and Section 3.3 to summarize the limited evaluations towards the effects of different aspects of model setup on model performance. However, inter-comparisons of different models are extremely valuable even though many aspects of model setup are not the same, which is demonstrated in the coordinated studies such as AQMEII and MICS-Asia and also in the last paragraph of Section 3.1 (Lines 273-280 in the revised manuscript). Figure 3 (e-h) indicates surface meteorological variables can be affected by aerosol feedbacks and Section 5.1.3 of original manuscript (now it is Section 5.1.2 in the revised manuscript) serves as a critical part of our overview and meta-analysis to reveal how turning on aerosol feedbacks impact model performance of meteorological variables in different two-way coupled models.

(17) Line 974: When reporting daily results are these day and night together?

Response: When PSI presented daily SI, we categorized them as "daily" that should include results during day and night together. On the other hand, hourly results reported by PSI during day or night time were put into the "hourly" category.

(18) Lines 1018-1020: This sentence is unclear. Which effect increased (decreased)?

Response: This sentence is revised as "*Under the high emission levels as well as at slightly different humidity levels of RH > 85 % with increasing emissions, the ACI effects of anthropogenic aerosols induced precipitation increase in the MRYR area of China. Over the same area, precipitation decreased due to the ACI effects of anthropogenic aerosols with the low emission levels and RH < 80 %.*"

(19) Lines 1020-1022: Again, doesn't make sense. Trying to say too much in single sentences.

Response: We rewrite this sentence as "*In PRD, wintertime precipitation was enhanced by the ACI effects of anthropogenic aerosols but inhibited by ARI. In SK, summertime precipitation was both enhanced and inhabited by the ACI and ARI effects of anthropogenic aerosols.*"

(20) Lines 1056: what increase (decrease)?

Response: The whole sentence is revised as "*Simulation results showed that turning on aerosol feedbacks in coupled models generally made $PM_{2.5}$ concentrations increased in different regions of Asia at various time scales, which stemmed from decrease of shortwave radiation, T2, WS10 and PBLH and increase of RH2.*"

(21) Lines 1079-1081: Way too many parentheses constructs. Can't follow.

Response: We rewrite this sentence as "*The seasonal $SO_2$ reduction was rather large, which related to higher PBLH induced by the ACI effects of dust aerosols in the NCP area of EA (Wang K. et al., 2018). The slight increase of seasonal $SO_2$ was reported in the whole domain of EA due to lower PBLH caused by ARI effects of anthropogenic aerosols (Nguyen et al., 2019b).*"

(22) Line 1108: severe rather than server?
Response: The typo is corrected.

---

## Author Comment (AC2)

On behalf of all the co-authors, we want to express our sincere gratitude to Reviewer 2 for providing very thoughtful comments and valuable suggestions regarding our manuscript. Following these comments and suggestions, we revised and reorganized the manuscript to improve its quality. Following the Reviewer's comments in black, please find our point-to-point responses in blue. The new texts in the revised manuscript are in blue and italic.

General Comments:
Review of "Two-way coupled meteorology and air quality models in Asia: a systematic review and meta-analysis of impacts of aerosol feedbacks on meteorology and air quality," by Gao et al., submitted to Atmospheric Chemistry and Physics Discussions.

This paper reviews air coupled meteorology-air quality models applied to Asia. It is quite detailed, almost too much in parts of it. It could benefit from more organization, better figure captions, and more specific conclusions. Below are some additional comments.

Response: To improve the paper's organization, we changed the title of Section 3 to "Basic overview", added a new Section 3.3 (Summary of modeling methodologies), and moved Section 5.1.2 and Section 5.2.2 to Appendix C in the revised manuscript. With respect to figure captions as well as table captions, we went through the manuscript and revised them accordingly. All the revisions of captions are listed in our response to the sixth comment. In the conclusion section, we refined several takeaways of our study and strengths and limitations of two-way coupled models, and the corresponding response is detailed in the ninth comment.

All responses to the additional comments are:

1. Introduction. "Online models or coupled models are designed and developed to consider the two-way feedbacks and attempted to accurately simulate both meteorology and air quality." It seems that this would be a good place to identify the origin of such models. According to Zhang (2008), the GATOR-GCMOM model is "the first fully-coupled online model in the history that accounts for all major feedbacks among major atmospheric processes based on first principles (Jacobson, 1994, 1997; Jacobson et al., 1996)."

Response: According to this suggestion, in Introduction section we added "*As Zhang (2008) pointed out, Jacobson (1994, 1997) and Jacobson et al. (1996) pioneered the development of a fully-coupled model named Gas, Aerosol, Transport, Radiation, General Circulation, Mesoscale, and Ocean Model (GATOR-GCMOM) in order to investigate all the processes related to ARI and ACI.*" before "Currently, there are three representative two-way coupled meteorology and air quality models, namely the Weather Research and Forecasting-Chemistry (WRF-Chem) (Grell et al., 2005), WRF coupled with Community Multiscale Air Quality (CMAQ) (Wong et al., 2012) and WRF coupled with a multi-scale chemistry-transport model for atmospheric composition analysis and forecast (WRF-CHIMERE) (Briant et al., 2017)".

2. Introduction. "Currently, there are three representative two-way coupled meteorology and air quality models." What does that mean? There are several more two-way coupled meteorology and air quality models, as cited later in the paragraph.

Response: To be more precise, the sentence is changed to "*Currently, there are three open-sourced two-way coupled meteorology and air quality models.*"

3. Introduction. Another coupled air quality-meteorological model used in Asia is GATOR-GCMOM. Its applications have included a study of the local and global fate of radionuclides from Fukushima (Ten Hoeve and Jacobson, 2012), where the model was run in both nested and global mode, and studies of the impact of urbanization in Beijing (Jacobson et al., 2015) and New Delhi (Jacobson et al., 2019) on air quality and meteorology. It seems that these papers meet the criteria listed.

Response: Thanks for providing this helpful information. We added relevant information into our revised manuscript and the details are listed as follows:

[revised manuscript text omitted]

(14). Line 1334: Adding information involving *GATOR-GCMOM* in Figure A1.

[Figure]

*Figure A1. Flowchart of literature search and identification*

(15). Lines 1340-1341: Adding information concerning *GATOR-GCMOM* in Table B1.

4. The discussion could be improved by identifying how different models treat aerosol size and composition. Do they use lognormal modes or discrete size sections. How many size distributions in either case are treated? What aerosol physical processes are treated? Coagulation? Condensation/evaporation? Internal-aerosol thermodynamic equilibrium? Hydration?

Response: We absolutely agree that it would improve the scientific quality of our manuscript by adding more detailed information and discussion about how aerosol size and composition are treated in two-way coupled models. In the new added Table 5 of the revised manuscript, we listed the methodologies representing aerosol composition and aerosol size distribution in different aerosol mechanisms. This table is also to response the comment by Reviewer 1 about how ARI and ACI being calculated in two-way coupled models.

Regarding to the questions raised here, we searched relevant papers through Google scholar and Web of Science and found three important review papers. Zhang (2008) and Baklanov et al. (2014) had systematically reviewed how aerosol size and composition were treated in two-way coupled models before 2013. Stevens and Dastoor (2019) outlined representations of aerosol mixing state and size distribution in 39 aerosol modules used in all available atmospheric models. Based on the thorough summary listed in Table 1 of Stevens and Dastoor (2019), we further dug out more detailed numerical settings of aerosol size distribution (namely, geometric diameter and standard deviation for modal approach or bin ranges for sectional method) in the five two-way coupled models used in Asia and compiled them in a new Table 3 in our revised manuscript. Please note that the values were extracted from published papers

or/and source codes in different versions of these five models.

We added a new Section 3.3 titled "Summary of modeling methodologies" with the following contents:

"*How accurately ARI and ACI are simulated also rely on the representation of aerosol size distribution and composition in two-way coupled models. Three typical approaches (bulk, modal and sectional methods) are adopted by the five two-way coupled models and WRF-Chem offers all the three approaches, but other models only support one specific option. The Global Ozone Chemistry Aerosol Radiation and Transport (GOCART) model (Ginoux et al., 2001) in WRF-Chem is the only one that is based on a combination of bulk (for water, BC, OC, and sulfate aerosols) and sectional (for dust and sea salt aerosols) approaches. In the five two-way coupled models applied in Asia, modal and sectional approaches are widely used and their detailed numerical settings of aerosol size distribution (namely, geometric diameter and standard deviation for modal approach or bin ranges for sectional method) and the corresponding aerosol compositions are compiled in Table 3. Regarding the modal method, same parameter values for Aitken and accumulation modes and geometric diameters for coarse mode in the latest version of WRF-Chem (v4.3.3) and older version of WRF-CMAQ (before v5.2) are set as default, except the standard deviations for coarse mode are slightly different. In the official version of WRF-CMAQ released after v5.2, there are some modifications to the default setting of geometric diameters in Aitken, accumulation and coarse modes, from 0.010 to 0.015 µm, 0.070 to 0.080 µm and 1.00 to 0.600 µm, respectively. For the GRAPES-CUACE model, the geometric diameters and standard deviations for certain aerosol species in the accumulation mode were updated from its older version (Zhou et al., 2012) to newer one (Zhang et al., 2021). With respect to the sectional approach, 4 or 8 (from 0.039 to 10 µm), 12 (from 0.005 to 20.48 µm) and 14 (from 0.002 to 50 µm) particle size bins are defined in WRF-Chem, CUACE and GATOR-GCMOM, respectively. As shown in Tables 2 and 3, GATOR-GCMOM considered 47 aerosol species, and others coupled models adopted different numbers of species groups (such as 6, 5, 7, 8 aerosol species groups in WRF-Chem, CMAQ, NAQPMS and CUACE, respectively). Recently, more studies with two-way coupled models focused on aerosol feedbacks of light-absorbing aerosols, especially BrC emitted from BB (Jiang et al., 2012; Yao et al., 2017; Simeon et al., 2021). Some observational studies had applied the single particle soot photometer to investigate the optical properties of tarball particles released from BB (Adachi et al., 2019; Corbin et al., 2019; Yuan et al., 2021), but only GATOR-GCMOM had taken tarballs into account as a specific component. In addition, mineralogical compositions of dust aerosols were incorporated in a specific version of WRF-Chem (Li and Sokolik, 2018) to explore their ARI effects (Li and Sokolik, 2018).*"

*Table 5. Summary of numerical representations of aerosol size distribution and composition in two-way coupled models applied in Asia.*

| Model | Aerosol mechanism | Modal approach | | | | | | Compositions | Reference |
|---|---|---|---|---|---|---|---|---|---|
| | | Aitken | | Accumulation | | Coarse | | | |
| | | Geometric diameters (µm) | Standard deviations (µm) | Geometric diameters (µm) | Standard deviations (µm) | Geometric diameters (µm) | Standard deviations (µm) | | |

| Model | Scheme | | | | | | | Components | Reference |
|---|---|---|---|---|---|---|---|---|---|
| WRF-Chem v4.3.3 | MADE/SORGAM | 0.010 | 1.7 | 0.07 | 2.0 | 1.0 | 2.5 | Water, BC, OC, and sulfate, dust and sea salt | WRF-Chem codes※ |
| WRF-Chem | MAM3 | 0.013 (sulfate and secondary OM) | 1.6 (sulfate and secondary OM) | 0.068 (sulfate, secondary OM, primary OM, BC, dust and sea salt) | 1.8 (sulfate, secondary OM, primary OM, BC, dust and sea salt) | 2.0 (sea salt), 1.0 (dust) | 1.8 (sea salt and dust) | Sulfate, methane sulfonic acid (MSA), OM, BC, sea salt and dust | Easter et al. (2004) Liu et al. (2012) |
| | MAM7 | 0.013 (sulfate and secondary OM and BC) | 1.6 (sulfate, OM and BC) | 0.068 (sulfate and BC) 0.068 (primary OM) 0.2 (sea salt) 0.11 (dust) | 1.8 (sulfate and BC) 1.6 (primary OM) 1.8 (sea salt) 1.8 (dust) | 2.0 (sea salt) 1.0 (dust) | 2.0 (sea salt) 1.8 (
[revised manuscript text omitted]

5. How are clouds treated in the models? Are they treated with lognormal modes or discrete size distributions or without size information? How do clouds interact with aerosol particles?

Response: To address the questions raised here, we added a new Table 3 and a paragraph about how clouds properties and aerosol-cloud interactions are represented in coupled models in Section 3.3 of our revised manuscript as follows:

[revised manuscript text omitted]
 literature." Please identify exactly which models are included and where the results are applicable to in the figure caption. Same with other captions.

Response: Thank you for your suggestion and we rewrote the captions of Figure 3, Figure 4, Figure 5, Figure 6, Figure 7, Figure 8, Table 6, Table 7, Table 8, Table B2, Table B3, Table B4 and Table B5 in the revised manuscript as follows:

Caption of Figure 3 is revised as "*Figure 3. Quantile distributions of R, MB and RMSE for simulated surface meteorological variables by the five coupled models (WRF-Chem, WRF-CMAQ, GRAPES-CUACE, WRF-NAQPMS and GATOR-GCMOM) (a-d) and comparisons of statistical indices with/out ARI (e-h) in Asia.*"

Caption of Figure 4 is revised as "*Figure 4. Quantile distributions of the statistical indices for simulated surface meteorological variables by WRF-Chem, WRF-CMAQ, GRAPES-CUACE, WRF-NAQPMS and GATOR-GCMOM in Asia.*"

Caption of Figure 5 is revised as "*Figure 5. Quantile distributions of statistical indices for simulated $PM_{2.5}$ and $O_3$ (a-b) by the five two-way coupled models (WRF-Chem, WRF-CMAQ, GRAPES-CUACE, WRF-NAQPMS and GATOR-GCMOM) and comparisons of statistical indices with/out ARI (c-d) in Asia.*"

Caption of Figure 6 is revised as "*Figure 6. Quantile distributions of R, MB and RMSE of $PM_{2.5}$ and $O_3$ simulated by WRF-Chem, WRF-CMAQ, GRAPES-CUACE, WRF-NAQPMS and GATOR-GCMOM in Asia.*"

Caption of Figure 7 is revised as "*Figure 7. Variations of shortwave and longwave radiative forcing (SWRF and LWRF) simulated by two-way coupled models (WRF-Chem, WRF-CMAQ, GRAPES-CUACE, WRF-NAQPMS and GATOR-GCMOM) with aerosol feedbacks at the bottom and top of atmosphere (BOT and TOA), and in the atmosphere (ATM) in Asia.*"

Caption of Figure 8 is revised as "*Figure 8. Responses of shortwave radiation forcing to aerosol feedbacks in different areas/periods in Asia (a) and the inter-regional comparisons of its variations in Asia, Europe and North America (b).*"

Caption of Table 6 is revised as "*Table 4. Summary of variations of surface meteorological variables and planetary boundary layer height (PBLH) caused by aerosol feedbacks simulated by two-way coupled models (WRF-Chem, WRF-CMAQ, GRAPES-CUACE, WRF-NAQPMS and GATOR-GCMOM) in different regions of Asia and at different temporal scales.*"

Caption of Table 7 is revised as "*Table 5. Summary of changes of cloud properties and precipitation characteristics due to aerosol feedbacks simulated by two-way*

*coupled models (WRF-Chem, WRF-CMAQ, GRAPES-CUACE, WRF-NAQPMS and GATOR-GCMOM) in Asia.*"

Caption of Table 8 is revised as "*Table 6. Compilation of aerosol-induced variations of $PM_{2.5}$ and gaseous pollutants simulated by two-way coupled models (WRF-Chem, WRF-CMAQ, GRAPES-CUACE, WRF-NAQPMS and GATOR-GCMOM) in different regions of Asia and at different temporal scales.*"

Caption of Table B2 is revised as "*Table B2. The compiled number of publications (NP) and number of samples (NS) for papers that providing statistical indices (SI) of meteorological variables.*"

Caption of Table B3 is revised as "*Table B3. The compiled number of publications (NP) and number of samples (NS) for papers that providing statistical indices (SI) of air quality variables.*"

Caption of Table B4 was revised as "*Table B4. The compiled number of publications (NP) and number of samples (NS) for papers that simultaneously providing the statistical indices (SI) of meteorological variables simulated by coupled models (WRF-Chem, WRF-CMAQ, GRAPES-CUACE, WRF-NAQPMS and GATOR-GCMOM) with/out ARI.*"

Caption of Table B5 was revised as "*Table B5. The compiled number of publications (NP) and number of samples (NS) for papers that simultaneously providing the statistical indices (SI) of air quality variables simulated by coupled models (WRF-Chem, WRF-CMAQ, GRAPES-CUACE, WRF-NAQPMS and GATOR-GCMOM) with/out ARI.*"

7. In the figures, it would be useful to know what the overall mean percent error is in addition to the absolute errors

Response: We agree that it would be useful to add the overall mean percent errors in our figures depicting statistical indices, but we can only find very limited studies reporting this kind of information. According to our compiled data, there were only 13 studies reporting normalized mean error (NME) (%) of surface meteorological and air quality variables simulated by two-way coupled models (WRF-Chem and WRF-CMAQ) in Asia, which is summarized in Table B7 of our revised manuscript. It should be noted that no NME of meteorological variables simulated by two-way coupled models with and without enabling the ARI effects was mentioned in these studies. To reflect this additional information towards the meta-analysis, we also add two new paragraphs in Section 5.1.1 and Section 5.2.1 of the revised manuscript, respectively, as follows:

For meteorological variables in Section 5.1.1: "*Besides the SI discussed above, very limited papers reported the normalized mean error (NME) (%) of surface meteorological variables (T2, SH2, RH2 and WS10) simulated by two-way coupled* models (WRF-Chem and WRF-CMAQ) *in Asia, which is summarized in Table B7 of Appendix B. The evaluations with two-way coupled models in Asia showed that the overall mean percent errors of T2, SH2, RH2 and WS10 were 22.71%, 10.32%, 13.94%, and 51.28%, respectively. The ranges of NME (%) values were quite wide for T2 (from -0.48 to 270.20 %) and WS10 (from 0.33 to 112.28%) reported by the limited studies.*

*Note that no NME of surface meteorological variables simulated by two-way coupled models simultaneously with and without enabling the ARI effects was mentioned in these studies*."

For air quality variables in Section 5.2.1: "*In addition to the SI analyzed above and similar to the surface meteorological variables, the NME (%) of $PM_{2.5}$ and $O_3$ is listed in Table B7. The limited studies with WRF-Chem and WRF-CMAQ indicated that the overall mean percent errors of $PM_{2.5}$ and $O_3$ were 47.63% (from 29.55 to 104.70 %) and 43.03% (from 21.10 to 127.00 %), respectively. With the ARI effects enabled in WRF-Chem in different seasons over the China domain, the NME (%) of $PM_{2.5}$ increased slightly during most seasons, except during a spring month with little change (Zhang et al., 2018). Another study by Nguyen et al. (2019b) revealed that the NME (%) of $PM_{2.5}$ and $O_3$ simulated by WRF-CMAQ became a little worse in SEA comparing to the simulations without ARI.*"

*Table B7. Summary of normalized mean error (NME) (%) of surface meteorological and air quality variables using two-way coupled models (WRF-Chem and WRF-CMAQ).*

| T2 | SH2 | RH2 | WS10 | PM₂.₅ | O₃ | PM₂.₅ with ARI (ARI) or without ARI (NO) | O₃ with ARI (ARI) or without ARI (NO) | Model | Region | Reference |
|---|---|---|---|---|---|---|---|---|---|---|
| | | | | | 23.60, 38.50, 55.70, 39.80 | | | WRF-Chem | EA | Liu X. et al. (2016) |
| 0.80, 0.60, 0.60, 0.60 | | 19.10, 16.50, 10.00, 10.10 | 58.90, 41.60, 44.90, 49.50 | 37.31, 37.61, 35.77, 34.69, 35.34, 35.41, 45.22, 44.33, 43.09, 39.29, 39.49, 39.07 | | 37.61, 35.34, 44.33, 39.49 (ARI) 35.77, 35.41, 43.09, 39.07 (NO) | | WRF-Chem | China | Zhang et al. (2018) |
| 270.20, 22.30, 12.50, 17.60 | | | | | | | | WRF-Chem | EA | Zhang Yang et al. (2016a) |
| | | | | 44.99, 29.55, 37.28 | | | | WRF-Chem | NCP | Yang et al. (2015) |
| 15.50, 15.80, 13.90, 9.90 | 10.40, 10.40, 9.90, 9.90 | | 31.30, 31.30, 32.50, 32.50 | 49.80, 65.30, 49.80, 65.60, 88.30, 56.90, 88.40, 57.00 | 127.00, 32.20, 25.40, 126.10, 32.10, 25.00, 79.90, 25.80, 21.40, 45.80, 77.90, 25.60, 21.10, 39.50 | | | WRF-Chem | EA | Zhang Y. et al. (2015a) |
| 14 | 11 | | 32 | 52.70, 58.00, 104.70, 62.00 | 87.50, 28.60, 23.30, 52.90, 32.40, 28.20 | | | WRF-Chem | EA | Chen Y. et al. (2015) |
| -0.48, 0.19, 0.21, 0.05, 0.08, 0.13, 0.05, 0.04, 0.04, 0.05, 0.02, 0.02, 0.06, 0.05, 0.04, 0.02 | | | 0.33, 1.92, 0.71, 0.78, 0.28, 1.72, 0.61, 0.64, 0.24, 1.76, 0.00, 0.45, 0.34, 1.29, 0.44, 0.56 | | | | | WRF-Chem | NCP | Chen D. et al. (2015) |
| 16.60, 10.50, 8.90, 12.90, 10.50, 10.20 | | | | | | | | WRF-Chem | EA | Wang K. et al. (2018) |
| 6.52, 6.58 | | 15.76, 12.15 | 112.28, 97.26 | | | | | WRF-Chem | NEA | Park et al. (2018) |
| | | | | 36.00, 33.00 | 31.00, 22.00 | | | WRF-Chem | China | Zhao et al. (2017) |
| | | | | 44.00, 44.60, 40.10, 54.30 | | | | WRF-Chem | NCP | Gao M. et al. (2015) |
| | | | | 41.48, 41.00, 51.77, 55.70 | 26.68, 26.71, 34.43, 34.64 | 41.00, 55.70 (ARI) 41.48, 51.77 (NO) | 26.71, 34.64 (ARI) 26.68, 34.43 (NO) | WRF-CMAQ | SEA | Nguyen et al. (2019b) |
| | | | | 37.99, 35.06, 38.59, 35.44, 34.39 | | | | WRF-CMAQ | China | Chang (2018) |

8. Figure 9. Please provide details of the models used and the region covered.
Response: In the revised manuscript, Figure 9 becomes Figure 7. The caption of Figure 7 is revised to "*Figure 7. Variations of shortwave and longwave radiative forcing (SWRF and LWRF) simulated by two-way coupled models (WRF-Chem, WRF-CMAQ, GRAPES-CUACE, WRF-NAQPMS and GATOR-GCMOM) with aerosol feedbacks at the bottom and top of atmosphere (BOT and TOA), and in the atmosphere (ATM) in Asia.*" As per the reviewer's suggestion, a new table is added in Supplement as Table S5 in our revised manuscript and organized in the same order as Table 1, to illustrate

the detailed information about the variations of SWRF and LWRF generated by which model and in which region/area in Asia. In addition, we revise the sentence in Lines 1087-1089 of the revised manuscript to "*Figure 7 presents the variations of simulated SWRF and LWRF at the bottom (BOT) and TOA and in the ATM due to aerosol feedbacks, and detailed information of these variations are compiled in Table S5 of Supplement.*"

*Table S5. Summary of aerosol-induced variations of simulated shortwave and longwave radiative forcing (SWRF and LWRF) at the bottom and top of atmosphere (BOT and TOA) and in the atmosphere (ATM) in Asia.*

| No. | ΔSWRF at BOT (W/m²) | ΔLWRF at BOT (W/m²) | ΔSWRF in ATM (W/m²) | ΔLWRF in ATM (W/m²) | ΔSWRF at TOA (W/m²) | ΔLWRF at TOA (W/m²) | Model | Region | Reference |
|---|---|---|---|---|---|---|---|---|---|
| 1 | -8.05, -6.07, -0.45, -1.34 | -0.28, -0.1, -0.02, -0.06 | † | † | † | -1.91, -0.52, -0.48, -0.50 | WRF-Chem | India | Singh et al. (2020)* |
| 2 | † | † | † | † | † | † | WRF-Chem | India | Bharali et al. (2019) |
| 3 | † | † | † | † | † | † | WRF-Chem | India | Shahid et al. (2019) |
| 4 | -73.71 | † | † | † | † | † | WRF-Chem | NCP | Wang et al. (2019) |
| 5 | † | † | † | † | † | † | WRF-Chem | NCP | Wu et al. (2019a) |
| 6 | † | † | † | † | † | † | WRF-Chem | NCP | Wu et al. (2019b) |
| 7 | † | † | † | † | † | † | WRF-Chem | NWC | Yuan et al. (2019) |
| 8 | -40.6, -82.2, -38.4, -49.9 | † | † | † | † | † | WRF-Chem | NCP | Zhang et al. (2019) |
| 9 | -38 | † | † | † | † | † | WRF-Chem | NCP | Zhou et al. (2019) |
| 10 | -19.3 | † | † | † | -14.2 | † | WRF-Chem | WA | Bran et al. (2018) |
| 11 | † | † | † | +0.86, +1.21 | -3.07, -4.39 | † | WRF-Chem | China & India | Gao et al. (2018b) |
| 12 | -8.4 | † | † | † | † | † | WRF-Chem | CA | Li M. M. et al. (2018) |
| 13 | -83.4, -91.4, -116.3, -82.9, -95.6, -139.1 | +39, +45, +26.8, +38.6, +39.1, +26.8 | +68.9, +82.3, +127.5, +67.8, +88.9, +164.8 | -32.5, -36.4, -21.2, -32.2, -31.5, -21 | -14.5, -9.1, +11.2, -15, -6.7, +25.7 | +6.5, +8.6, +5.5, +6.4, +7.6, +5.7 | WRF-Chem | YRD | Li and Sokolik (2018) |
| 14 | -69 | † | † | † | † | † | WRF-Chem | NCP | Liu et al. (2018) |
| 15 | † | † | † | † | † | † | WRF-Chem | NCP | Miao et al. (2018) |
| 16 | -16.20, -14.86, -13.25, -12.74 | +5.78, +5.29, +2.45, +2.52 | +20.20, +21.00, +17.06, +19.07 | -1.84, -4.26, +0.36, -1.80 | +4.00, +6.14, +3.80, +6.34 | +3.94, +1.03, +2.82, +0.72 | WRF-Chem | India | Soni et al. (2018) |
| 17 | † | † | † | † | † | † | WRF-Chem | NCP | Wang L. T. et al. (2018) |
| 18 | -5.9 | † | † | † | † | † | WRF-Chem | EC | Wang Z. L. et al. (2018) |
| 19 | † | † | -2, +2 | † | † | † | WRF-Chem | TP | Yang et al. (2018) |
| 20 | † | † | † | † | † | † | WRF-Chem | EA | Zhou et al. (2018) |
| 21 | † | † | † | † | † | † | WRF-Chem | EC | Gao et al. (2017c) |
| 22 | -52.3 | † | † | † | † | † | WRF-Chem | YRD | Li et al. (2017a) |
| 23 | -130 | † | † | † | † | † | WRF-Chem | YRD | Li et al. (2017b) |
| 24 | -54.6, -18, -36.1 | † | † | † | † | † | WRF-Chem | NCP | Qiu et al. (2017) |
| 25 | † | † | † | † | † | † | WRF-Chem | NCP | Yang and Liu (2017a) |
| 26 | † | † | † | † | † | † | WRF-Chem | NCP | Yang and Liu (2017b) |
| 27 | † | † | † | † | +0.79 | † | WRF-Chem | EC | Yao et al. (2013) |
| 28 | † | † | † | † | † | † | WRF-Chem | SEC | Zhan et al. (2017) |
| 29 | -9.3, -14.2, -11.7 | † | +6.3, +9.3, +6.3 | † | -3, -4.9, -5.4 | † | WRF-Chem | India | Feng et al. (2016) |
| 30 | † | † | † | † | † | † | WRF-Chem | NCP | Gao et al. (2016b) |
| 31 | -6.5, -8.3, -12.1, -8.5 | † | † | † | † | † | WRF-Chem | EA | Liu et al. (2016) |
| 32 | -21.1, -13.1 | † | +12.7, +4.8 | † | † | † | WRF-Chem | NCP | Liu et al. (2016) |
| 33 | † | † | † | † | † | † | WRF-Chem | NCP | Miao et al. (2016) |

| | | | | | | | | | |
|---|---|---|---|---|---|---|---|---|---|
| 34 | -20, -30.8, -27.1, -25.8, -22.8 | † | † | † | † | † | WRF-Chem | EA | Wang et al. (2016) |
| 35 | † | † | † | † | † | † | WRF-Chem | NWC | Yang et al. (2016) |
| 36 | † | † | † | † | † | † | WRF-Chem | EA | Zhong et al. (2016) |
| 37 | † | † | † | † | † | † | WRF-Chem | India | Govardhan et al. (2015) |
| 38 | -10.2, -12.6, -7.5, -3.3, -4.8 | † | † | † | † | † | WRF-Chem | China | Huang et al. (2015) |
| 39 | † | † | † | † | † | † | WRF-Chem | EA | Wang et al. (2015) |
| 40 | -14, -10 | † | +2, +9 | † | -5, -8 | † | WRF-Chem | EA | Chen et al. (2014) |
| 41 | -10.6, -2.9, -3.2 | † | +4.2, +4.6, +0.4 | † | -6.5, +1.7, -2.8 | † | WRF-Chem | SEA | Gao et al. (2014) |
| 42 | † | † | † | † | +20 | † | WRF-Chem | India | Ge et al. (2014) |
| 43 | -8 | † | +5.1 | † | -2.9 | † | WRF-Chem | NCP | Kumar et al. (2014) |
| 44 | † | † | † | † | † | † | WRF-Chem | EA | Li et al. (2014) |
| 45 | -30.93 | +4.08 | +25.45 | -3.34 | -5.48 | +0.74 | WRF-Chem | NWC | Lin et al. (2014) |
| 46 | -5.58 | † | +1.61 | † | -3.97 | † | WRF-Chem | India | Chen et al. (2013) |
| 47 | † | † | † | † | † | † | WRF-Chem | India | Dipu et al. (2013) |
| 48 | † | † | † | † | † | † | WRF-Chem | India | Kumar et al. (2012a) |
| 49 | † | † | † | † | † | † | WRF-Chem | India | Kumar et al. (2012b) |
| 50 | -30 | † | † | † | † | † | WRF-Chem | China | Seethala et al. (2011) |
| 51 | † | † | † | † | +0.75, +1.024, +5.5, +7 | † | WRF-Chem | PRD | Zhuang et al. (2011) |
| 52 | † | † | † | † | † | † | WRF-Chem | NCP | Liu et al. (2020)* |
| 53 | † | † | † | † | † | † | WRF-Chem | EC | Jia et al. (2019) |
| 54 | † | † | † | † | † | † | WRF-Chem | China | Wang et al. (2019) |
| 55 | † | † | † | | +0.45, +1.04, +0.89, +1.77, -0.13, +0.05 | +0.04, +0.18, +0.05, +0.20, +0.04, +0.15 | WRF-Chem | YRD | Nicholls et al. (2019) |
| 56 | † | † | † | † | † | † | WRF-Chem | India | Li et al. (2019) |
| 57 | † | † | † | † | † | † | WRF-Chem | India | Kedia et al. (2019a) |
| 58 | † | † | † | † | † | † | WRF-Chem | PRD | Kedia et al. (2019b) |
| 59 | † | † | † | † | † | † | WRF-Chem | EC | Huang et al. (2019) |
| 60 | -25, -75 | † | † | † | † | † | WRF-Chem | NCP | Ding et al. (2019) |
| 61 | † | † | † | † | † | † | WRF-Chem | EA | An et al. (2019) |
| 62 | -7.74 | † | † | † | † | † | WRF-Chem | MRYR | Liu et al. (2018) |
| 63 | † | † | † | † | -5.38 | † | WRF-Chem | PRD | Liu et al. (2018) |
| 64 | † | † | † | † | † | † | WRF-Chem | China | Zhang et al. (2018) |
| 65 | † | † | † | † | † | † | WRF-Chem | YRD | Gao et al. (2018) |
| 66 | -6.8 | † | † | † | † | † | WRF-Chem | EA | Zhang et al. (2017) |
| 67 | † | † | † | † | † | † | WRF-Chem | EC | Wu et al. (2017) |
| 68 | -88 | † | † | † | † | † | WRF-Chem | YRD | Sun et al. (2017) |
| 69 | † | † | † | † | † | † | WRF-Chem | YRD | Zhong et al. (2017) |
| 70 | -29.9 | † | +27.0 | † | -2.9 | † | WRF-Chem | NCP | Gao et al. (2017a) |
| 71 | † | † | † | † | † | † | WRF-Chem | NCP | Gao et al. (2017b) |
| 72 | -21.9, -29.1, -14.6, -12.1, -14.8, -21.5, -10.6 | † | † | † | † | † | WRF-Chem | China | Ma et al. (2017) |
| 73 | † | † | † | † | † | † | WRF-Chem | India | Lau et al. (2017) |
| 74 | † | † | † | † | † | † | WRF-Chem | NCP | Kajino et al. (2017) |
| 75 | † | † | † | † | † | † | WRF-Chem | TP & India | Yang et al. (2017) |
| 76 | † | † | † | † | † | † | WRF-Chem | EA | He et al. (2017) |

| # | | | | | | | Model | Region | Reference |
|---|---|---|---|---|---|---|---|---|---|
| 77 | † | † | † | † | † | † | WRF-Chem | YRD | Campbell et al. (2017) |
| 78 | † | † | † | † | † | † | WRF-Chem | EC | Zhang et al. (2016) |
| 79 | -9.12, -8.53, -10.94, -11.23 | † | † | † | † | † | WRF-Chem | China | Ma et al. (2016) |
| 80 | † | † | † | † | † | † | WRF-Chem | EC | Zhang et al. (2016a) |
| 81 | -7.1, -9.8, -11.7, -7.8 | † | † | † | † | † | WRF-Chem | EC | Zhang et al. (2016b) |
| 82 | -45.5 | † | † | † | +14.9 | † | WRF-Chem | EC | Huang et al. (2016) |
| 83 | † | † | † | † | † | † | WRF-Chem | YRD | Xie et al. (2016) |
| 84 | † | † | † | † | † | † | WRF-Chem | India | Srinivas et al. (2016) |
| 85 | † | † | † | † | † | † | WRF-Chem | India | Kedia et al. (2016) |
| 86 | † | † | † | † | † | † | WRF-Chem | India | Jin et al. (2016a) |
| 87 | † | † | † | † | † | † | WRF-Chem | India | Jin et al. (2016b) |
| 88 | † | † | † | † | † | † | WRF-Chem | NCP | Gao et al. (2016a) |
| 89 | -58, -115 | -10 | † | † | † | † | WRF-Chem | NCP | Gao et al. (2016) |
| 90 | † | † | † | † | † | † | WRF-Chem | EC | Ding et al. (2016) |
| 91 | -26.51 | † | † | † | † | † | WRF-Chem | NCP | Yang et al. (2015) |
| 92 | -18.15, -18.50, -17.64, -23.15 | † | † | † | † | † | WRF-Chem | NCP | Shen et al. (2015) |
| 93 | † | † | † | † | † | † | WRF-Chem | EA | Zhang et al. (2015a) |
| 94 | † | † | † | † | † | † | WRF-Chem | EA | Chen et al. (2015) |
| 95 | † | † | † | † | † | † | WRF-Chem | NCP | Zhong et al. (2015) |
| 96 | † | † | † | † | † | † | WRF-Chem | India | Jin et al. (2015) |
| 97 | † | † | † | † | † | † | WRF-Chem | India | Jena et al. (2015) |
| 98 | -20, -140 | † | +20, +120 | † | † | † | WRF-Chem | NCP | Gao Y. et al. (2015) |
| 99 | † | † | † | † | † | † | WRF-Chem | SWC | Fan et al. (2015) |
| 100 | -11.03, -9.84, -5.84, -12.37 | † | † | † | † | † | WRF-Chem | NCP | Chen et al. (2015) |
| 101 | † | † | † | † | † | † | WRF-Chem | EC | Zhang et al. (2015) |
| 102 | † | † | † | † | † | † | WRF-Chem | EA | Wu et al. (2013) |
| 103 | † | † | † | † | † | † | WRF-Chem | India | Beig et al. (2013) |
| 104 | † | † | † | † | † | † | WRF-Chem | NCP | Jia et al. (2012) |
| 105 | † | † | † | † | † | † | WRF-Chem | EA | Zhang et al. (2012) |
| 106 | † | † | † | † | † | † | WRF-Chem | China | Gao et al. (2012) |
| 107 | † | † | † | † | † | † | WRF-Chem | MRYR | Bai et al. (2020)† |
| 108 | † | † | † | † | † | † | WRF-Chem | YRD | Liu et al. (2019) |
| 109 | -7.5 | † | † | † | † | † | WRF-Chem | EA | Wang K. et al. (2018) |
| 110 | † | † | † | † | † | † | WRF-Chem | EA | Su et al. (2018a) |
| 111 | -2.19, -1.94 | +1.44, +1.19 | +1.56, +1.44 | -1.26, -0.88 | -0.63, -0.49 | +0.18, +0.31 | WRF-Chem | EA | Su et al. (2018b) |
| 112 | -86, -94.5 | † | † | † | † | † | WRF-Chem | NEA | Park et al. (2018) |
| 113 | † | † | † | † | † | † | WRF-Chem | EC | Gao and Zhang (2018) |
| 114 | † | † | † | † | † | † | WRF-Chem | SEC | Shen et al. (2017) |
| 115 | † | † | † | † | † | † | WRF-Chem | China | Zhao et al. (2017) |
| 116 | † | † | † | † | † | † | WRF-Chem | India | Bhattacharya et al. (2017) |
| 117 | † | † | † | † | † | † | WRF-Chem | PRD | Jiang et al. (2016) |
| 118 | -5.4 | +0.9, +20.1 | † | † | † | † | WRF-Chem | EA | Zhang et al. (2015b) |
| 119 | † | † | † | † | † | † | WRF-Chem | India | Sarangi et al. (2015) |
| 120 | -12 | † | † | † | † | † | WRF-Chem | EA | Zhang et al. (2014) |
| 121 | † | † | † | † | † | † | WRF-Chem | EC | Lin et al. (2014) |
| 122 | † | † | † | † | † | † | WRF-Chem | SEC | Bennartz et al. (2011) |
| 123 | † | † | † | † | † | † | WRF-Chem | China | Zhong et al. (2019) |

| # | | | | | | | Model | Region | Reference |
|---|---|---|---|---|---|---|---|---|---|
| 124 | † | † | † | † | † | † | WRF-Chem | India | Conibear et al. (2018a) |
| 125 | † | † | † | † | † | † | WRF-Chem | India | Conibear et al. (2018b) |
| 126 | † | † | † | † | † | † | WRF-Chem | India | Ghude et al. (2016) |
| 127 | † | † | † | † | † | † | WRF-Chem | NCP | Gao M. et al. (2015) |
| 128 | † | † | † | † | -5, -9, -10, -20 | † | WRF-CMAQ | EA | Dong et al. (2019) |
| 129 | † | † | † | † | † | † | WRF-CMAQ | NEA | Jung et al. (2019) |
| 130 | -10.98, -17.8, -4.31 | † | † | † | † | † | WRF-CMAQ | EA | Nguyen et al. (2019a) |
| 131 | -16.47, -22.54, -15.63, -12.99, -14.71 | † | † | † | † | † | WRF-CMAQ | SEA | Nguyen et al. (2019b) |
| 132 | -50 | † | † | † | † | † | WRF-CMAQ | NEA | Yoo et al. (2019) |
| 133 | † | † | † | † | † | † | WRF-CMAQ | EA | Sekiguchi et al. (2018) |
| 134 | -7.5, -7, -21.8 | † | † | † | † | † | WRF-CMAQ | EA | Hong et al. (2017) |
| 135 | † | † | † | † | † | † | WRF-CMAQ | China | Xing et al. (2017) |
| 136 | † | † | † | † | † | † | WRF-CMAQ | EA | Xing et al. (2016) |
| 137 | † | † | † | † | † | † | WRF-CMAQ | EC | Xing et al. (2015a) |
| 138 | † | † | † | † | † | † | WRF-CMAQ | EC | Xing et al. (2015b) |
| 139 | -9.9, -13 | † | † | † | -4.9, -6.5 | † | WRF-CMAQ | EC | Xing et al. (2015c) |
| 140 | -32.41, -37.04 | † | † | † | † | † | WRF-CMAQ | China | Wang et al. (2014) |
| 141 | -23.9, -16.6, -19.9 | † | +19.1, +10.8, +14.7 | † | † | † | WRF-CMAQ | China | Chen et al. (2019b) |
| 142 | † | † | † | † | † | † | WRF-CMAQ | China | Chang et al. (2018) |
| 143 | † | † | † | † | † | † | WRF-CMAQ | EA & India | Hong et al. (2019) |
| 144 | † | † | † | † | † | † | GRAPES-CUACE | NCP | Wang et al. (2017) |
| 145 | † | † | † | † | † | † | GRAPES-CUACE | EC | Wang H. et al. (2018) |
| 146 | † | † | † | † | † | † | GRAPES-CUACE | EA | Wang et al. (2013a) |
| 147 | -45.1 | +12.2 | † | † | -23.9 | +6 | GRAPES-CUACE | EA | Wang et al. (2013b) |
| 148 | † | † | † | † | † | † | GRAPES-CUACE | NCP | Zhou et al. (2012) |
| 149 | -10, -80, -200, -233 | † | † | † | -120, -140, -20, -60 | † | GRAPES-CUACE | EA | Wang et al. (2010) |
| 150 | † | † | † | † | † | † | GRAPES-CUACE | EC | Zhou et al. (2016) |
| 151 | † | † | † | † | † | † | WRF-NAQPMS | EA | Li J. et al. (2018) |
| 152 | -23.9 | † | † | † | † | † | WRF-NAQPMS | NCP | Wang et al. (2014) |
| 153 | † | † | † | † | † | † | WRF-NAQPMS | EC | Wang et al. (2014) |
| 154 | † | † | † | † | † | † | GATOR-GCMOM | NEA | Ten Hoeve and Jacobson, 2012 |
| 155 | † | † | † | † | † | † | GATOR-GCMOM | India | Jacobson et al. (2019) |
| 156 | † | † | † | † | † | † | GATOR-GCMOM | NCP | Jacobson et al. (2015) |
| 157 | † | † | † | † | † | † | Multi-model comparison | EA | Chen et al. (2019a) |
| 158 | † | † | † | † | † | † | Multi-model comparison | EA | Li et al., (2019) |
| 159 | † | † | † | † | † | † | Multi-model comparison | NCP | Gao et al. (2018a) |
| 160 | † | † | † | † | † | † | Multi-model comparison | India | Govardhan et al. (2016) |

†: Unclear. *: A preprint version of this study was available online on October 31, 2019, and was formally published on January 1, 2020.

(EA: East Asia, NEA: Northeast Asia, SEA: Southeast Asia, EC: East China, NCP: North China Plain, YRD: Yangtze River Delta, SEC: Southeast China, NWC: Northwest China, TP: Tibetan Plateau, MRYR: middle reaches of the Yangtze River, SWC: Southwest China; PRD: Pearl River Delta).

9. Overall, it is difficult to determine what the main scientific takeaways from the paper are. Are the existing models sufficient to provide reliable estimates going forward?

What are the main limitations and strengths of the models?

Response: According to your suggestion, we further discussed with other co-authors and here concisely summarized three takeaways as follows:

(1) Enabling aerosol feedbacks in two-way coupled models could improve their simulation/forecast capabilities of meteorology and air quality in Asia.

(2) Meta-analysis results showed that a wide range of differences exist among the previous studies due to various model configurations (selections of model versions and parameterization schemes). Projects covering more comprehensive intercomparisons of two-way coupled models need to be conducted in Asia.

(3) Large uncertainties mainly exist in ACI processes, and more investigations should be conducted by the modeling community in the future.

The two-way coupled models serve as a powerful tool for investigating how aerosols interacting with meteorology and the associated physiochemical processes, which is not possible with offline models. Our bibliometric and meta- analysis results revealed that the current two-way coupled models can sufficiently simulate surface meteorological and chemical variables but may not be able to accurately simulate variables affected by ACI effects. For numerical representations of ACI processes in coupled models, large uncertainties exist in cloud microphysics, cumulus cloud and ice nucleation parameterizations, and recent advances of observational studies have not been implemented into coupled models. At the same time, turning on aerosol feedbacks could lead to higher computational cost compared to offline models, but this shortcoming can be overcome with the new developments of cluster computing technology (i.e., GPU-accelerated computing and cloud computing). All of above assessments are reflected in the revised Conclusion section:

[revised manuscript text omitted]